# Discovery of FoTO1 and Taxol genes enables biosynthesis of baccatin III

Conor James McClune[1,2,7], Jack Chun-Ting Liu[3,7], Chloe Wick[1], Ricardo De La Peña[1], Bernd Markus Lange[4], Polly M. Fordyce[5,6] & Elizabeth S. Sattely[1,2✉]

Plants make complex and potent therapeutic molecules[1,2], but sourcing these molecules from natural producers or through chemical synthesis is difficult, which limits their use in the clinic. A prominent example is the anti-cancer therapeutic paclitaxel (sold under the brand name Taxol), which is derived from yew trees (*Taxus* species)[3]. Identifying the full paclitaxel biosynthetic pathway would enable heterologous production of the drug, but this has yet to be achieved despite half a century of research[4]. Within *Taxus*' large, enzyme-rich genome[5], we suspected that the paclitaxel pathway would be difficult to resolve using conventional RNA-sequencing and co-expression analyses. Here, to improve the resolution of transcriptional analysis for pathway identification, we developed a strategy we term multiplexed perturbation × single nuclei (mpXsn) to transcriptionally profile cell states spanning tissues, cell types, developmental stages and elicitation conditions. Our data show that paclitaxel biosynthetic genes segregate into distinct expression modules that suggest consecutive subpathways. These modules resolved seven new genes, allowing a de novo 17-gene biosynthesis and isolation of baccatin III, the industrial precursor to Taxol, in *Nicotiana benthamiana* leaves, at levels comparable with the natural abundance in *Taxus* needles. Notably, we found that a nuclear transport factor 2 (NTF2)-like protein, FoTO1, is crucial for promoting the formation of the desired product during the first oxidation, resolving a long-standing bottleneck in paclitaxel pathway reconstitution. Together with a new β-phenylalanine-CoA ligase, the eight genes discovered here enable the de novo biosynthesis of 3'-*N*-debenzoyl-2'-deoxypaclitaxel. More broadly, we establish a generalizable approach to efficiently scale the power of co-expression analysis to match the complexity of large, uncharacterized genomes, facilitating the discovery of high-value gene sets.

Plants defend themselves with complex chemical arsenals that are an essential source of therapeutics[1,2]. Paclitaxel (Taxol), a potent microtubule-stabilizing agent discovered in yew (*Taxus*) plants as part of a National Cancer Institute screening campaign during the 1960s and 1970s, remains one of the most valuable chemotherapeutics used in the clinic[6]. After its approval by the US Food and Drug Administration (FDA) in 1992 for the treatment of ovarian cancer, this diterpenoid became a best-selling pharmaceutical and remains the active component of diverse formulations, derivatizations and biological conjugates[7]. This extensive use, combined with the chemical complexity and low natural abundance (0.001–0.050% dried weight in *Taxus* bark)[7,8] of Taxol has made it one of the most sought-after molecules for synthesis. Although elegant synthetic routes have been developed[9], none are economically viable; drug supply still relies on extracting late-stage intermediates, such as baccatin III (**16**), from yew tissue[10]. The promise of a biomanufacturing strategy has made the discovery of the complete *Taxus* enzyme set for heterologous Taxol biosynthesis a grand challenge for natural product chemistry.

The search for the complete Taxol biosynthetic gene set, which was originally proposed to involve 19 enzymes, including 14 enzymes to baccatin III (**16**), began in the late 1990s. By 2006, the Croteau laboratory and others had discovered 12 enzymes, including the scaffold-forming enzyme, taxadiene synthase (TDS), as well as several tailoring oxidases and acyltransferases (Fig. 1a and Supplementary Table 1). Progress mostly stalled for two decades, until recent reports identified a taxane oxetanase (TOT) that installs Taxol's unique oxetane moiety, and several additional enzymes that are thought to act in the pathway[11–15]. However, several of the crucial functional groups on Taxol, such as the C-1β hydroxyl, still lack an assigned enzyme with direct biochemical evidence. In addition to missing pathway enzymes, heterologous reconstitution of the Taxol pathway has been stymied by the inefficiency of the first proposed oxidation. Despite extensive troubleshooting efforts in a variety of heterologous systems[16], the first Taxol oxidase, taxadiene 5α-hydroxylase, (T5αH), mainly produces side products with rearranged carbon bonds instead of the proposed 'on-pathway'

[1]Department of Chemical Engineering, Stanford University, Stanford, CA, USA. [2]Howard Hughes Medical Institute, Stanford University, Stanford, CA, USA. [3]Department of Chemistry, Stanford University, Stanford, CA, USA. [4]Institute of Biological Chemistry, Washington State University, Pullman, WA, USA. [5]Department of Bioengineering, Stanford University, Stanford, CA, USA. [6]Department of Genetics, Stanford University, Stanford, CA, USA. [7]These authors contributed equally: Conor James McClune, Jack Chun-Ting Liu. ✉e-mail: sattely@stanford.edu

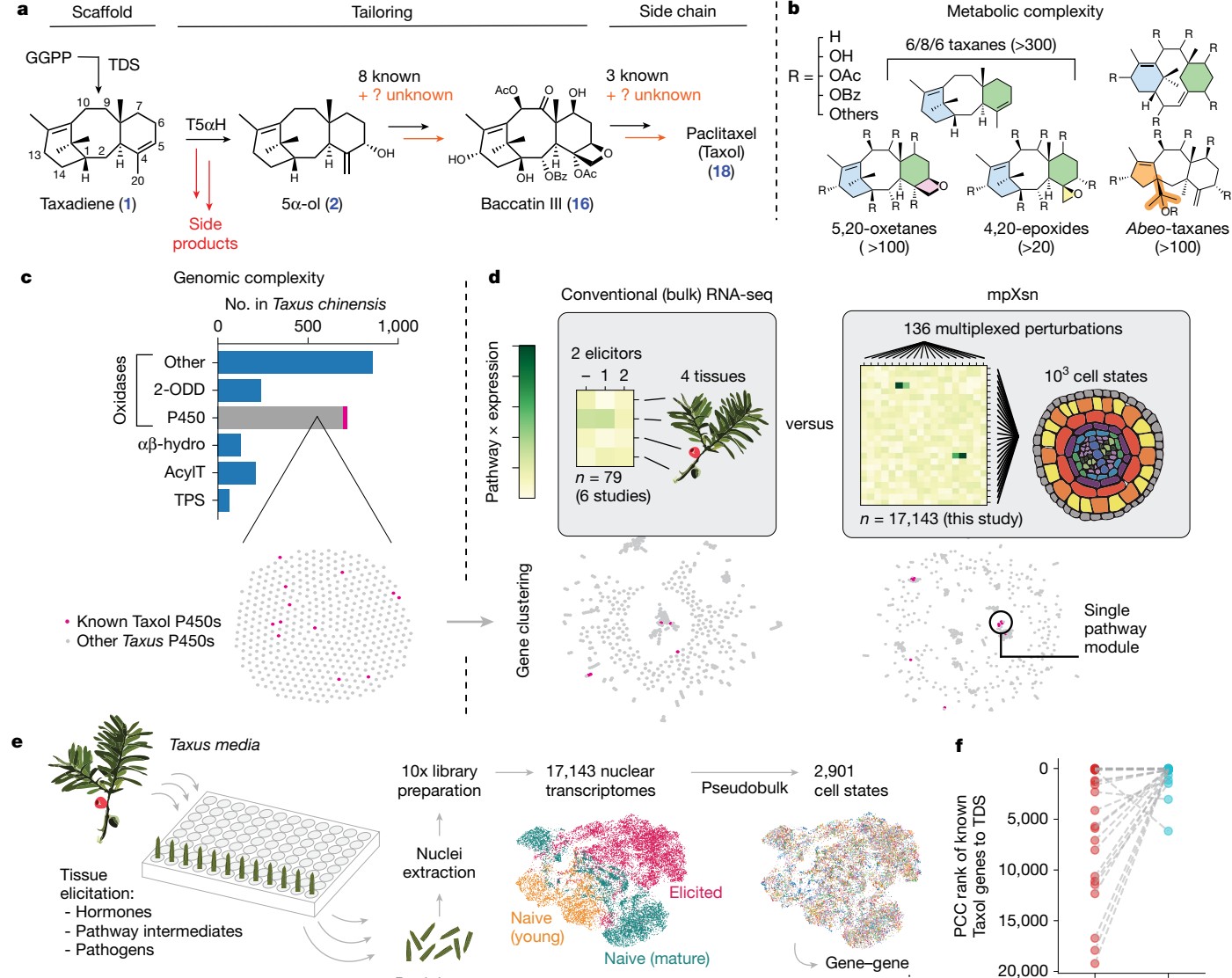

**Fig. 1 | A platform combining multiplexed perturbation and snRNA-seq (mpXsn) to overcome the challenges of Taxol biosynthetic gene discovery. a**, Proposed Taxol biosynthesis pathway, with gaps highlighted in orange. In addition to our incomplete knowledge of the biosynthetic gene set, inefficiencies (red arrows) of the first oxidase, T5αH, prevent reconstitution and discovery of the Taxol pathway. **b**, Prominent classes of taxane metabolites that have been isolated from *Taxus* species. The tailoring acyl groups on taxanes include acetyl, benzoyl, short-chain fatty acid residues and phenylisoserine derivatives. **c**, Number of enzymes in the *Taxus chinensis* genome belonging to secondary-metabolism-related families. P450, cytochrome P450; αβ-hydro,

α/β-hydrolase; AcylT, acyltransferase; TPS, terpene synthase. **d**, Overview of the differences between conventional co-expression approaches and the mpXsn methodology described here. Dot networks are visualizations of the co-expression network, in which nodes are linked when mutual rank is lower than 20, using either bulk RNA-seq or our mpXsn data. For visual clarity, only P450s are shown. **e**, Experimental overview of mpXsn, with uniform manifold approximation and projection (UMAP) of single-nucleus transcriptomes. **f**, Rank of each known Taxol gene by PCC to TDS using either bulk (*n* = 79 samples) or mpXsn (*n* = 17,143 cells across 3 experiments) data.

intermediate, taxadien-5α-ol (**2**) (refs. 17–20). These two challenges highlight the major gap in our understanding of the endogenous biochemistry, and remain hurdles to heterologous Taxol production (Fig. 1a).

The missing components of the Taxol pathway have probably eluded scientists because of the metabolic and genomic complexity of *Taxus*. Taxol is one of almost 600 taxanes that have been isolated from *Taxus* species[7,21], including hundreds of 6/8/6-taxanes that differ from Taxol only by subtle tailoring modifications[7] (Fig. 1b). Within the *Taxus* genome, Taxol pathway genes are a minute fraction of the hundreds of oxidases, acyltransferases and other enzymes that are involved in the primary and secondary metabolism[5] (Fig. 1c). Consequently, despite extensive transcriptional profiling of *Taxus* tissues[5,22] (see also the Yew

Genomics Resource; http://langelabtools.wsu.edu/ygr/), conventional transcriptional co-expression approaches have not identified the complete Taxol biosynthetic gene set. We hypothesized that an updated approach, which captured a much larger diversity of transcriptional states, would be required to improve co-expression resolution and discriminate between different branches of metabolism (Fig. 1d). Such improvements in resolution would be especially important if the Taxol pathway uses unanticipated genes.

To improve gene-association resolution within the taxane metabolic network, we developed a single-nucleus approach to efficiently profile *Taxus* cell states across a vast set of cell types and perturbations. Differential transcriptional activation of *Taxus*'s diverse biosynthetic processes enabled the discovery of several Taxol

transcriptional modules, from which we identified eight new genes in Taxol biosynthesis. Highlighting the importance of this approach, most identified genes do not belong to previously proposed Taxol gene families, and would not have been anticipated from the proposed biosynthetic model. None of the three oxidases identified here belong to the cytochrome P450 CYP725A subfamily that has been the focus of previous search efforts[11–15]. Three other enzymes catalyse the addition and removal of cryptic acetylations absent in Taxol, but their inclusion is essential for progression through the pathway, akin to the protection–deprotection strategy used by chemists. Finally, we identified a protein from the NTF2-like family, not previously implicated in plant metabolism, that is crucial for high yields of the reconstituted pathway by resolving the inefficiency during the first taxane oxidation[17–20]. With these 8 genes, together with 11 that have been described previously, we constructed a total biosynthesis for the direct Taxol precursors baccatin III and 3′-N-debenzoyl-2′-deoxypaclitaxel in *N. benthamiana*. Without further optimization, our system heterologously produces baccatin III, the industrial semi-synthesis precursor for taxane therapeutics, at levels comparable with the natural abundance in yew, showcasing its tremendous potential as a sustainable source.

## Multiplexed perturbation improves resolution

First, we developed a single-nucleus RNA sequencing (snRNA-seq) protocol for *Taxus* (Methods) and assessed whether natural cell-type heterogeneity alone would be sufficient to identify new Taxol genes by co-expression. However, after profiling 6,077 cells from unelicited ('naive') mature tissues, we observed that several Taxol biosynthetic enzymes, including TDS, T5αH and 10-deacetylbaccatin III-10-O-acetyl transferase (DBAT), were not highly expressed in any cell (Extended Data Fig. 1a). This exemplifies one of the core challenges of using transcriptomes to find secondary-metabolism genes: tissues must be in a state of active biosynthesis to capture pathway-associated transcripts[23], but the search for such a tissue state can be difficult, because it might require a specific developmental age[24] or exposure to a specific biotic stressor[23].

To mitigate the difficulty of identifying biosynthetic cell states by individually testing large panels of perturbations, we made use of the scale of single-cell transcriptomics to develop a method we term multiplexed perturbation × single nuclei (mpXsn), which simultaneously tests hundreds of perturbations. Although single-cell transcriptomics have been developed into parallelized screens in mammalian systems[25–27], no comparable platform was available for plants. Unlike most large-scale perturbation technologies, we designed mpXsn to require no genetic tools, so that it would be generalizable to diverse, non-model species.

By pooling diverse tissues and conditions before a single snRNA-seq library synthesis step, individual sample processing is no longer limiting. This approach enabled us to affordably test a large number of samples (272), spanning conditions and time points, in a single experiment (Fig. 1e). To maximize the probability of activating biosynthetic states, we compiled a panel of hormones, microorganisms and other potential elicitors (Supplementary Table 2). We subjected both young and mature *Taxus media* needles to this panel for one to four days before pooling all tissues and time points for snRNA-seq library synthesis. Compared with the naive cell states we originally profiled, a subset of the elicited cell states now exhibited high expression of the early Taxol pathway (Extended Data Fig. 1a).

To determine whether these single-cell transcriptional data provide new information to identify Taxol enzymes, we directly compared co-expression analyses using either our mpXsn data or bulk RNA-seq data from six previous studies spanning tissues and elicitation conditions (Methods). Using either bulk RNA-seq datasets (79 samples) or the mpXsn data (2,901 pseudobulk cell states), we ranked each gene

in the *Taxus* genome by Pearson correlation coefficient (PCC) to TDS (Fig. 1f). Of the 14 genes previously associated with Taxol biosynthesis (Supplementary Table 1), all but 2 ranked higher in the mpXsn analysis than in the bulk RNA-seq analysis (Fig. 1f and Extended Data Fig. 1b–d). This suggested that the Taxol pathway was better resolved in the mpXsn dataset than in compiled bulk RNA-seq datasets.

## Identification of three Taxol biosynthetic modules

The Taxol pathway has been hypothesized to involve 19 transformations, and at least 13 enzymes have been characterized (Fig. 2a and Supplementary Table 1). Although the expression of the first enzyme in the pathway, TDS, does correlate with most Taxol genes (Fig. 1f and Extended Data Fig. 1c–e), some known Taxol genes show stronger co-expression relationships with one another, forming distinct subclusters of co-expression (Fig. 2b). These subclusters prompted us to analyse the mpXsn dataset with an untargeted approach to identify gene co-expression modules across the *T. media* transcriptome.

To systematically organize genes into co-expressed modules, we factored the gene-by-cell matrix from the mpXsn dataset using a consensus non-negative matrix factorization (cNMF) approach (Fig. 2c) that was previously developed for single-cell datasets[28]. The modules and corresponding gene scores produced by this approach can reveal patterns of corresponding gene expression that are not readily apparent from linear correlation analysis (Methods). We ran cNMF analysis with different numbers of total modules (range 50–400). In all runs with more than 125 modules, known Taxol genes consistently dominated not one, but three separate gene modules (subsequently called Taxol modules 1, 2 and 3) (Fig. 2d,e,h,i and Supplementary Fig. 1). We proceeded with an intermediate value (200 total modules) for subsequent analysis to avoid over- or under-clustering. The observation that different subsets of Taxol genes rank highly in separate modules, roughly segregating by proposed order of biosynthesis (Fig. 2e and Supplementary Data 1), suggests that Taxol biosynthesis consists of separately regulated transcriptional programs. Furthermore, Taxol module 1 is enriched in genes of the methylerythritol phosphate (MEP) pathway, but not the mevalonate pathway, highlighting a link between primary and secondary metabolism (Fig. 2f). This finding aligns with the current consensus that the MEP pathway supplies precursors for diterpenoids in gymnosperms[29].

The enrichment of P450, 2-oxoglutarate dependent dioxygenase (2-ODD) and acetyltransferase genes among the top 100 genes of the 3 Taxol modules (Fig. 2g) suggests that these modules are involved in *Taxus* secondary metabolism. In addition, these three modules were expressed in different subsets of cells (Fig. 2h). Elicitation was crucial for activating module 1, consisting of the early Taxol pathway, because it was not expressed in the naive young and mature *Taxus* tissues we profiled (Fig. 2h). Indeed, metabolomic analysis suggests that elicitors such as chitosan and methyl jasmonate lead to increased accumulation of an early pathway intermediate in *Taxus* needles (Supplementary Fig. 2), in line with previous reports that these are elicitors of gymnosperm stress responses[30] and taxane biosynthesis[31]. An unfiltered analysis of the top genes in Taxol module 1 revealed all the genes that we had previously used to reconstitute the early Taxol pathway[18], including TDS, T5αH, taxadien-5α-ol-O-acetyltransferase (TAT), taxane 10β-hydroxylase (T10βH) and DBAT (Fig. 2i). We therefore began our search for the missing components of the Taxol pathway by examining the uncharacterized genes in this highly co-expressed gene module.

## Discovery of FoTO1

The first oxidation in the Taxol pathway, by T5αH, is highly inefficient and yields a large set of closely related products, most of which do not seem to be productive pathway intermediates. Numerous studies, including our own, have investigated and attempted to optimize

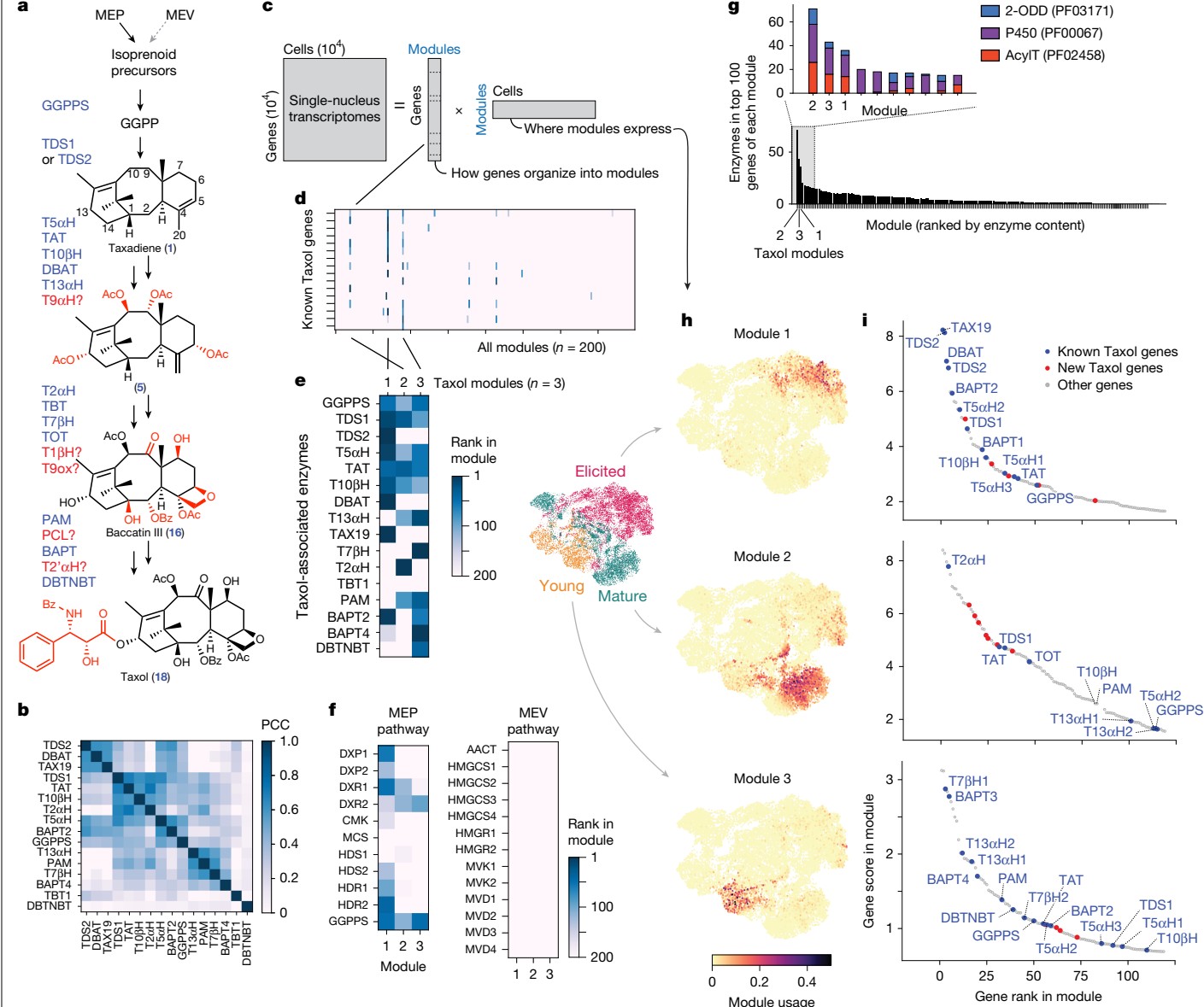

**Fig. 2 | Identification of taxane biosynthetic gene modules. a**, Schematic of Taxol biosynthesis and previously hypothesized gene order. Blue, previously identified Taxol biosynthesis enzymes; red, hypothesized enzymes. **b**, PCC between known Taxol-related genes using mpXsn data. To identify substructures, genes were hierarchically clustered (SciPy fcluster, Euclidean distance) on both axes. **c**, Schematic for matrix factorization. mpXsn data were factorized using cNMF[28]. **d**, Heat map showing the rank of known Taxol biosynthetic genes in each of the modules produced by matrix factorization. **e**, As in **d**, but showing only the three modules enriched in Taxol genes (modules 1, 2 and 3). **f**, Heat map of Taxol modules, showing module rankings for the two isoprenoid pathways in the primary metabolism potentially upstream of the Taxol biosynthesis.

Only the MEP pathway is co-expressed with the first Taxol module, supporting its role in synthesizing Taxol precursors. **g**, All gene modules ranked by the total number of 2-ODD, P450 and acetyltransferase (AcylT) genes in the top 100 genes of each module. **h**, Module usage of each cell, which is analogous to gene expression, plotted onto the single-nucleus transcriptomic UMAP. Taxol modules 1–3 are expressed in non-overlapping cell states, and were mainly identified in different experiments. **i**, Unfiltered lists of the top genes in each module, plotted as module rank and score. Blue, previously identified genes associated with Taxol biosynthesis; red, new biosynthetic genes identified in this study.

T5αH across various contexts[16–20]. Previously, we reported[18] the de novo reconstitution of the six early steps in Taxol biosynthesis, consisting of TDS, T5αH, TAT, T10βH, DBAT and taxane 13α-hydroxylase (T13αH), in *N. benthamiana*; this resulted in the production of 5α,10β-diacetoxytaxadien-13α-ol (**4**) (Fig. 3a). This required extensive tuning of the expression of T5αH, which increased yields of the on-pathway intermediate taxadien-5α-ol (**2**) (ref. 32) relative to cyclotaxane (OCT; **2'a**), iso-OCT (**2'b**) and other rearranged side products (**2'c**). Despite our optimization efforts[18], these side products still accumulated as the dominant products of T5αH. Notably, OCT-derived products do not naturally accumulate to notable levels in *Taxus* plants.

We hypothesized that Taxol module 1 contains unanticipated proteins that facilitate this initial oxidation and prevent the formation of side products. During our testing of approximately 77 gene candidates that were highly ranked in Taxol module 1 (Fig. 3b), we found that the addition of a single protein, a NTF2-like protein ranked 13 in module 1, increased the yield of our pathway (Supplementary Fig. 3). We hypothesized that this protein is involved in altering the product flux of the early oxidative steps. Expressing this gene, which we later named FoTO1 (facilitator of taxane oxidation), resulted in a 10–17-fold increase in yields from early Taxol pathways (Fig. 3c). To determine whether FoTO1 ameliorates side-product formation during the first

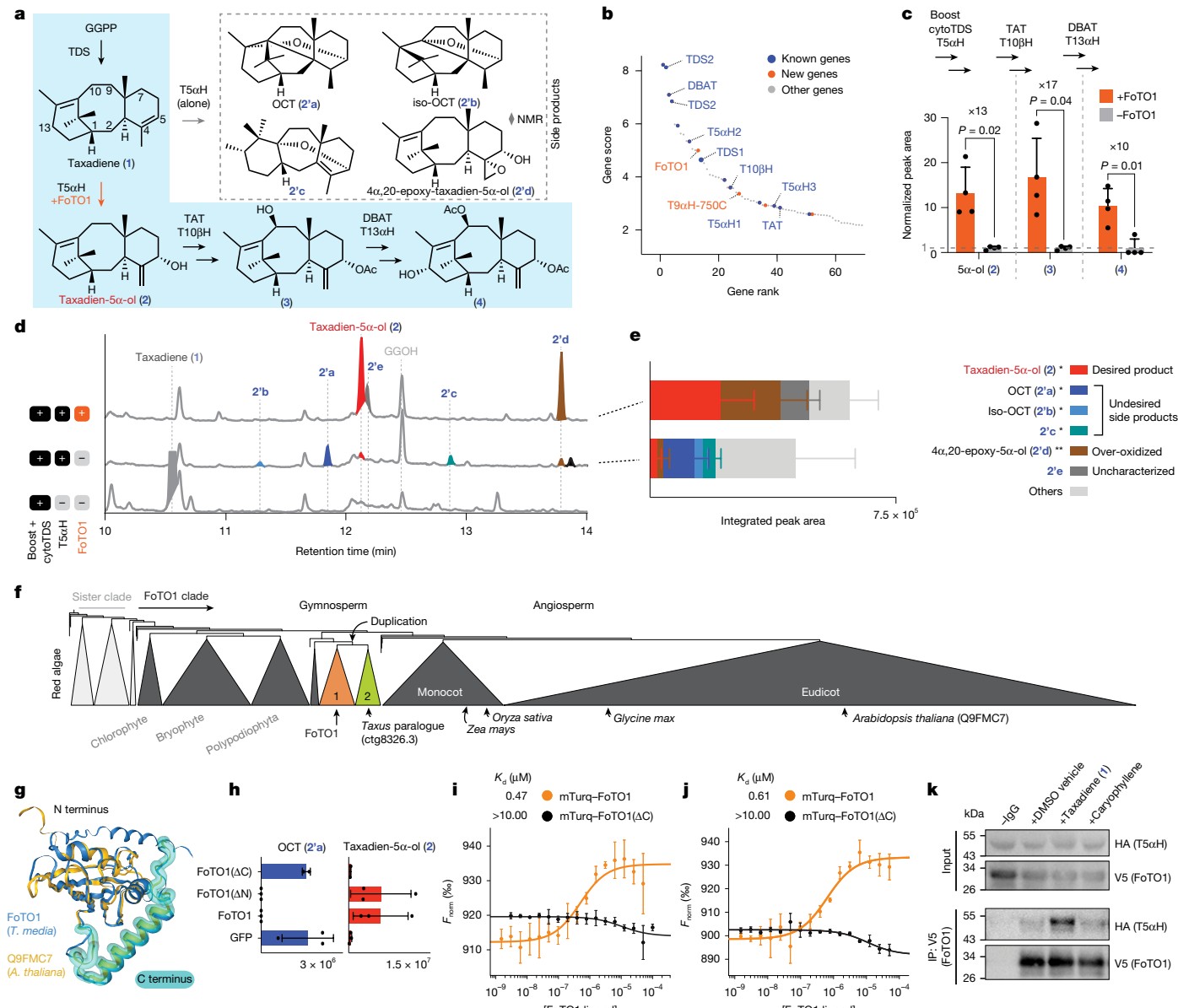

**Fig. 3 | Characterization of FoTO1. a**, Early Taxol biosynthetic pathway and the T5αH product divergence. Blue shading highlights the biosynthetic pathway towards Taxol. Diamond indicates the structure is supported by NMR. **b**, Rank and score of genes in Taxol module 1. **c**, Bar graph showing the FoTO1-induced fold change in end-products' peak area of subpathways when transiently expressed in *N. benthamiana* leaves. Fold change is calculated by quantifying the GC–MS total ion chromatogram (TIC) peak area of compounds **2**–**4** and normalizing to the −FoTO1 condition. cytoTDS, cytosolic TDS. Data are mean ± s.d., *n* = 3 biological, independent leaf samples. Statistical analyses were performed using a two-sided, unpaired Welch's *t*-test. **d**, GC–MS TIC of *N. benthamiana* leaves transiently expressing the indicated genes. **e**, Bar graph of total oxidized taxanes for the +T5αH and +T5αH+FoTO1 conditions. Data are mean ± s.d. *n* = 6 biological, independent leaf samples. One asterisk indicates previously characterized; two asterisks indicate characterized in this study.

**f**, Phylogenetic tree of FoTO1 homologues identified by HMMER. The tree was produced with FastTree, rooted with red algae homologues. **g**, Structural model of FoTO1, generated by AlphaFold3 and aligned with the *Arabidopsis thaliana* orthologue with FoldSeek. **h**, Bar graphs showing the integrated peak area of **2** and **2'a** when N- or C-terminally truncated FoTO1 is transiently expressed in *N. benthamiana* leaves together with TDS and T5αH. Data are mean ± s.d., *n* = 3 biological replicates. **i**, Quantification of binding between purified T5αH and FoTO1 or FoTO1(ΔC) using microscale thermophoresis. Data are mean ± s.d., *n* = 3 replicates. **j**, Quantification of binding between purified TDS2 and FoTO1 or FoTO1(ΔC), as in **i** (*n* = 3 replicates). N-terminal transmembrane domains of T5αH and TDS2 are removed for purification purposes. Data are mean ± s.d., *n* = 3. **k**, Immunoblot of the co-IP of T5αH–HA (prey) by V5–FoTO1 (bait) in *N. benthamiana* leaves expressing both proteins (Supplementary Fig. 7).

oxidation, we compared gas chromatography–mass spectrometry (GC–MS) analyses of *N. benthamiana* leaf extracts expressing TDS and T5αH with and without FoTO1. Without FoTO1, T5αH yields mainly undesired side products, including **2'a**–**c** and many uncharacterized compounds, and very little of the desired product, taxadien-5α-ol (**2**) (Fig. 3d,e). However, inclusion of FoTO1 markedly alters the product profile: side products **2'a**–**c** are no longer produced, and taxadien-5α-ol (**2**)

and an over-oxidized product, 4β,20-epoxy-taxadien-5α-ol (**2'd**, Supplementary Note 1), become the major products (Fig. 3d,e and Supplementary Fig. 4).

To gain insight into how FoTO1 modulates T5αH product formation, we further analysed its sequence and phylogeny. FoTO1 is a 195-amino-acid protein in the NTF2-like family, which has not previously been implicated in plant metabolism. Although some fungal

NTF2 proteins have evolved catalytic activity[33,34], plant and animal NTF2 proteins have been studied mainly for their capacity to mediate protein transport to the nucleus[35]. Using HMMER[36], we identified 1,957 homologues in plant genomic databases. We found that FoTO1 homologues are widespread across Viridiplantae, and generally present as a single copy per genome, suggesting a conserved function (Fig. 3f). Gymnosperms, however, contain multiple paralogues that derive from both an ancient duplication (Fig. 3f) and recent duplications, including in the genus *Taxus* (Supplementary Fig. 5). These duplications might have allowed one paralogue to evolve alternative functions and contribute to taxane biosynthesis. Supporting this functional divergence, we found that neither the FoTO1 paralogue from *T. media* nor the homologue from *Arabidopsis thaliana* could produce the same metabolomic change for early Taxol pathway reconstitution as FoTO1 (Fig. 3g and Extended Data Fig. 2).

We anticipated that FoTO1 could be operating through several mechanisms, including (i) scaffolding or allosteric support of Taxol enzymes, (ii) transport or positioning of taxane intermediates or (iii) enzymatic resolution of an unstable intermediate. FoTO1 does not affect the production of taxadiene (**1**) and iso-taxadiene by TDS1 or TDS2 (Extended Data Fig. 3a), suggesting that it has no enzymatic activity on taxadiene (**1**). To further test for potential active sites that could be important for catalysis or substrate binding, we generated mutations of residues within the protein's cavity on the basis of the AlphaFold-predicted structure of FoTO1 (Fig. 3g and Extended Data Fig. 3b–d). Although none of the tested amino acid substitutions caused FoTO1 to lose its capacity to suppress oxidation side products (Extended Data Fig. 3b–d), deletion of the C-terminal α-helix, but not the N-terminal helix, eliminated FoTO1's in planta phenotype (Fig. 3g,h).

To determine whether the function of FoTO1 involved a direct interaction with Taxol enzymes and perhaps a scaffolding role, we purified FoTO1 (fused to N-terminal mTurquoise2) and the soluble portions of TDS2 and T5αH. Using microscale thermophoresis, we found that FoTO1 binds to both T5αH and TDS2 with high nanomolar dissociation constant ($K_d$) values (Fig. 3i,j). Deletion of the C-terminal helix, which disrupts FoTO1's in planta metabolic phenotype (Fig. 3g), also eliminated binding affinity with both proteins (Fig. 3i,j), suggesting that this region is involved in protein–protein interactions. To determine whether this physical interaction was physiologically relevant, we conducted co-immunoprecipitation (co-IP) of epitope-tagged proteins, V5–FoTO1 and T5αH–HA, co-expressed in *N. benthamiana* leaves (Fig. 3k). Immunoprecipitation of V5–FoTO1 using a V5 antibody was able to capture T5αH–HA (Fig. 3k and Supplementary Figs. 6 and 7). Furthermore, the T5αH–HA co-IP signal increased fourfold when leaf lysates were incubated with 45 μM taxadiene (**1**), but not when they were incubated with a mock terpene (caryophyllene) (Fig. 3k and Supplementary Figs. 6 and 7).

Together, these data support a mechanism involving a direct interaction between FoTO1, T5αH and possibly TDS. The influence of taxadiene (**1**) on the efficiency of co-IP of T5αH and FoTO1 could indicate that the metabolite has a direct role in this interaction. Although TDS, T5αH and FoTO1 localize to different subcellular regions—plastid, endoplasmic reticulum (ER) membrane and cytoplasm, respectively (Supplementary Fig. 8)—direct contacts between the outer lamina of these membranes are known sites of lipid trafficking[37] and the biosynthesis of other diterpenoids such as gibberellin[38].

## Independently evolved T9αHs for different pathways

The presence of FoTO1 and all biosynthetic enzymes for **4** in gene module 1 suggested a coordinated regulation of early Taxol biosynthetic enzymes, and an opportunity to discover missing Taxol enzymes from this module (Fig. 3b). The next oxidation after C-5α, C-10β and C-13α hydroxylation is proposed to be the C-9α hydroxylation. In *N.*

*benthamiana*, we screened 2-ODD and P450 oxidases within the top 50 genes of module 1 by co-expressing candidate genes in batches together with the upstream pathway to **4**, a taxane with three oxidation and two acylation modifications (3O2A; hereafter, the nomenclature *n*O*m*A describes the collection of taxane isomers bearing *n* oxy groups, for example, hydroxylation and epoxidation, and *m* acylations, for example, acetylation and benzoylation, on the taxadiene scaffold). We found that the 27th gene in module 1, a P450 in the CYP750C family (T9αH-750C), resulted in depletion of 3O2A (**4**) and concurrent production of a mass corresponding to a new 4O3A (**5**) intermediate with an additional oxidation and an acetylation on **4** (Fig. 4b). This additional acetylation was unexpected given that the acetyltransferases expressed, TAT and DBAT, were known mainly for acetylation activity on the C-5 and C-10 hydroxyls, respectively. To confirm the function of T9αH-750C and to provide support for the structural assignment of **5**, we co-expressed the biosynthetic genes to **5** with TAX19, the previously characterized C-13α-*O*-acetyltransferase[39], and isolated the resulting acetylated 4O4A products from *N. benthamiana* leaves. Through nuclear magnetic resonance (NMR) analysis and tandem mass spectrometry (MS/MS) comparison with a standard, we structurally characterized the products as taxusin (**6**) and its isomer 13β-taxusin (**6'**) (Fig. 4b, Supplementary Table 3 and Extended Data Fig. 4a,b). This structural analysis supports the role of T9αH-750C as a taxane 9α-hydroxylase (T9αH) and TAT as a bi-functional C-5α/9α-*O*-acetyltransferase (Extended Data Fig. 4c and Supplementary Note 2).

Several groups have independently reported that a CYP725A P450, with less than 20% identity at the protein level to T9αH-750C, is the T9αH in Taxol biosynthesis (referred to here as T9αH-725A, for distinction). This enzyme was identified by transcriptome-informed screening of CYP725A genes, a *Taxus*-specific enzyme family that includes all previously known Taxol P450s[12–14]. Although both T9αH-750C and T9αH-725A can act as T9αH to produce taxusin (**6**) when TAX19 is present, only T9αH-750C can deplete 3O2A to yield 4O3A (Fig. 4b, Supplementary Fig. 9 and Extended Data Fig. 4a,b). By contrast, T9αH-725A is unable to produce 4O3A and seems to require a substrate with a C-13α acetoxy group. Extremely low sequence conservation between these enzymes suggests the independent evolution of T9αH activities in two distinct P450 families (CYP725A and CYP750C), and their different substrate specificities suggest that T9αH-725A is involved in the biosynthesis of C-13α-acetoxyl taxanes, whereas T9αH-750C is involved in the biosynthesis of C-13α-hydroxyl taxanes. Because Taxol and its precursor baccatin III (**16**) lack the C-13α-acetoxy moiety that is required for T9αH-725A, we used T9αH-750C, instead of T9αH-725A, as the 9α-hydroxylase for all subsequent taxane pathway reconstitutions (Fig. 4c and Supplementary Fig. 10).

## Discovery of T7AT and two T1βHs

The missing C-1β hydroxylation in Taxol is proposed to occur after functionalization by several known mid-pathway enzymes: taxoid 2α-hydroxylase (T2αH) and 7β-hydroxylase (T7βH)[40,41], taxane 2α-*O*-benzoyltransferase (TBT)[42] and taxane oxetanase (TOT)[11–13]. When we co-expressed T2αH and TBT with the pathway to **5**, we detected a mass feature corresponding to the expected hydroxylated and benzoylated product (**7-Bz**) (Fig. 4c). Notably, we also detected a mass feature corresponding to the acetylated product (Fig. 4c), suggesting that TBT also catalyses acetylation. Testing of TBT with 1-hydroxybaccatin I and baccatin VI showed that TBT mediates the interconversion between C-2α-benzoyl and acetyl groups (Supplementary Fig. 11), consistent with a previous report[43], and this might explain the prevalence of C-2α-acetoxy taxanes in nature[7,21].

Expression of T7βH with the 5O4A gene set yielded the expected benzoylated and acetylated 6O4A products, but subsequent addition of TOT did not convert these 6O4A compounds to a hepta-oxidized (7O) product (Fig. 4d). Given that many of the highly oxygenated taxanes

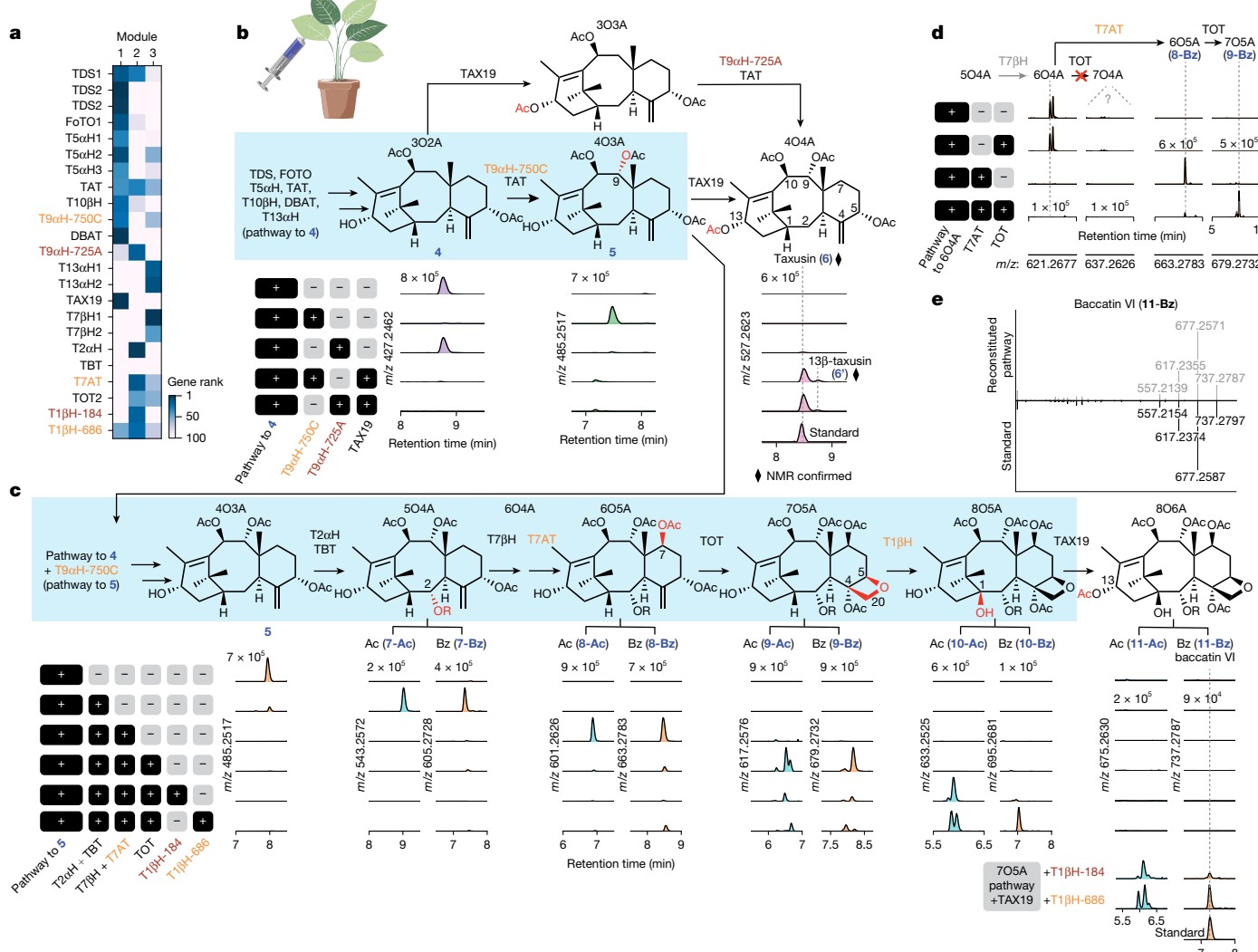

**Fig. 4 | Discovery and characterization of T9αH, T7AT and two T1βHs. a**, Heat map showing the ranks of new T9αH and T1βHs and other Taxol biosynthetic genes in the three modules. T9αH-725A is the T9αH independently reported by other groups[12–14]. Black, previously known Taxol genes. Orange, new Taxol pathway genes identified in this figure. Red, *Taxus* enzymes proposed to act in other taxane pathways. **b**, Proposed biosynthetic pathway from **4**, the latest intermediate we reported recently[18], to taxusin (**6**), and the corresponding extracted ion chromatograms (EICs) of products **4**–**6** when the indicated sets of genes were expressed in *N. benthamiana* leaves. Blue shading highlights the biosynthetic pathway towards Taxol, which only involves **4** and **5**. **c**, Proposed biosynthetic pathway from **5** to baccatin VI (**11-Bz**), and the corresponding EICs of intermediates when the indicated sets of genes were expressed in *N. benthamiana* leaves. Blue shading highlights the biosynthetic pathway

towards Taxol. Structures of **5**, **7-Ac**, **7-Bz**, **8-Ac**, **8-Bz**, **9-Ac**, **9-Bz**, **10-Ac**, **10-Bz** and **11-Ac** are proposed on the basis of the functions of enzymes previously characterized (TAT, TAX19, T2αH, TBT, T7βH and TOT) and described in this study (T9αH-750C, T7AT and T1βH). TAX19 is used to generate known 13-*O*-acetylated products, including taxusin (**6**) and baccatin VI (**11-Bz**), for structural analysis. Diamond indicates the structure is supported by NMR. **d**, EICs of expected products when the pathway to 6O4A is expressed with TOT, T7AT or both in *N. benthamiana*. In the absence of T7AT, no notable 6O4A depletion or product formation by TOT is observed. **e**, MS/MS fragmentation patterns of heterologously produced baccatin VI (**11-Bz**) in *N. benthamiana* compared with that of **11-Bz** standard. MS/MS fragmentations were generated using [M+Na]$^+$ (*m/z* = 737.2788) as the precursor ion and fragmented with a collision energy of 30 eV.

with an oxetane or epoxide moiety originating from TOT activity are also C-7β-*O*-acetylated[21] (for example, baccatin I, baccatin IV and baccatin VI; Supplementary Fig. 12), we reasoned that the installation of a C-7β-acetoxy group might be a prerequisite for TOT function. Therefore, we screened acyltransferase candidates from the top 30 genes of module 2 and found that the 15th gene in module 2, which we named taxane C-7β-*O*-acyltransferase (T7AT), was capable of various C-7β-*O*-acylations, including acetylation (Supplementary Fig. 13). Expression of T7AT with the upstream 6O4A pathway resulted in the acetylated and benzoylated 6O5A products, **8-Ac** and **8-Bz**, respectively (Fig. 4c). In contrast to 6O4A products (**7-Ac** and **7-Bz**), these 6O5A products are depleted after the addition of TOT, yielding dominant mass features that correspond to the hepta-oxidized products **9-Ac**

and **9-Bz** (Fig. 4c,d). These major peaks (**9-Ac** and **9-Bz**) are likely to correspond to the non-interchangeable epoxide and oxetane products generated by TOT (refs. 11–13). T7AT was independently reported in a recent publication, but its importance for TOT function has not been described[13].

After reconstituting a pathway to **9-Ac** and **9-Bz** (7O5A), we screened 37 oxidases (2-ODDs and P450s) of module 2 to identify the missing C-1β hydroxylase. This revealed two 2-ODDs (2-ODD184 and 2-ODD686, protein sequence identity 72%), the 20th and 24th genes of module 2, that yielded multiple mono-oxidized products when expressed with various upstream pathways (Fig. 4c, Supplementary Fig. 14 and Extended Data Fig. 5a–d). Expression of the 7O5A pathway and TAX19 (C-13α-*O*-acetyltransferase) with either 2-ODD184 and 2-ODD686

resulted in the production of baccatin VI (**11-Bz**), as confirmed by MS/MS comparison with the standard (Fig. 4c,e) as well as several 1β-hydroxybaccatin I isomers (**11-Ac** peaks; Supplementary Fig. 15). This result suggests that either 2-ODD can function as the missing taxane 1β-hydroxylase (T1βH), because all other functional groups in baccatin VI (**11-Bz**) can be explained by other enzymes included in the reconstitution. Of note, both 2-ODDs resulted in two major products with the 2α-*O*-acetylated pathways but only one major product with the 2α-*O*-benzoylated pathways (Fig. 4c and Extended Data Fig. 5a–d). Among the two major products in the 2α-*O*-acetylated pathways, one is presumably the 1β-hydroxylated product, but the other remains unidentified. Therefore, we isolated the two products from 2-ODD184 co-expressed with the taxusin (**6**) pathway in *N. benthamiana* and structurally characterized them as 1β-hydroxytaxusin (**6-O1**) and its structural isomer, 15-hydroxy-11(15→1)*abeo*-taxusin (**6-O2**) (Supplementary Tables 4–6 and Extended Data Fig. 5). We propose that the non-classical 11(15→1)*abeo*-taxane scaffold arises from a radical rearrangement associated with 2-ODD-mediated 1β-hydroxylation (Extended Data Fig. 5e). These data support a role for these 2-ODDs in the C-1β hydroxylation, and thus we hereafter refer to them as T1βH-184 and T1βH-686.

Together, these results reveal the discovery of an independently evolved T9αH, a T7AT important for the function of TOT, and two T1βHs, which allow us to reconstitute the biosynthesis of highly oxygenated taxanes (**10-Ac** and **10-Bz**) and their C-13α-acetoxy counterparts (**11-Ac** and **11-Bz**). T1βH-686 results in significantly higher levels of the 2α-*O*-benzoylated product **10-Bz** than does T1βH-184, which is desirable for Taxol production; thus, all subsequent pathway reconstitution was done with T1βH-686.

## Deacetylases and T9ox enable baccatin III biosynthesis

The structure of baccatin III (**16**), the direct precursor to Taxol before side-chain installation, suggests that it requires nine oxidations (seven hydroxylations, one oxetane formation and one ketone formation) and three acylations (two acetylations and one benzoylation) on the taxadiene (**1**) scaffold. Our latest intermediate **10-Bz** has two additional acetylations that are not found in baccatin III (**16**), and is lacking a C-9 ketone oxidation (Fig. 5a). The two additional acetylations come from (i) TAT that promiscuously *O*-acetylates the C-9-hydroxyl (Supplementary Note 2), and (ii) T7AT that *O*-acetylates the C-7-hydroxyl, which is crucial for the function of TOT (Fig. 4c,d). Although the role of these additional acetylations is unknown, we considered that they might serve as transient protecting groups during the biosynthesis—a strategy used in the biosynthesis of other plant terpenoids[44]—and might subsequently be removed by downstream deacetylases. To test for relevant deacetylases and identify the C-9 oxidase, we fed substrate baccatin VI (**11-Bz**) or 9-dihydro-13-acetylbaccatin III (9DHAB, **13**), which structurally resemble our latest intermediate **10-Bz**, to *N. benthamiana* leaves expressing top candidate genes to screen for desired activities.

Screening of 27 α/β-hydrolase candidates from modules 2 and 3 revealed two deacetylases (DeAc898 and DeAc1023) capable of the stepwise removal of two acetyls from baccatin VI (**11-Bz**) to yield a product matching a 9DHAB (**13**) standard (Supplementary Fig. 16), which lacks the C-7 and C-9 *O*-acetyl groups, as supported by MS/MS analysis (Supplementary Fig. 17a and Extended Data Fig. 6a). Expressing the full baccatin VI (**11-Bz**) pathway with the two deacetylases also resulted in the production of 9DHAB (Extended Data Fig. 6b). Screening of oxidase candidates identified a putative taxane C-9-oxidase (T9ox) in the 2-ODD family capable of oxidizing 9DHAB with the loss of two protons, presumably through ketone formation at C-9 (Supplementary Fig. 18). This gene has been independently reported by another group[14]. When we combined T9ox, DeAc898 and DeAc1023 with our 14-gene pathway to **10-Bz**, an abundant and notable new taxane product formed (Fig. 5b,c, Supplementary Fig. 17b and Extended Data Fig. 7a), which was subsequently isolated and confirmed as baccatin III (**16**), on the basis

of NMR and MS/MS analysis compared with a standard (Fig. 5d and Extended Data Fig. 7b). Stepwise assembly of the final steps of the pathway revealed that DeAc898 is a prerequisite for T9ox, suggesting that this enzyme hydrolyses the C-9 acetyl group (Fig. 5c). Consequently, we renamed DeAc898 and DeAc1023 as taxane 9α-*O*-deacetylase (T9dA) and taxane 7β-*O*-deacetylase (T7dA), respectively.

## Discovery of PCL for side chain biosynthesis

The side-chain installation and maturation of Taxol from baccatin III (**16**) is proposed to involve phenylalanine aminomutase (PAM), β-phenylalanine-CoA ligase (PCL), baccatin III:3-amino-3-phenylpropanoyl transferase (BAPT), taxane 2′α-hydroxylase (T2′αH) and 3′-*N*-debenzoyl-2′-deoxypaclitaxel-*N*-benzoyl transferase (DBTNBT)[22,45] (Fig. 5b and Supplementary Table 1). Although all five enzymes have been separately reported, we failed to observe Taxol when attempting to reconstitute complete biosynthesis by co-expressing these reported enzymes with our pathway in *Nicotiana*. Specifically, despite previous reports[14,22,46] that two separate *Taxus* acyl-activating enzymes (AAEs) can act as PCL, neither resulted in the production of the expected 3′-*N*-debenzoyl-2′-deoxypaclitaxel (**17**) when expressed with PAM and BAPT with fed-in baccatin III (**16**) in our system. To identify the missing PCL, we examined the top AAEs in the Taxol expression modules (Fig. 5e), which revealed a single prominent candidate: AAE-867.5. Co-expression of AAE-867.5, PAM and BAPT with fed-in baccatin III (**16**) or our baccatin III pathway in planta resulted in a mass corresponding to 3′-*N*-debenzoyl-2′-deoxypaclitaxel (**17**) (Fig. 5f), indicating that AAE-867.5 is the functional PCL. This constitutes the first (to our knowledge) de novo biosynthetic production of the late-stage paclitaxel precursor 3′-*N*-debenzoyl-2′-deoxypaclitaxel (**17**).

Similarly, although T2′αH and DBTNBT have been previously reported[45,47], we were not able to produce Taxol (**18**) when we added them to our pathway to **17**. However, when DBTNBT was expressed with PCL, PAM and DBAT with fed-in baccatin III (**16**) we detected a mass corresponding to 2′-deoxypaclitaxel at very low levels (Supplementary Fig. 19). This result suggests that DBTNBT might be functional but requires upstream 2′α-hydroxylation by T2′αH to react efficiently, consistent with previous specificity measurements[48].

## Expression and essentiality of Taxol genes

Of the eight Taxol genes discovered in this work (Supplementary Tables 1 and 11), few were clear candidates for the pathway when performing co-expression analysis using bulk RNA-seq datasets (Fig. 5g). Correlation with TDS ranked these genes as 1,000th–10,000th priority when using bulk RNA-seq data, but within the top tens to hundreds when using mpXsn data (Fig. 5g), showing that our mpXsn strategy was crucial for efficient gene discovery. Furthermore, the transcriptional modules of these genes provide insights into the organization of the biosynthetic pathway. When ordered by presumed biosynthetic order, the early pathway forms an especially discrete cluster in Taxol module 1 (Fig. 5h), suggesting that the pathway up to 4O3A (**5**) is controlled by separate transcriptional regulation from the later pathway. The latter pathway is not as clustered by order, suggesting that either our presumed biosynthetic order does not reflect the real order, or the pathway is regulated at various post-transcriptional stages.

For our 17-gene reconstituted baccatin III (**16**) pathway, dropping most of the genes completely abolishes the production of baccatin III (**16**), with the exception of FoTO1, T5αH, DBAT and T9ox (Fig. 5i and Supplementary Figs. 20 and 21). Of these four genes, only T5αH can be left out without a substantial loss of baccatin III yield. Moreover, our 17-gene pathway yields levels of baccatin III (**16**) that are considerably higher than those obtained with two previously published gene sets[12,14] (Fig. 5i and Supplementary Figs. 20 and 21), both of which did not produce baccatin III above the limit of detection in our system

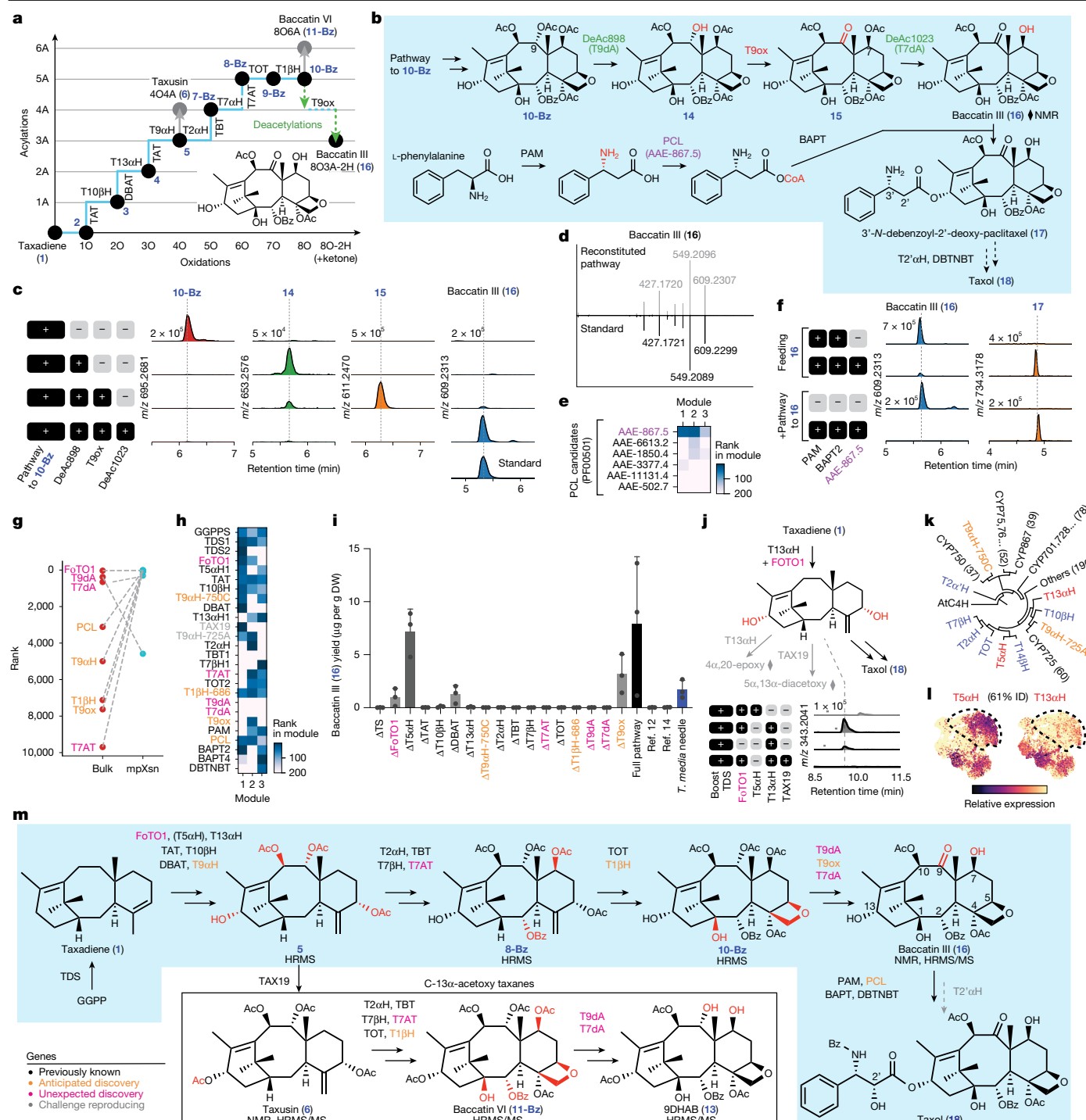

**Fig. 5 | Total biosynthesis of baccatin III (16) and 3′-N-debenzoyl-2′-deoxypaclitaxel (17) in *N. benthamiana*. a**, Simplified representation of biosynthetic transformations (acetylation and oxidation) from taxadiene (**1**) to baccatin III (**16**). **b**, Proposed biosynthetic pathway from **10-Bz** to Taxol (**18**). **c**, EICs of intermediates when the indicated sets of genes were expressed in *N. benthamiana* leaves. **d**, MS/MS of heterologously produced **16** compared with that of **16** standard. **e**, Heat map showing ranks of PCL candidates in Taxol gene modules. **f**, EICs of **16** and **17** from feeding **16** to *N. benthamiana* leaves expressing PAM, PCL and BAPT or from expressing the complete gene set. **g**, Ranks of new Taxol biosynthetic genes discovered in this paper by PCC to TDS using either bulk (*n* = 79 samples) or mpXsn (*n* = 17,143 cells across 3 experiments) data. Orange indicates anticipated discovery; pink indicates unexpected discovery. **h**, Heat map showing ranks of updated Taxol biosynthetic genes in the modules. **i**, Baccatin III (**16**) yields in µg per g dried weight (DW),

quantified with standards (Methods), of our 17-gene pathway with each single-gene dropout tested in *N. benthamiana*. Yields from replicating published gene sets[12,14] and from *T. media* needles are shown for comparison. Data are mean ± s.d., *n* = 3 biological replicates (Supplementary Data 2). **j**, EIC showing the proposed taxadien-5α,13α-diol produced by T13αH and facilitated by FoTO1 (Extended Data Fig. 8). Over-oxidized derivative and TAX19-acetylated product are confirmed by NMR (diamond). **k**, Phylogenetic tree of *Taxus* P450s. See Extended Data Fig. 9. **l**, Single-cell expression of T5αH and T13αH. **m**, Taxol biosynthesis (blue shading) and the reconstituted 13α-acetoxy taxane biosynthesis. Structures assigned on the basis of high-resolution mass spectrometry (HRMS) (HRMS-predicted chemical formula), NMR (NMR analysis followed by purification) and/or HRMS/MS (MS/MS spectra comparison with authentic standards). A detailed biosynthetic pathway is shown in Extended Data Fig. 10.

(Supplementary Fig. 20). The low yield of these enzyme sets is likely to be due to the absence of several biosynthetic genes identified here, the functions of which might be partially compensated for by multi-functional *Taxus* enzymes[15] or endogenous host enzymes in other systems. Finally, exchanging the T9αH-750C discovered in this work for a recently reported[12–14] alternative T9αH-725A also yielded no detectable baccatin III (**16**) in our system (Supplementary Fig. 21 and Extended Data Fig. 7c), as would be anticipated because of T9αH-750C's specificity requirement of a C-13α acetoxy that is absent from Taxol and our pathway (Fig. 4b).

Notably, our dropout experiment revealed that T5αH is not essential in our reconstituted 17-gene baccatin III (**16**) biosynthesis (Fig. 5i and Supplementary Figs. 20 and 21). Despite the widely accepted model that T13αH is the second oxidase after T5αH, we found that T13αH can directly oxidize taxadiene (**1**), producing multiple oxidized products, some of which are the same as those formed by T5αH, including OCT (**2'a**), iso-OCT (**2'b**) and **2'c** (Extended Data Fig. 8a). This prompted us to further investigate whether T13αH can compensate for T5αH. Similar to its effect on the product profile of T5αH, FoTO1 also streamlines T13αH's products, eliminating **2'a–c**, and selectively boosting the formation of taxadien-5α,13α-diol and its derivatives (Fig. 5j and Extended Data Fig. 8a–c). We structurally characterized an over-oxidized derivative, 4α,20-epoxy-5α-hydroxy-taxadien-13-one (Supplementary Note 3 and Supplementary Tables 8 and 9), and a TAX19 derivative, 5α,13α-diacetoxy-taxadiene (Extended Data Fig. 8d and Supplementary Table 10) by NMR, confirming that T13αH can compensate for T5αH by catalysing both 5α- and 13α-hydroxylation (Fig. 5j), consistent with the previously reported multifunctionality of T13αH (converting taxadiene (**1**) to taxadien-5α,10β,13α-triol)[12]. This exemplifies a repeated observation of enzyme functional overlap, including pairs of differentially expressed C-5-hydroxylases (T5αH and T13αH; Fig. 5k,l), TDSs, T9αHs, T1βHs and C-5-*O*-acetyltransferases, during our dissection of the Taxol biosynthetic pathway (Extended Data Fig. 9).

## Discussion

In this investigation, we have identified eight new genes in the Taxol biosynthetic pathway, and used them to build 17-gene and 20-gene pathways for the de novo biosynthesis of baccatin III (**16**) and 3'-*N*-debenzoyl-2'-deoxypaclitaxel (**17**), respectively (Fig. 5m and Extended Data Fig. 10), in *N. benthamiana*. This updates our long-held model of Taxol biosynthesis: two additional acetylations seem to be necessary for the functions of intermediate oxidases, and downstream deacetylations by two deacetylases furnish the baccatin III (**16**) end-product (Extended Data Fig. 10). Future engineering or discovery of the final oxidase, T2'αH, would enable de novo total biosynthesis of Taxol. Two of the new enzymes, T7AT and T9ox, have recently been independently reported by other groups[13,14] (Supplementary Table 1). Without optimization, our reconstituted 17-gene pathway yields 10–30 μg g$^{-1}$ baccatin III (**16**) in *N. benthamiana* leaves, equivalent to its natural abundance in *T. media* needles (Extended Data Fig. 7d). Because baccatin III (**16**) extracted from *Taxus* is the main precursor for industrial semi-synthesis, our work represents a major step towards the sustainable production of Taxol and other taxane-based therapeutics.

Our discovery of FoTO1 and its interaction with T5αH challenges a long-standing assumption that T5αH is the sole actor in the first oxidative steps of the Taxol pathway. Although most P450s require interaction with a cytochrome P450 reductase partner, few are known to associate with scaffold proteins. In *Arabidopsis*, membrane steroid-binding proteins have been identified as the scaffolds for three lignin biosynthetic P450s on the ER membrane[49]. More recently, a cellulose synthase-like protein, GAME15, was discovered to be an ER-localized scaffold for two P450s and one 2-ODD in the biosynthesis of steroids in *Solanaceae*[50,51]. Together with our work, this hints at a broader role for scaffolding proteins in plant secondary metabolism (Supplementary Note 4).

The identities of the new Taxol genes shed light on why they remained unknown for so long. Pathway reconstitution required unanticipated gene families (for example, NTF2-like proteins, 2-ODDs and deacetylases) and functionalizations (for example, acetylation and deacetylation). Moreover, 'red herring' enzymes further complicated pathway dissection, including: (i) enzymes that diverged biosynthesis towards other classes of taxanes (for example, *abeo*-taxanes and 13α-acetoxy taxanes; Extended Data Fig. 10) and (ii) homologues with relevant activity, but with substrate specificities incompatible with our Taxol pathway (for example, T9αHs and T1βHs). These challenges, exacerbated by the sheer quantity of specialized metabolism enzymes in *Taxus*, were overcome by developing a scalable transcriptomic strategy which prioritized candidates with improved specificity (Figs. 1f and 5g). Our mpXsn strategy uses scalable profiling of cell states with differentially perturbed biosynthetic processes, enabling better discrimination between their defining gene sets. In addition to gene discovery, mpXsn data provide biological context, including links with primary metabolism (Fig. 2f) and the partitioning of Taxol enzymes into modules that seem to be separately regulated.

Beyond Taxol biosynthesis, mpXsn will be useful for studying gene sets of interest in other non-model organisms. In mammalian systems, single-cell techniques with parallelized genetic[25] or chemical[26,27] perturbation experiments have enabled the de-orphaning of genes and dissection of gene networks. However, most organisms and biological systems lack genetic interrogation tools, and thus researchers rely heavily on observational experiments, such as transcriptomics and other 'omics'. Eukaryotes, especially, pose major challenges for functional genomics and gene-guided discovery, because they generally lack the comprehensive gene clusters that are found in prokaryotes. The advent of methods such as mpXsn, which affordably capture precise gene covariance across hundreds of transcriptional states, might help to overcome this long-standing challenge in functional genomics.

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

# Methods

## Chemical and biological materials

Chemical standards were purchased from the following vendors (with catalogue number listed): taxusin (TargetMol; TN6763), 1-hydroxy-baccatin I (LKT Labs; T0092), baccatin VI (Santa Cruz Biotechnology; sc-503244), 10-deacetylbaccatin III (Sigma-Aldrich; D3676), baccatin III (MedChemExpress; HY-N6985) and 9-dihydro-13-acetyl-baccatin III (TargetMol; T5132). Taxadien-5α-ol was synthesized as previously described[18]. *Taxus media* var. *hicksii* was obtained from FastGrowingTrees.

## Tissue preparation and single-nucleus sequencing

The cells of *Taxus* species, like those of many plants, are often two to three times larger than the 35-μm diameter limit for the standard 10x Genomics Chromium single-cell library devices. Consequently, single-cell isolation approaches (such as protoplasting) risked introducing a severe cell-type bias, and we instead adapted previously described nuclei isolation methods[52] into a conifer-compatible snRNA-seq protocol. *Taxus media* var. *hicksii* aerial tissues (needles, stems and bud scales) were manually disrupted by razor blade and detergent treatment, followed by DNA staining, fluorescence-activated cell sorting (FACS) purification and library synthesis in the 10x Chromium platform. Nuclei extraction buffer (NIB) consisted of 5 mM $MgCl_2$, 10 mM HEPES pH 7.6, 0.8 M sucrose, 0.1% Triton X-100 and (for density matching to prevent nuclei settling during flow sorting) 1% dextran T40 and 2% Ficoll. On the day of use, NIB was supplemented with 1 mM dithiothreitol. All nuclei-extraction steps were performed at 4 °C and wide-bore pipette tips were used when handling nuclei. Steps between tissue collection and loading into the Chromium device were completed within 90 min to avoid RNA loss. To isolate nuclei, approximately 1 g of *T. media* tissue was removed from the plant and immediately placed in a Petri dish with 10 ml NIB. Tissue was chopped by hand at around 200 rpm with a fresh razor blade for 5 min until most of the large tissue was broken down, and was then gently rocked at 4 °C for 15 min. To remove large debris, disrupted tissue was then passed through a pre-wet 100-μm cell strainer stacked on top of a 40-μm cell strainer. Nuclei were gently pelleted at 300$g$ at 4 °C for 5 min and resuspended in 1 ml NIB with 5 ng μl$^{-1}$ 4,6-diamidino-2-phenylindole (DAPI, Thermo Fisher Scientific) and 5 ng μl$^{-1}$ propidium iodide. Using a Sony SH800 cell sorter with a 70-μm chip, 140,000–200,000 nuclei were sorted into a tube containing 1 ml PBS+ (PBS, 0.1% bovine serum albumin and 20 U ml$^{-1}$ Invitrogen ribonuclease inhibitor). The gating strategy is shown in Supplementary Fig. 22. Nuclei were centrifuged at 300$g$ at 4 °C for 5 min, then gently resuspended in 40 μl PBS+. Nuclei were immediately loaded onto a 10x Genomics Chromium controller and libraries were generated using v3 chemistry. Libraries were sequenced on an Illumina NextSeq 3000.

## Multiplexed tissue elicitation

For the multiplexed elicitation experiment, *Taxus* needles were subjected to perturbation in deep-well 96-well plates with 200 μl MS medium (7.5 g l$^{-1}$ Murashige and Skoog macronutrients (Fisher), 3 g l$^{-1}$ sucrose, pH 5.7) supplemented with elicitor. Two needles (biological replicates), each from two developmental stages (young and mature), were treated with each elicitation condition (17 conditions listed in Supplementary Table 2) for each time point (1, 2, 3 and 4 days), resulting in 272 tissue samples (2 replicates of 136 perturbations). To minimize contamination, needles were washed thoroughly in sterile water before moving to MS plates, which were sealed with breathable rayon film (VWR) and placed under 18-h light cycles. Tissue elicitation was started at staggered times so that all tissues could be collected simultaneously. To extract nuclei from elicited tissues, all tissues were combined in a wire mesh, washed with water and subjected to the above nuclei-extraction protocol.

## Analysis of single-cell data

Reads were cleaned with Trimmomatic[53] and mapped to the genomes of *T. chinensis*[5] with STARsolo (v.2.7.10b)[54] (STAR…–runThreadN 32–alignIntronMax 10000–soloUMIlen 12–soloCellFilter EmptyDrops_CR–soloFeatures GeneFull–soloMultiMappers EM–soloType CB_UMI_Simple). Ambient RNA was removed with CellBender (v.0.3.0)[55]. Using the doubletdetection (v.4.2) library[56], doublets were removed, as well as cells with outlier numbers of reads or in which most reads were the most expressed genes (pct_counts_in_top_20_genes < 25). Genes were removed from analysis if expressed in fewer than 50 cells. For integrated UMAP plots, scVI was used to integrate cells from multiple single-cell experiments[57]. Scanpy (v.1.10.1)[58] was used for processing and plotting post-filtered nuclear transcriptomes. For co-expression analysis and gene–gene correlation calculations, scVI-normalized[57] transcriptomes (8,039 elicited transcriptomes, 3,027 naive transcriptomes from young tissues and 6,077 naive transcriptomes from mature tissues) were clustered into 2,901 cell states (around 10 cells per state) by Leiden clustering[58], and then raw reads from each cluster were pooled to yield pseudobulk transcriptomes. These pseudobulk transcriptomes were used to calculate gene–gene correlations. For module analysis, raw reads were analysed by a cNMF package[28] run with default parameters, except 'total modules') to yield gene modules and their usage across cells. Factorization approximates the observed dataset as the product of two smaller, meaningful matrices: (i) a gene–module matrix (a weight value for each gene in each module); and (ii) a cell–module matrix (expression values of each module in each cell) (Fig. 2c). The weight values of the gene–module matrix can be used as scores that identify the genes that dominate each module; top-scoring genes from the same module have coordinated expression patterns and are likely to be part of the same molecular processes. This approach adapts to the rich but noisy data inherent in single-cell analysis, and reveals patterns of coordinated gene expression that might not be apparent from linear correlation analysis. For example, it allows for genes to be in multiple, overlapping modules, which is likely to better represent how genes in a highly branched metabolism may be expressed. The 'total modules' parameter was scanned from $k = 50$ to $k = 400$ to determine the sensitivity of the results on this parameter (Supplementary Fig. 1).

## Bulk RNA-seq analysis

Raw fastq files from six previous studies[5,59–63] were downloaded from NCBI (PRJNA493167, PRJNA251671, PRJNA733140, PRJNA427840, PRJNA497542, PRJNA499080 and PRJNA864083), cleaned with Trimmomatic[53] and aligned to the *T. chinensis* genome[5] (STAR map[64]). Gene–gene correlation was calculated with numpy. Mutual rank (mr), used to calculate the gene linkage maps (Fig. 1d), is defined as:

$$mr_{ij} = \sqrt{rank_{ij} \times rank_{ji}},$$

where rank$_{ij}$ indicates the Pearson correlation rank of gene $i$ to gene $j$.

## Cloning of *Taxus* genes

The cloning of cytosolic diterpenoid boost genes (tHMGR and GGPPS), cytosolic TDS1 and TDS2, T5αH, TAT, T10βH, DBAT, T13αH and TAX19 genes has been described previously[18,65]. Candidate genes were amplified from *T. media* gDNA or cDNA (generated with SuperScript IV, Thermo Fisher Scientific) by PCR (PrimeStar, Takara Bio R045B, primers in Supplementary Table 13), and the PCR products were ligated with AgeI- and XhoI- (New England Biolabs) linearized pEAQ-HT vector[66] using HiFi DNA assembly mix (New England Biolabs). Gene annotations used for cloning were taken from the *T. chinensis* genome[5] by default, but were BLAST-searched against the *T. media* genome (NCBI PRJNA1136025) to determine whether alternative gene models were available. Constructs were transformed into 10-beta competent *E. coli* cells (New England Biolabs). Plasmid DNA was isolated using the

QIAprep Spin Miniprep kit (QIAGEN) and the sequence was verified by whole-plasmid sequencing (Plasmidsaurus).

## Transient expression of *Taxus* genes in *N. benthamiana* by *Agrobacterium*-mediated infiltration

pEAQ-HT plasmids containing the *Taxus* gene were transformed into *Agrobacterium tumefaciens* (strain GV3101) cells using the freeze–thaw method. Transformed cells were grown on bacteria screening medium 523-agar (Phytotech Labs) plates containing kanamycin and gentamicin (50 μg ml$^{-1}$ and 30 μg ml$^{-1}$, respectively; same for the 523 medium below), at 30 °C for two days. Single colonies were then picked and grown overnight at 30 °C in 523-kanamycin–gentamicin liquid medium. The overnight cultures were used to make dimethyl sulfoxide (DMSO) stocks (7% DMSO) for long-term storage in the −80 °C fridge. For routine *N. benthamiana* infiltration experiments, individual *Agrobacterium* DMSO stocks were streaked out on 523-agar containing kanamycin and gentamicin and grown for around one to two days at 30 °C. Patches of cells were scraped off from individual plates using 10-μl inoculation loops and resuspended in around 1–2 ml of *Agrobacterium* induction buffer (10 mM MES pH 5.6, 10 mM MgCl$_2$ and 150 μM acetosyringone; Acros Organics) in individual 2-ml safe-lock tubes (Eppendorf). The suspensions were briefly vortexed to homogeneity and incubated at room temperature for 2 h. The optical density at 600 nm (OD$_{600 nm}$) of the individual *Agrobacterium* suspensions was measured, and the final infiltration solution, in which the OD$_{600 nm}$ was 0.2 for each strain (except for TDS, T7AT and T7dA; OD$_{600 nm}$ of 0.6, 0.4 and 0.1, respectively), was prepared by mixing individual strains and diluting with the induction buffer. Leaves of four-week-old *N. benthamiana* were infiltrated using needleless 1-ml syringes from the abaxial side. Each experiment was tested on leaf 6, 7 and 8 (numbered by counting from the bottom) of the same *N. benthamiana* plant, as three biological replicates.

For the reconstitution of pathways that involve TBT, the following modifications were made to the procedure above to increase the production of the desired benzoylated products: *N. benthamiana* plants were watered with 2 mM benzoic acid in water (buffered to pH 5.6) a day before *Agrobacterium* infiltration, 1 mM benzoic acid was added to the induction buffer and the pH was adjusted to 5.6 before being used for the resuspension of *Agrobacterium* and preparation of the final infiltration solution.

## Phylogenomic analysis

FoTO1 homologues were identified by scanning the Thousand Plant Transcriptome (1KP)[67], RefSeq plants and Uniprot *Viridiplantae* databases with jackhmmer[36] (command: jackhmmer -o tempout.txt -E 1e-5 -N 4). Hits with greater than 40% sequence gaps to the original query were discarded. A phylogenetic tree was generated with the remaining protein sequences with FastTree[68].

## Metabolite extraction of *N. benthamiana* leaves

Five days after *Agrobacterium* infiltration, *N. benthamiana* leaf tissue was collected using a leaf disc cutter 1 cm in diameter and placed inside a 2-ml safe-lock tube (Eppendorf). Each biological replicate consisted of four leaf discs from the same leaf (approximately 40 mg fresh weight). The leaf discs were flash-frozen and lyophilized overnight. Analyses of the more hydrophobic metabolites (for example, compounds **1**–**6**) were done by GC–MS, and analyses of the more hydrophilic metabolites (for example, compounds **4**–**18**) were done by liquid chromatography–mass spectrometry (LC–MS). To extract metabolites, ethyl acetate (ACS reagent grade; J.T. Baker) or 75% acetonitrile (high-performance liquid chromatography (HPLC) grade; Fisher Chemical) in 500 μl water was added to each sample along with one 5-mm stainless steel bead for GC–MS or LC–MS analysis, respectively. The samples were homogenized in a ball mill (Retsch MM 400) at 25 Hz for 2 min. After homogenization, the samples were centrifuged at 18,200 g for 10 min. For GC–MS samples, the supernatants were transferred to 50-μl glass

inserts, placed in 2 ml vials and subjected to analysed by the GC–MS instrument. For LC–MS samples, the supernatants were filtered using 96-well hydrophilic PTFE filters with a pore size of 0.45 μm (Millipore) and analysed by the LC–MS instrument.

## GC–MS analysis

GC–MS samples were analysed using an Agilent 7820A gas chromatography system coupled to an Agilent 5977B single quadrupole mass spectrometer. Data were collected with Agilent Enhanced MassHunter and analysed by MassHunter Qualitative Analysis B.07.00. Separation was done using an Agilent VF-5HT column (30 m × 0.25 mm × 0.1 μm) with a constant flow rate of helium of 1 ml per min. The inlet was set at 280 °C in split mode with a 10:1 split ratio. The injection volume was 1 μl. Oven conditions were as follows: start and hold at 130 °C for 2 min, ramp to 250 °C at 8 °C per min, ramp to 310 °C at 10 °C per min and hold at 310 °C for 5 min. The post-run condition was set to 320 °C for 3 min. MS data were collected with a mass range 50–550 *m/z* and a scan speed of 1,562 u s$^{-1}$ after a 4-min solvent delay. The MSD transfer line was set to 250 °C, the MS source was set to 230 °C and the MS Quad was set to 150 °C.

## LC–MS analysis

LC–MS samples were analysed on either or both of our two instruments: (1) an Agilent 1260 HPLC system coupled to an Agilent 6520 Q-TOF mass spectrometer or (2) an Agilent 1290 HPLC system coupled to an Agilent 6546 Q-TOF mass spectrometer. Typically, the 6520 system shows better sensitivity for the more hydrophobic metabolites, such as **4**–**6**, whereas the 6546 system works better for the more hydrophilic, highly modified taxanes. Data were collected with Agilent MassHunter Workstation Data Acquisition and analysed by MassHunter Qualitative Analysis 10.0. Separation was done using a Gemini 5-μm NX-C18 110-Å column (2 × 100 mm; Phenomenex) with a mixture of 0.1% formic acid in water (A) and 0.1% formic acid in acetonitrile (B) at a constant flow rate of 400 μl per min at room temperature. The injection volume was 2 μl or 1 μl for the 6520 or the 6546 system, respectively. The following gradient of solvent B was used: 3% 0–1 min, 3%–50% 1–2 min, 50%–97% 2–12 min, 97% 12–14 min, 97%–3% 14–14.5 min and 3% 14.5–21 min (6520 system) and 3% 0–1 min, 3%–50% 1–5 min, 50%–97% 5–10 min, 97% 10–12 min, 97%–3% 12–12.5 min and 3% 12.5–15 min (6546 system). MS data were collected using electrospray ionization (ESI) in positive mode with a mass range of 50–1,200 *m/z* and a rate of one spectrum per second (6520 system), or Dual AJS ESI in positive mode with a mass range of 100–1,700 *m/z* and a rate of one spectrum per second (6546 system). The ionization source was set as follows: 325 °C gas temperature, 10 l min$^{-1}$ drying gas, 35 psi nebulizer, 3,500 V VCap, 150 V fragmentor, 65 V skimmer and 750 V octupole 1 RF Vpp (6520 system), or 325 °C gas temperature, 10 l min$^{-1}$ drying gas, 20 psi nebulizer, 3,500 V VCap, 150 V fragmentor, 65 V skimmer and 750 V octupole 1 RF Vpp (6546 system). MS/MS fragmentations were generated using [M+Na]$^+$ as the precursor ion and fragmented with a collision energy of 30 eV unless otherwise stated.

## Quantification of baccatin III (16)

The samples in Fig. 5i were analysed by an Agilent 1290 HPLC system coupled to an Agilent 6470 triple quadrupole (QQQ) mass spectrometer to accurately quantify the concentration of baccatin III. Data were collected with Agilent MassHunter Workstation Data Acquisition and analysed by MassHunter Quantitative Analysis 10.1 and Microsoft Excel. Separation was done using a ZORBAX RRHD Eclipse Plus C18 Column (2.1 × 50 mm, 1.8 μm; Agilent) with a mixture of 0.1% formic acid in water (A) and 0.1% formic acid in acetonitrile (B) at a constant flow rate of 600 μl per min at 30 °C. The injection volume was 0.5 μl. The following gradient of solvent B was used: 30% 0–1 min, 30%–100% 1–5 min, 100% 5–6.5 min, 100%–30% 6.5–7 min and 30% 7–8 min. MS data were collected using AJS ESI in positive mode. Multiple reaction

monitoring was used to monitor the 609.2 to 549.2 ion transition at a collision energy of 24 eV as the quantifier, and the 609.2 to 427.1 ion transition at a collision energy of 32 eV as the qualifier. The ionization source was set as follows: 250 °C gas temperature, 12 l min⁻¹ drying gas, 25 psi nebulizer, 300 °C sheath gas temperature, 12 l min⁻¹ sheath gas flow, 3,500 V VCap, 0 V nozzle voltage.

### Extraction and purification of taxanes from *N. benthamiana*
*Nicotiana benthamiana* plants were infiltrated with the combinations of biosynthetic genes shown in Supplementary Table 12 for the purification of taxusin (**6**), taxusin (**6′**), 1β-hydroxytaxusin (**6-O1**) and 15-hydroxy-11(15→1)*abeo*-taxusin (**6-O2**). Lyophilized *N. benthamiana* materials were cut into small pieces and extracted with 1 l ethyl acetate (ACS reagent grade; J.T. Baker) in a 2-l flask for 48 h at room temperature with constant stirring. Extracts were filtered using vacuum filtration and dried using rotary evaporation. Two rounds of chromatography were used to isolate compounds of interest. The chromatography conditions for each compound are summarized in Supplementary Table 12. In brief, the first chromatography was performed using a 7-cm-diameter column loaded with P60 silica gel (SiliCycle) and using hexane (HPLC grade; VWR) and ethyl acetate as the mobile phases. The second chromatography was performed on an automated Biotage Selekt system with a Biotage Sfär C18 Duo 6-g column using Milli-Q water and acetonitrile as the mobile phases. Fractions were analysed by LC–MS to identify those containing the compound of interest. Desired fractions were pooled and dried using rotary evaporation (first round) or lyophilization (second round). Purified products were analysed by NMR.

### NMR analysis of purified compound
CDCl₃ (Acros Organics) was used as the solvent for all NMR samples. ¹H, ¹³C and 2D-NMR spectra were acquired on a Varian Inova 600-MHz or a Bruker NEO 500-MHz spectrometer at room temperature using VNMRJ 4.2, and the data were processed and visualized on MestReNova v.14.3.1-31739. Chemical shifts were reported in ppm downfield from Me₄Si by using the residual solvent (CDCl₃) peak as an internal standard (7.26 ppm for the ¹H and 77.16 ppm for the ¹³C chemical shift). Spectra were analysed and processed using MestReNova v.14.3.1-31739.

### Taxane feeding experiments
*Taxus* genes were expressed in *N. benthamiana* leaves using the *Agrobacterium*-mediated infiltration method described above. Three days after *Agrobacterium* infiltration, taxanes (purified 3O2A (**4**), taxusin (**6**), 10-deacetylbaccatin III or 9-dihydro-13-acetylbaccatin III (**13**); unless otherwise specified, a 100-μM solution after diluting with 10 mM DMSO stock was used) were fed into the leaves. Approximately 150 μl of solution was used per leaf to yield a circle with a diameter around 3 cm, which was marked for reference. After 18–24 h, four leaf discs were collected within the marked area with a 1-cm diameter cutter, and LC–MS samples were prepared following the methods described above.

### Construction of phylogenetic trees
Sequences from the *T. chinensis* genome were selected using Pfam to identify 672 P450s (PF00067), 218 2-ODDs (PF03171) and 195 acyltransferases (PF02458). P450s were further filtered to those longer than 300 amino acids (467 P450s). Multiple sequence alignment for each family was performed using Clustal Omega, and the phylogenetic trees were constructed using the neighbour-joining method in Geneious Prime (v.2024.0.4) with 100 bootstrap replicates for initial analysis. *Arabidopsis thaliana* cinnamate 4-hydroxylase (*At*C4H, accession NP_180607.1), *A. thaliana* gibberellin 20-oxidase1 (*At*GA20ox1, accession NP_194272.1), and *Hordeum vulgare* agmatine coumaroyltransferase (*Hv*ACT, accession AAO73071.1) were used as outgroups for the P450, 2-ODD and acyltransferase families, respectively. All analyses were performed with default settings unless otherwise specified.

Representative genes from major clades of the initial analyses and the Taxol biosynthetic genes were then selected to construct the final phylogenetic trees (Extended Data Fig. 9) using the neighbour-joining method with 1,000 bootstrap replicates.

### Purification of proteins and binding assays
All proteins were purified from standard pET28a vectors expressed in BL21DE3 cells (New England Biolabs, C2527H). FoTO1 and FoTO1(ΔCterm) were purified as C-terminal fusions: His6-3×Flag-TEV-mTurq2-GSG-FoTO1. T5αH and TDS were purified with N-terminal purification tags (His6-3×Flag-TEV-enzyme) with N-terminal signal peptides removed (T5αH, 47 amino acids removed; TDS2, 60 amino acids removed). Proteins were purified as previously described[69], with post-lysis steps done at 4 °C. In brief, 1 l of cells were grown to an OD₆₀₀ ₙₘ of 0.4–0.5, induced with 0.3 mM IPTG and expressed for 16 h overnight at 18 °C. Cell pellets were lysed in lysis buffer (0.5 M NaCl, 20 mM HEPES pH 8.0, 0.1% Triton X-100, 1 mg ml⁻¹ lysozyme, HALT protease cocktail (Thermo Fisher Scientific) and 1 μl ml⁻¹ DNASE I (New England Biolabs)) by sonication, clarified by centrifugation for 1 h at 8,000*g*. Proteins were purified on pre-equilibrated Ni-NTA beads (New England Biolabs) and exchanged into a protein storage buffer (10 mM HEPES-KOH pH 8.0, 50 mM KCL, 10% glycerol, 1 mM DTT and 1 mM EDTA). Purified proteins were quantified by Bradford assay, and SDS–PAGE gels were used to verify protein size and correct protein concentration.

For each multiscale thermophoresis experiment, one protein was first labelled with the NanoTemper His-Tag labelling kit (RED-tris-NTA v2, MO-L018) for 30 min at room temperature according to reagent protocols. MST experiments were performed in PBS with 0.05% Tween-20 with labelled query protein (T5αH- or TDS-labelled) at 100 nM and a titration series of target protein.

### Co-IP
*Nicotiana benthamiana* leaves were harvested four days after infiltration. Leaf tissue was homogenized in liquid nitrogen and resuspended in extraction buffer (50 mM Tris pH 7.5, 150 mM NaCl, 0.6% NP-40, 0.6% CHAPS and 1 mM β-mercaptoethanol)[70]. Lysates were kept on ice and centrifuged at 20,000*g* for 10 min at 4 °C. The protein content of the clarified extract was determined by Bradford assay (Abcam, 119216). Ten microlitres of protein-G-coated magnetic beads (Invitrogen, 10003d) were washed twice in binding buffer (50 mM Na₂HPO₄, 25 mM citric acid, pH 5.0) before a 1-h incubation at room temperature under agitation with 1 μl anti-V5 antibody. Lysates were incubated with the indicated compounds for 15 min under agitation. Antibody-bound beads were then washed twice in extraction buffer and incubated for 15 min under agitation at room temperature with lysate corresponding to 100 μg of total protein content (approximately 40 μl). After incubation, bead complexes were washed three times in extraction buffer and mixed with LDS sample buffer (Invitrogen, NP0007) for subsequent analysis by immunoblotting.

### Immunoblotting
Lysates were separated for 1.5 h at 80 V on a NuPAGE gel (Invitrogen, NP0321) before transfer onto a PVDF membrane using a Bio-Rad Trans-Blot Turbo Transfer System (Bio-Rad, 1704150). Immunoblots were incubated with the indicated antibodies (anti-V5 at 1:1,000 and anti-HA-HRP at 1:2,500) for 3 h at room temperature under agitation. Blots were subsequently washed and incubated with HRP–protein G (Genscript, M00090) for 1 h, then imaged on the iBright FL1500 Imaging System (Invitrogen, a44241). The extraction buffer was adapted from a previously published procedure[70].

### Reporting summary
Further information on research design is available in the Nature Portfolio Reporting Summary linked to this article.

## Data availability

Raw and processed single-nucleus transcriptome data have been deposited at the NCBI Gene Expression Omnibus (GEO) (accession GSE292840). The bulk RNA-seq data from six previous studies for comparison were downloaded from NCBI (accession PRJNA493167, PRJNA251671, PRJNA733140, PRJNA427840, PRJNA497542, PRJNA499080 and PRJNA864083). The Python code used for transcriptomic and metabolomic analysis is available at GitHub (https://github.com/mcclune/nature2025). The raw NMR free induction decay data of individual compounds have been deposited in the Natural Products Magnetic Resonance Database (https://np-mrd.org/) with the following IDs: 4α,20-epoxy-taxadien-5α-ol (**2′d**) (NP0350670); 4α,20-epoxy-5α-hydroxy-taxadien-13-one (NP0350671); 5α,13α-diacetoxy-taxadiene (NP0350849); taxusin (**6**) (NP0074778); 13β-taxusin (**6′**) (NP0341907); 1β-hydroxytaxusin (**6-O1**) (NP0341908); and 15-hydroxy-11(15→1)*abeo*-taxusin (**6-O2**) (NP0341909). The processed NMR data are shown in Supplementary Figs. 23–65 and Supplementary Tables 3–10.

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

**Acknowledgements** We thank all members of the E.S.S. laboratory (2016–2024) for constructive feedback on the project; C. Liou for help with QQQ data acquisition; P. Almhjell for comments on the manuscript; J. Bohlmann, J. Keasling, P. Zerbe and A. Osbourn for discussions; and HudsonAlpha and the Joint Genome Institute for their work on the *Taxus media* cv. *Hicksii* genome (PRJNA651763), which we used to identify complete versions of some genes. The assembly of the *T. media* genome (proposal: 10.46936/10.25585/60001097 led by J. Bohlmann) was performed by the US Department of Energy (DoE) Joint Genome Institute (https://ror.org/04xm1d337), a DoE Office of Science User Facility, and is supported by the Office of Science of the US DoE operated under contract number DE-AC02-05CH11231. We acknowledge the Stanford Genomics Core, the Macromolecular Structure Knowledge Center at ChEM-H and the Stanford Chemistry NMR facility for use of their instruments. This work is supported by NIH R01 AT010593 (E.S.S.), NIH K99 1K99AT012787 (C.J.M.) and the Damon Runyon Cancer Research Foundation (DRG-2421-21; C.J.M.).

**Author contributions** C.J.M., P.M.F. and E.S.S. conceived the mpXsn approach that was used to identify pathway candidates. C.J.M. developed and performed single-cell procedures, performed transcriptome analysis, selected and cloned gene candidates and performed initial activity screens. C.J.M. and J.C.-T.L. characterized *Taxus* enzymes and analysed the Taxol pathway products. J.C.-T.L. isolated and structurally characterized taxane intermediates by NMR and MS/MS, and performed enzyme phylogenetic analysis. C.W. and C.J.M. performed experiments to characterize the role of FoTO1. B.M.L. shared *Taxus* resources. R.D.L.P. performed preliminary experiments to establish procedures to characterize *Taxus* enzymes, which B.M.L. helped to analyse. E.S.S. and P.M.F. helped to analyse the data. C.J.M., J.C.-T.L. and E.S.S. wrote the manuscript.

**Competing interests** Stanford University has filed a provisional patent (SR24-220) on work from this manuscript, on which the authors are inventors.

**Additional information**
**Correspondence and requests for materials** should be addressed to Elizabeth S. Sattely.

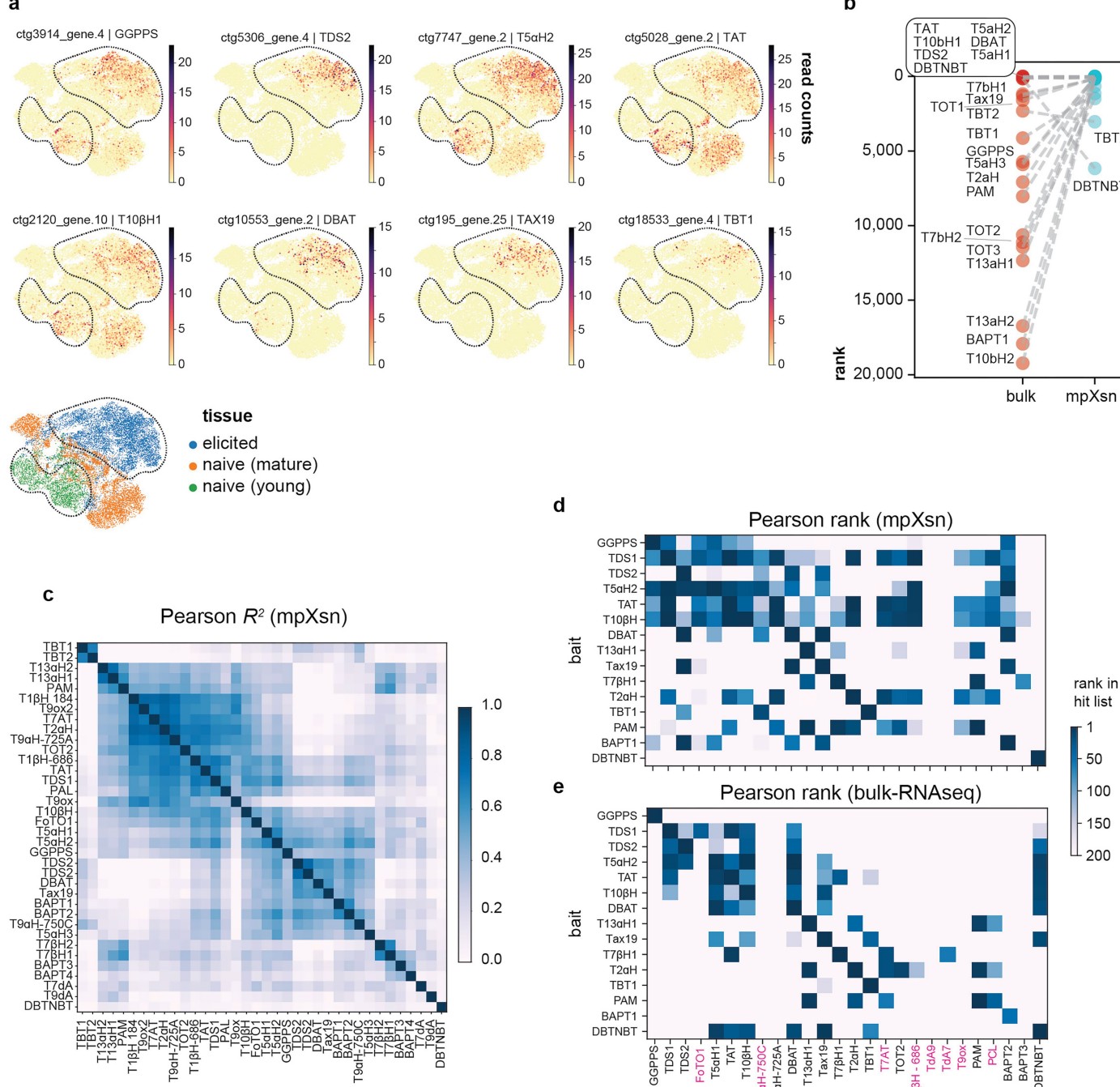

**Extended Data Fig. 1 | Expression and co-expression of Taxol genes.**
**a**, UMAP[52] of Taxol biosynthetic enzyme expression across single cells from naive *T. media* tissues and elicited *T. media* tissues. Three separate single-cell transcriptomic experiments were integrated (cellbender[55], scvi[57], including mature naive aerial tissues (orange), new needle and bud growth (green), and needles from the multiplexed perturbation (blue). Dotted black line encircles the elicited (top right) and young tissue (bottom left) cells in each plot.
**b**, Annotated correlation rank plot of known Taxol biosynthetic genes (with bulk RNA-seq vs mpXsn). Genes in the *T. chinensis* genome are ranked by their

Pearson correlation to TDS1 using either bulk RNA-seq ($n = 79$ samples[5,59–63]) or mpXsn ($n = 17,143$ cells across 3 experiments) data. The ranks of known Taxol biosynthetic enzymes and GGPPS are indicated in this plot. **c**, Pearson correlations between all Taxol genes in this study, using the mpXsn data. Genes are hierarchically ranked on both axes. **d**,**e**, Pearson correlation rank between Taxol enzymes. Using either mpXsn (**d**) or bulk RNA-seq (**e**) data, Pearson correlation ranks were calculated to all genes in this study ($x$ axis) using the previously known Taxol biosynthetic enzymes as bait ($y$ axis).

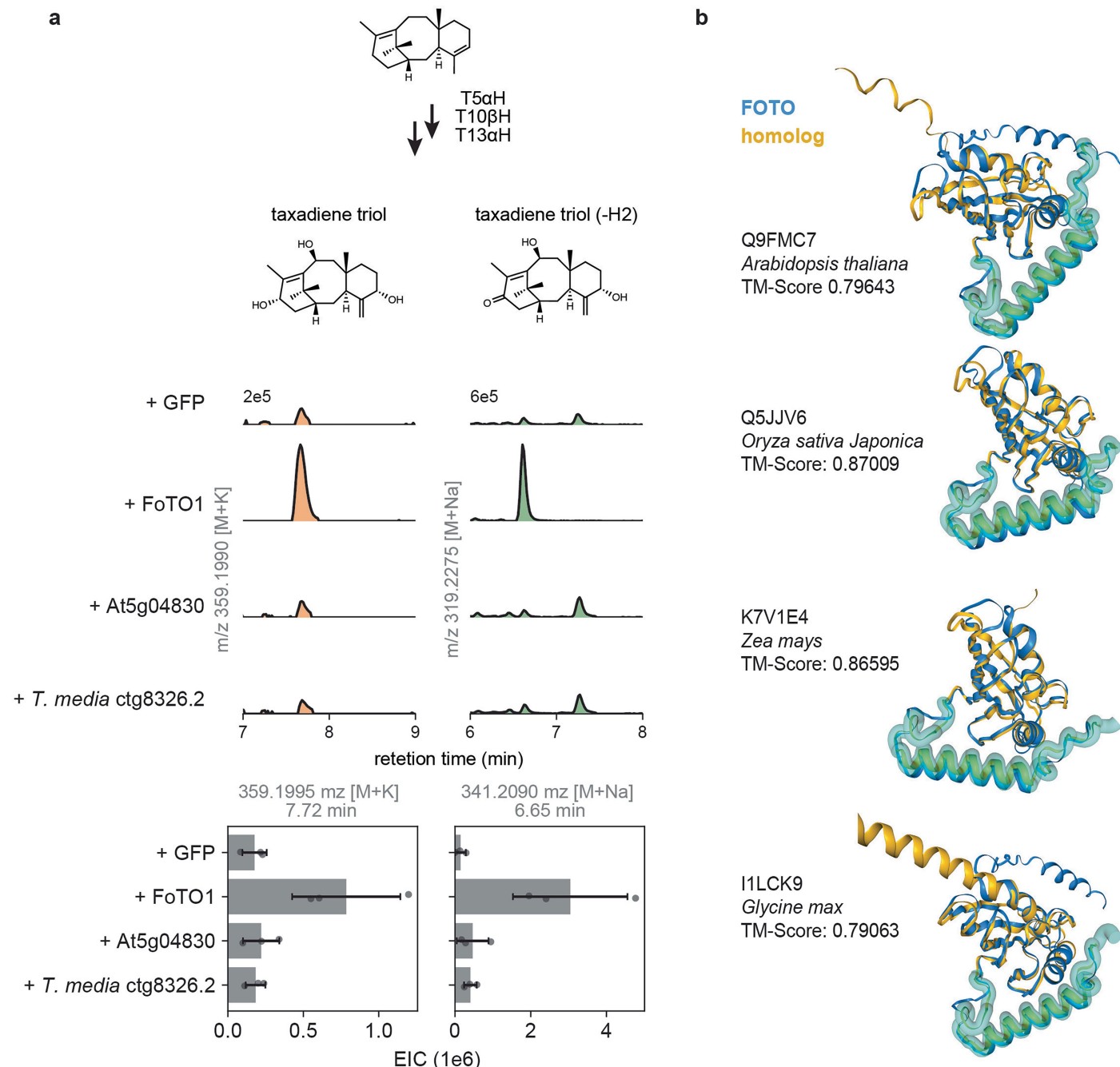

**Extended Data Fig. 2 | Functional testing and structural analysis of FoTO1 homologues. a**, FoTO1 homologues from *T. media* and *A. thaliana* have no effect on taxane yield increase. TDS2 and the first three pathway oxidases were co-expressed in *N. benthamiana* with either GFP, FoTO1, At5g04830 (*A. thaliana* homologue), or ctg8326.3 (*T. media* homologue). Masses corresponding to the expected products, 5α,10β,13α-triol and 5α,10β-diol 13-one, are displayed as EICs and bar graph quantified across three independent replicate leaves. Unlike FoTO1, neither FoTO1 homologue alters the product profile of this early subpathway. Data are shown as the mean ± standard deviation, *n* = 3. **b**, FoTO1 structural homologues identified in other land plants. Using FoldSeek v4[71] we searched all pre-folded protein databases in FoldSeek for full-length structural homologues of FoTO1. The top five hits were proteins from the genomes of model plants *Oryza sativa* Japonica (rice), *Zea mays* (corn), *A. thaliana* and *Glycine max* (soy). After the NTF2 domain, each of these homologues contain the α-helical C terminus (highlighted in transparent turquoise) that we found to be crucial for FoTO1's phenotype in vivo and for binding to TDS and T5αH. The identification of these structural homologues indicates that FoTO1 is not just restricted to gymnosperms, but has structural analogues across both angiosperms.

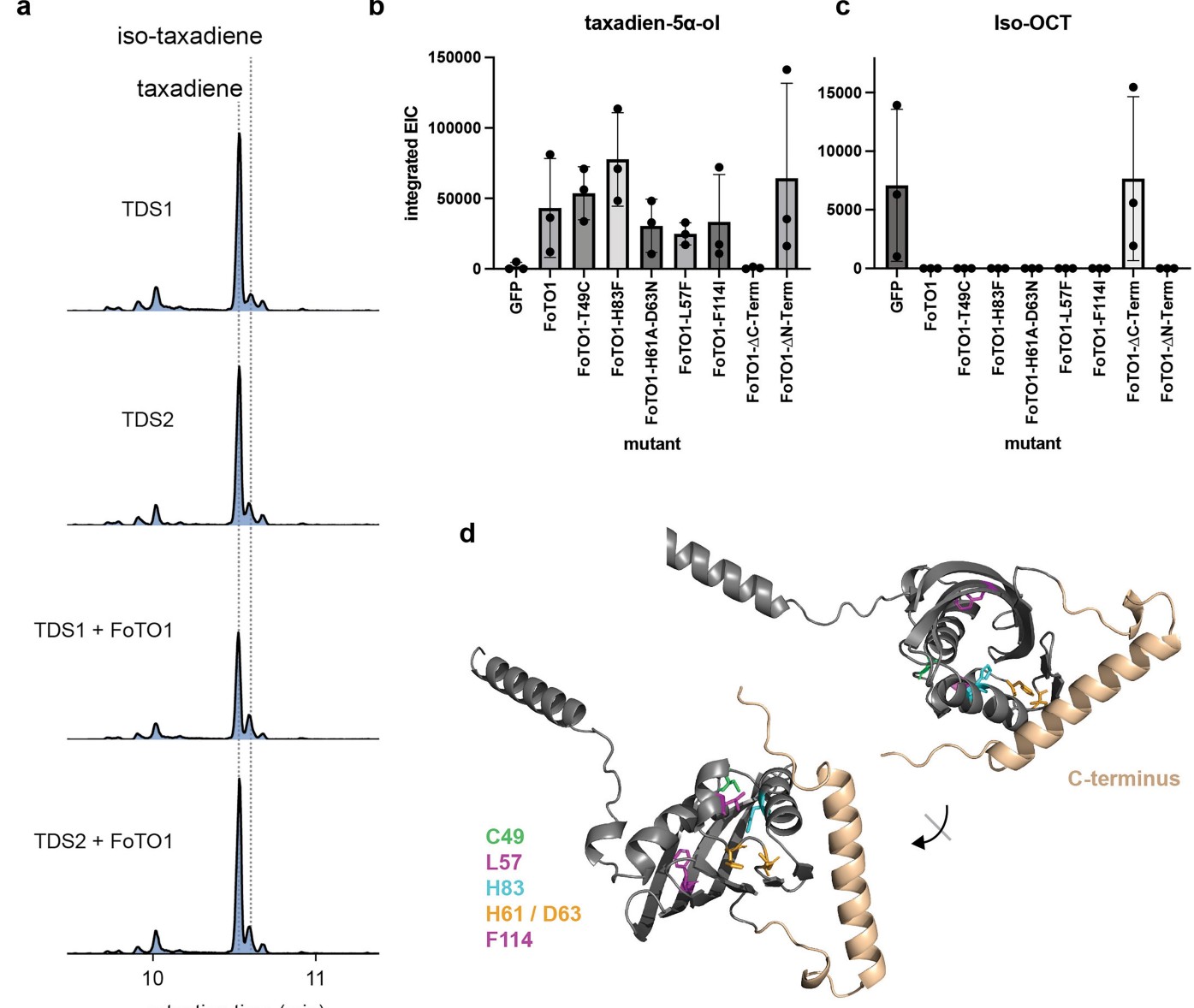

**Extended Data Fig. 3 | Additional functional characterization of FoTO1.**
**a**, FoTO1 does not affect the production of taxadiene by TDS. GC–MS total ion chromatogram (TIC) traces of *N. benthamiana* expressing boost (tHMGR, GGPPS) and TDS (TDS1 or TDS2) with and without FoTO1. When FoTO1 is co-expressed, the formations of taxadiene [**1**, taxa-4(5),11(12)-diene] or iso-taxadiene [taxa-4(20),11(12)-diene] remain the same and there are no observed new peaks. Representative traces of three biological replicates for

each condition are shown. **b,c**, Relative yields of taxadien-5α-ol (**b**; **2**, desired product) and iso-OCT (**c**; **2′b**, an undesired product) measured by GC–MS when TDS2 and T5αH were co-expressed with various FoTO1 mutants in *N. benthamiana*. Candidate residues for catalytic or substrate interaction in the cavity of the NTF2-like fold and regions of truncation were selected based on the AlphaFold3 structure. Data are shown as the mean ± standard deviation, *n* = 3. **d**, FoTO1 structure predicted by AlphaFold3[72].

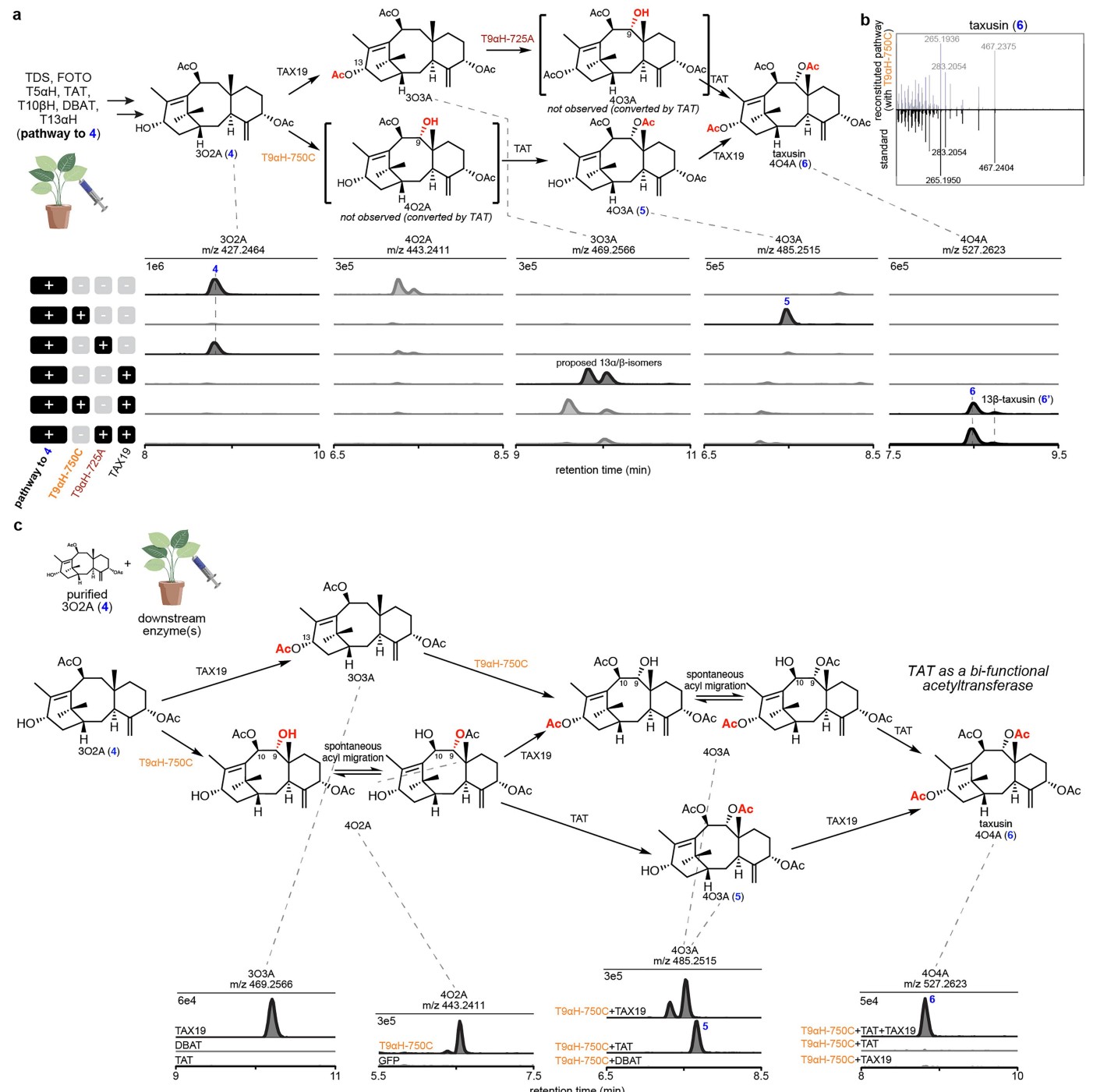

**Extended Data Fig. 4** | See next page for caption.

**Extended Data Fig. 4 | Metabolic pathways to taxusin (6): two convergently evolved T9αHs and the bi-functional TAT. a**, Combinations of genes, indicated on the left, are expressed in *N. benthamiana* via *Agrobacterium*-mediated infiltration, and the EICs of the corresponding intermediates are shown. Expression of T9αH-725A with the 3O2A (**2**) pathway did not yield any new product, while expression of T9αH-750C resulted in a 4O3A (**5**). 4O2A was proposed to be the intermediate; however, at the presence of TAT (within the early 3O2A pathway), it was quickly turned over to 4O3A (**5**) and was not detectable (see **c** for investigation of this transformation). TAX19 produced two major products which we proposed to be the C-13α/β isomers that eventually resulted in taxusin (**6**) and 13β-taxusin (**6'**) with the addition of either T9αH. As T9αH-725A did not affect 3O2A (**4**) accumulation but was able to produce taxusin (**6**) when TAX19 is added, it is only possible that T9αH-725A acts after TAX19, as shown on the upper route. The proposed 4O3A intermediate was is likely to have been quickly turned over by TAT and not detected in the TAX19 + T9αH-725A experiment. **b**, MS/MS fragmentation patterns of heterologously produced taxusin (**6**) in *N. benthamiana* compared to that of taxusin (**6**) standard. MS/MS fragmentations were generated using [M+Na]$^+$ (m/z = 527.2621) as the precursor ion and fragmented with a collision energy of 30 eV. Regions between m/z 100 to 800 are shown. The following gene set was heterologously expressed in *N. benthamiana* via

*Agrobacterium*-mediated transient expression: tHMGR, GGPPS, TDS, FoTO, T5αH, TAT, T10βH, DBAT, T13αH, T9αH-750C, and TAX19. **c**, Feeding experiment reveals the bi-functional role of acetyltransferase TAT. To investigate the origin of the acetyl group on 9α-hydroxy in taxusin (**6**) and study the biosynthetic routes, we fed purified 3O2A (**4**) to *N. benthamiana* leaves expressing different combinations of T9αH-750C, TAT, DBAT, and TAX19. We found that only TAX19, the previously reported C-13α-O-acetyltransferase, yielded an acetylated 3O3A product and neither DBAT nor TAT did. Both T9αH-750C and T9αH-750C + TAX19 condition resulted in the formation of two isomers of 4O2A and 4O3A, respectively, which we proposed is due to the spontaneous acyl migration between 9α- and 10β-hydroxyl groups. As taxane scaffolds are highly strained and congested, acyl migration of acetyl groups are poised to happen, for example, the migration from 10β- to 7β-hydroxy[73], and from 2α- to 5α-hydroxy[74] have been reported. T9αH-750C + TAT generated a single 4O3A acetylated product different from the two isomers generated by T9αH-750C + TAX19. This suggests a 9α-*O*-acetylated product and the involvement of TAT on 9α-*O*-acetylation. Lastly, T9αH-750C + TAT + TAX19 condition successfully resulted in the production of taxusin (**6**). Overall, these data suggest a metabolic network model from 3O2A (**4**) to 4O4A involving TAT, TAX19, and T9αH-750C, and reveal the bi-functional role of TAT on both 5α- and 9α-*O*-acetylation.

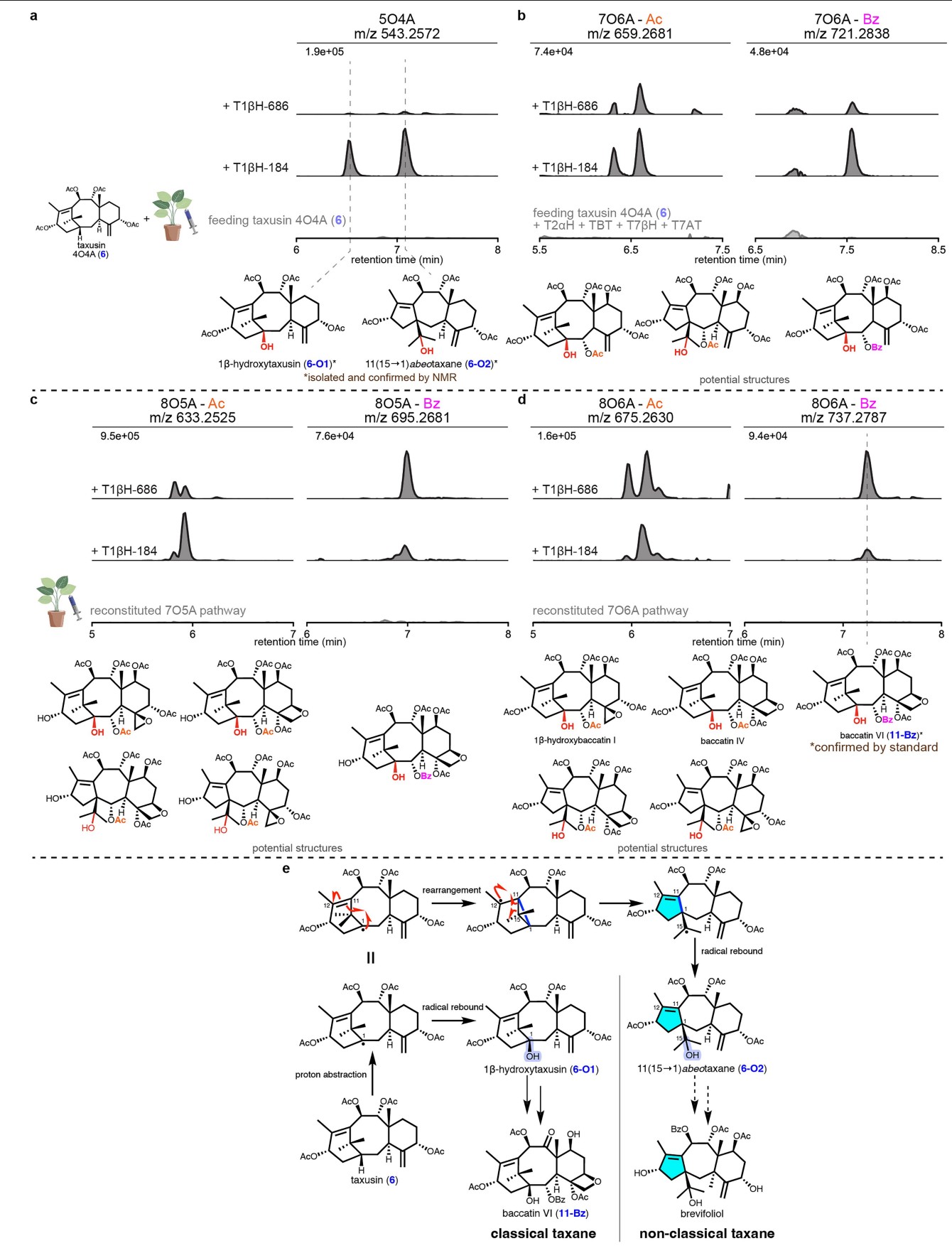

**Extended Data Fig. 5** | See next page for caption.

**Extended Data Fig. 5 | Product profiles of T1βH-184 and T1βH-686 with different upstream pathways and the proposed mechanism for 1β-hydroxylation and *abeo*-taxane rearrangement.** Two 2-ODDs, T1βH-184 and T1βH-686, were expressed in *N. benthamiana* with four different background conditions: **a**, Feeding taxusin. **b**, Feeding taxusin and co-expressing T2αH, TBT, T7βH, and T7Ac. **c**, Co-expressing full biosynthetic pathway to 7O5A, i.e. tHMGR, GGPPS, TDS, FOTO, T5αH, TAT, T10βH, DBAT, T13αH, T2αH, TBT, T7βH, T7AT, TOT. **d**, Co-expressing full biosynthetic pathway to 7O6A, i.e. 7O5A pathway and TAX19. EIC of mono-oxidized products on both C-2α-*O*-acetyl and -benzoyl precursors are shown as well as their potential structures. Among all products, 1β-hydroxytaxusin (**6-O1**) and 15-hydroxy-11(15→1)*abeo*-taxusin (**6-O2**) are confirmed by NMR (Supplementary Tables 5–7) while baccatin VI (**11-Bz**) is confirmed by comparing to chemical standard (Fig. 4c,e). The observed product diversity (formation of multiple peaks) is likely to arise from the dual-function of T1βH and TOT: T1βH performs both 1β-hydroxylation and rearrangement to 11(15→1)*abeo*-taxane, and TOT generates both epoxide and oxetane products. Notably, C-2α-*O*-benzoylated products mostly show as a single dominant peak. This suggests that the T1βH 1β-hydroxylation activity is selective toward C-2α-*O*-benzoylated/oxetane intermediates. **e**, Proposed mechanism of T1βH. Formation of the two characterized products 1β-hydroxytaxusin (**6-O1**) and 15-hydroxy-11(15→1)*abeo*-taxusin (**6-O2**) from taxusin (**6**) by T1βH-184 can be explained by the different fates of the C-1 radical after the first proton abstraction: direct hydroxyl radical rebound from the enzyme would yield 1β-hydroxytaxusin (**6-O1**) while radical rearrangement forming C-1/11 bond followed by ring opening and hydroxyl radical rebound would give 15-hydroxy-11(15→1)*abeo*-taxusin (**6-O2**). The rearrangement route is likely to lead to many non-classical *abeo*-taxanes, like brevifoliol, while the 1β-hydroxylation leads to classical taxanes like baccatin VI (**16**).

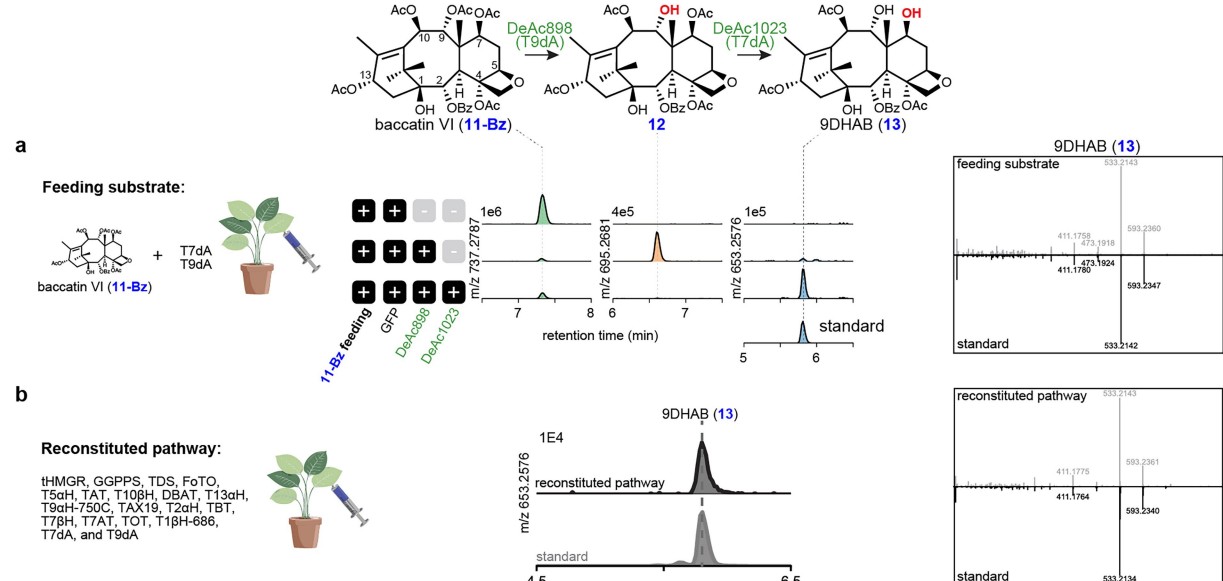

**Extended Data Fig. 6 | Characterization of deacetylases T7dA and T9dA.**
**a**, Conversion of baccatin VI (**11-Bz**) to 9-dihydro-13α-acetylbaccatin III (9DHAB, **13**) by T7dA and T9dA in *N. benthamiana* leaves. T7dA and T9dA were heterologously expressed in *N. benthamiana* via *Agrobacterium*-mediated transient expression and baccatin VI (**11-Bz**) 20 μM was fed to the leaves three days after infiltration. **b**, Reconstitution of 9DHAB (**13**) biosynthetic pathway in *N. benthamiana*. EICs of leaves expressing the reconstituted pathway compared to the 9DHAB standard are shown. MS/MS fragmentations were generated using [M+Na]⁺ (m/z = 653.2576) as the precursor ion and fragmented with a collision energy of 30 eV. Regions between m/z 100 and 800 are shown.

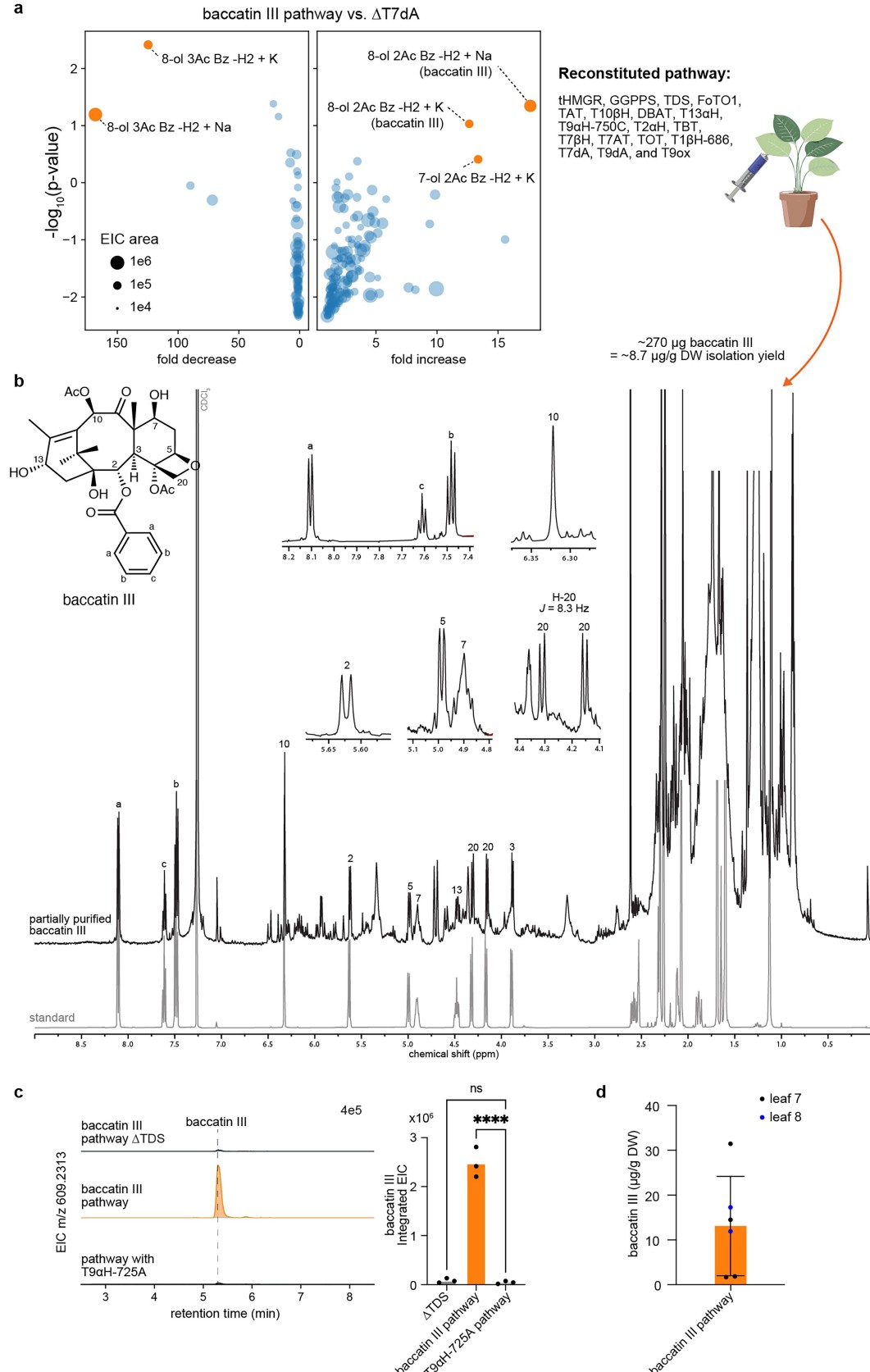

**Extended Data Fig. 7** | See next page for caption.

**Extended Data Fig. 7 | Heterologous production of baccatin III (16) in**
***N. benthamiana. a***, Untargeted analysis shows baccatin III (**16**) as a major
product in the final step of pathway. Linear volcano plot comparing
*N. benthamiana* leaves expressing the full baccatin III (**16**) pathway and full
pathway without the penultimate enzyme T7dA (ΔT7dA). Metabolomic
features with putative taxane masses are shown as dots whose sizes indicate
EIC integrated area. P-values calculated by two-sided t-test with a Bonferroni
correction for multiple hypothesis testing. The full baccatin III (**16**) pathway
include: tHMGR, GGPPS, TDS, FoTO1, T5αH, TAT, T10βH, DBAT, T13αH,
T9αH-750C, T2αH, TBT, T7βH, T7AT, TOT, T1βH-686, T9dA, T7dA, and T9ox.
**b**, $^1$H-NMR spectra of partially purified baccatin III (**16**) from *N. benthamiana*
and baccatin III (**16**) standard (CDCl$_3$, 500 MHz, 298 K). The spectra of our
partially purified baccatin III (**16**) align with the standard, exhibiting all
characteristic peaks (labelled with carbon number) as well as the H-20 coupling
constant ($J$ = 8.3 Hz) of the oxetane. The full baccatin III (**16**) pathway excluding
T5αH was used to infiltrate 53 *N. benthamiana* plants [30.70 g DW] to yield
baccatin III (**16**), whose yield (~270 μg) is derived from the total yield (1.33 mg)
with an estimated 20% purity. **c**, T9αH-725A cannot complement T9αH-750C
for baccatin III (**16**) production. Representative EIC of *N. benthamiana* leaves
expressing our full baccatin III pathway gene set, compared to full gene set
without TDS (ΔTDS) or an exchange of our T9αH-750C for the recently
reported T9αH-725A[12-14]. Bar graph of integrated EICs for baccatin III (**16**)
are shown. T9αH-725A pathway yields negligible baccatin III, statistically
indistinguishable to a ΔTDS negative control. This would be expected from
our finding that T9αH-725A appears to require a 13α-*O*-acetylation (Fig. 3b)
that is absent from our baccatin III (**16**) pathway and from Taxol. Data are shown
as the mean, *n* = 3 biological replicates. Significance indicates results of an
ordinary one-way ANOVA comparison to the full pathway (****$P$ = 5.63 × 10$^{-6}$).
**d**, Variations in baccatin III (**16**) production. Bar graph showing baccatin III (**16**)
yields from *N. benthamiana* leaves expressing the full 17-gene pathway from
different plants, calculated based on a baccatin III standard curve. Each sample
(dot) was taken from the 7th or 8th leaf counted from the bottom from separate
plants under the same experimental conditions. The yields range from
1.7 - 31.5 μg/g DW, comparable to those reported for the twig and leaf samples
from *T. chinensis, T. cupsidata*, and *T. media*[75]. Furthermore, our isolated yield
(~8.7 μg/g DW) aligns well with the calculated yield. Data are shown as the
mean ± standard deviation, *n* = 6.

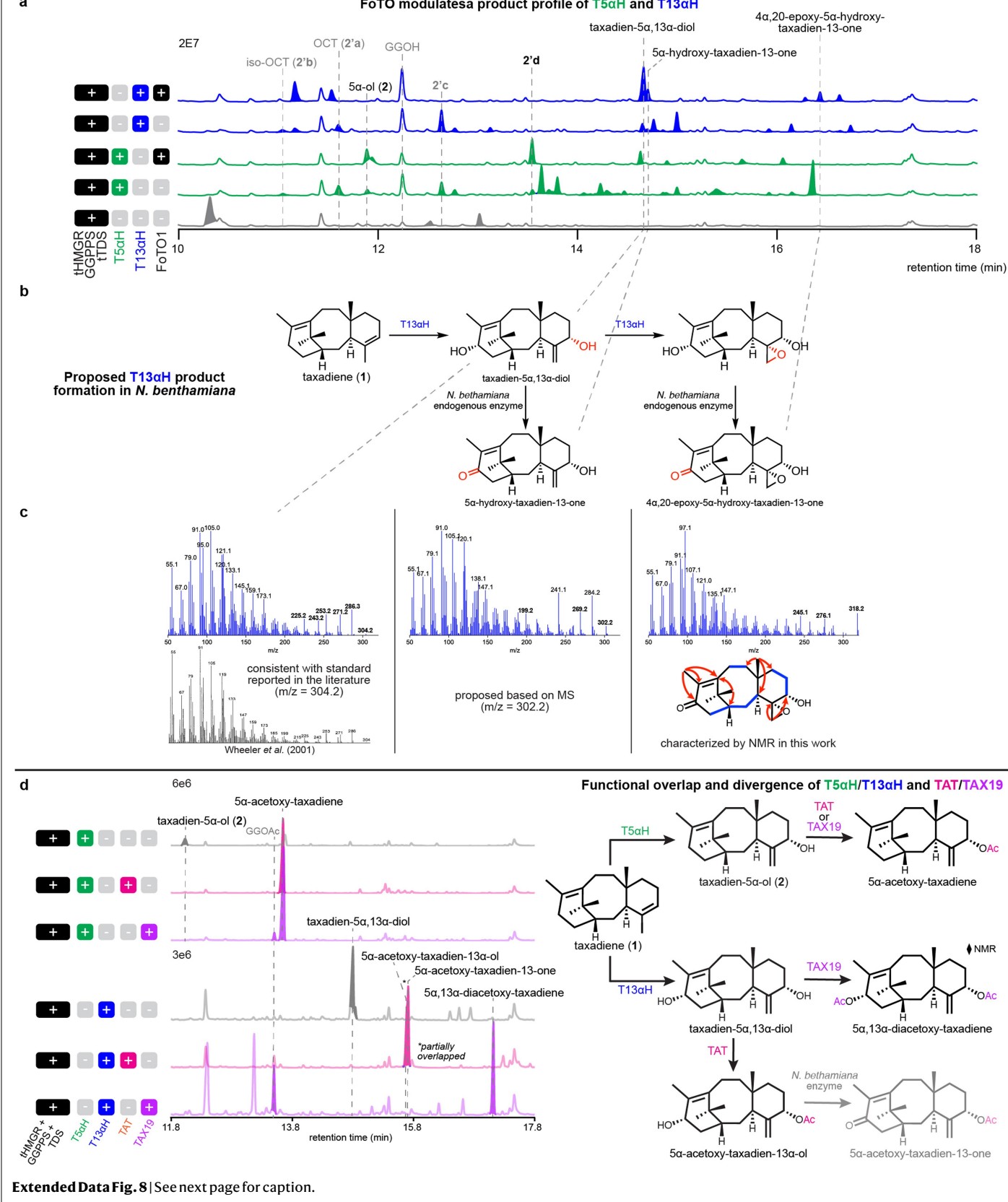

**Extended Data Fig. 8 |** See next page for caption.

**Extended Data Fig. 8 | Characterization of T13αH activity and products.**
**a**, FoTO1 modulates T13αH oxidation on taxadiene (**1**). GC–MS TIC of
*N. benthamiana* leaves expressing TDS + T13αH with and without FoTO1 and the
mass spectra (MS) of selected peaks are shown. Without FoTO1, T13αH oxidizes
taxadiene (**1**) to multiple products, including OCT (**2′a**), iso-OCT (**2′b**), rearranged
products (**2′c**) that are also made by T5αH, and other uncharacterized products.
Similar to the effect of FoTO1 on T5αH, FoTO1 significantly changes the product
profile of T13αH by suppressing the formation of **2′a**–**c** and selectively boosts the
production of several products. **b**, Three of these increased products are shown
here and annotated with their corresponding molecular structures (either
proposed or confirmed). The structure of taxadien-5α,13α-diol (m/z 304.2) is
confirmed based on MS comparison to previously published MS spectrum[76]
and its subsequent utilization by TAT and TAX19. The structure of 4α,20-epoxy-
5α-hydroxy-taxadien-13-one is confirmed after purification and NMR analysis

(Supplementary Table 8). Regions between m/z 50 to 320 are shown for the
MS spectrum. **c**, Proposed route for the formation of these taxanes in
*N. benthamiana*. **d**, Products of T5αH and T13αH can be acetylated by TAT or
TAX19. GC–MS TIC of *N. benthamiana* leaves expressing the indicated genes are
shown. Both TAX19 and TAT can acetylate taxadien-5α-ol (**2**) to 5α-acetoxy-
taxadiene. However, TAX19 and TAT acetylate T13αH products differently: TAT
results in two overlapping, major products, presumably 5α-acetoxy-taxadien-
13α-ol and 5α-acetoxy-taxadien-13-one (proposed based on MS spectra), while
TAX19 yields one major product, 5α,13α-diacetoxy-taxadiene, whose structure is
confirmed by NMR (Supplementary Table 10). These data are consistent with
previous characterization of TAX19's regioselective 5α- and 13α-*O*-acetylation
activity[39]. All acetylated products are proposed based on the characterized
TAT/TAX19 functions and MS fragmentation patterns.

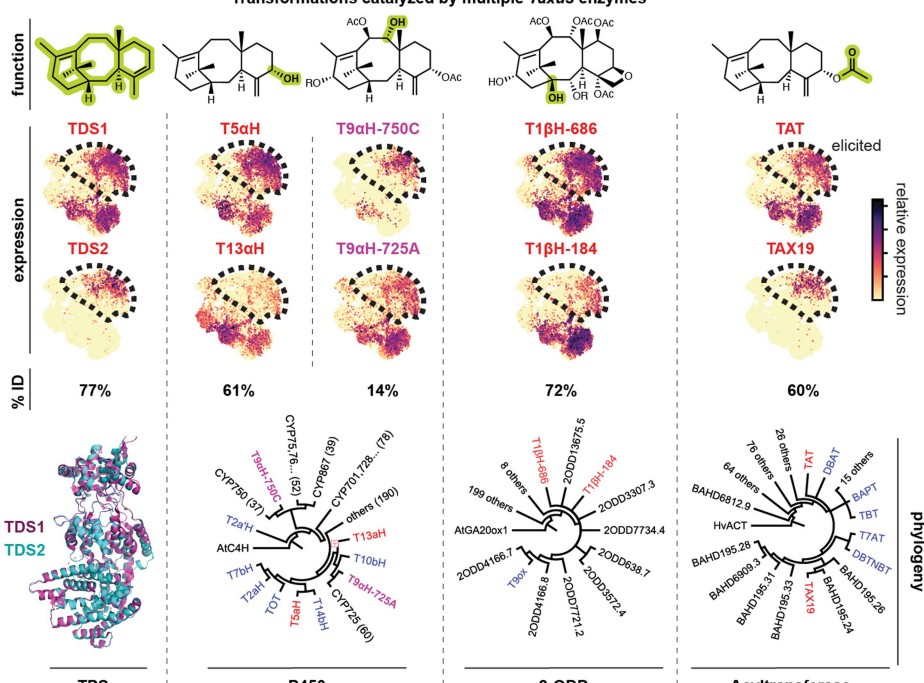

**Extended Data Fig. 9 | Functional redundancy in Taxol biosynthesis.**
Five pairs of functionally redundant enzymes in the Taxol biosynthetic pathway with their functions, single-cell expression patterns, percentage identity (% ID) in protein sequence, and protein structures or phylogenetic trees shown. Beginning with the first committed step of taxane biosynthesis, which can be catalysed by either of two TDS paralogues[5], we found that many taxane functional groups could be installed by multiple genes with different sequences and expression patterns. In the extreme case of the T9αHs, highly divergent sequences imply convergent evolution. For the T9αH and T1βH variants we found, different substrate and product specificity suggests that these enzymes may be involved in different branches of taxane metabolism, like C-13α-acetoxy taxanes and 11(15→1)*abeo*-taxanes. Regardless of the

mechanism, the prevalence of functional redundancy across the four major enzyme classes in taxane biosynthesis complicates both the dissection of various branches of endogenous taxane metabolism and the identification of optimal enzyme sets for heterologous taxane production. Crystal structure of TDS1 (Protein Data Bank 3P5P) and the AlphaFold3 predicted structure of TDS2 are shown. Phylogenetic trees of the three main tailoring enzyme families involved in Taxol biosynthesis − P450s, 2-ODDs, and acyltransferases. Protein sequences from each family were aligned using Clustal Omega, and the phylogenetic trees were constructed using the neighbour-joining method in Geneious Prime with 1,000 bootstrap replicates. AtC4H, AtGA20ox1, and HvACT were selected as outgroups for each family.

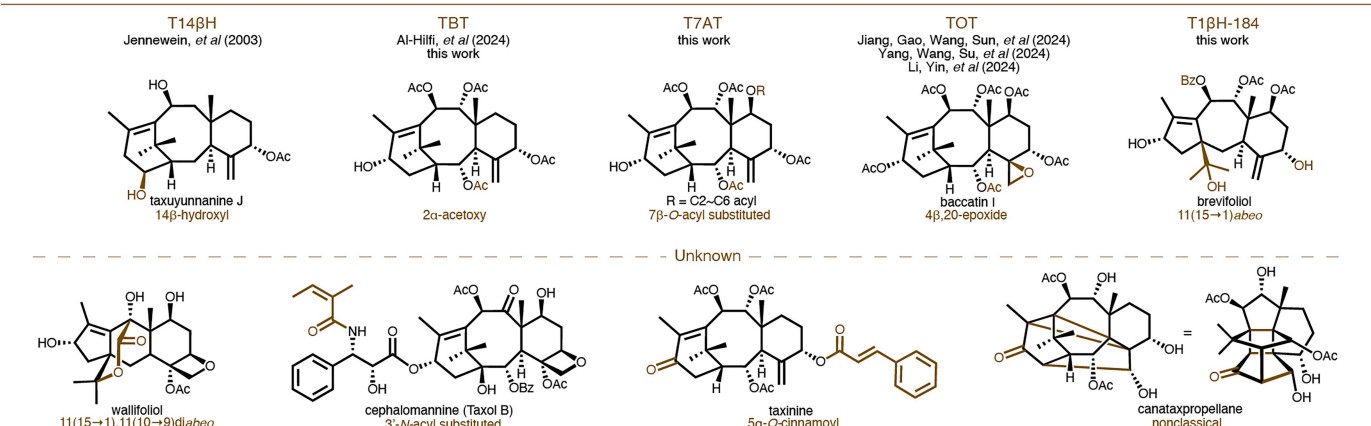

**Extended Data Fig. 10 |** See next page for caption.

**Extended Data Fig. 10 | Biosynthetic pathway reconstituted in *N. benthamiana* and proposed final steps to Taxol.** In this work, we demonstrated the complete transformation from taxadiene (**1**) to 3′-*N*-debenzoyl-2′-deoxypaclitaxel, which is proposed to be two steps away to Taxol, in *N. benthamiana*. While T2′αH has been previously reported[45,47], we were unable to reconstitute its activity. The first oxidation of taxadiene (**1**) can be conducted by either T5αH or T13αH, under the modulation by FoTO1. Branching of the Taxol pathway occurs at multiple enzymatic steps, including previously characterized T14βH[77], and more recently reported TBT[43], T7AT, TOT[11–13], and T1βH-184, leading to other taxanes (shown in the last panel). The structural differences of other taxanes to Taxol are highlighted in brown. Furthermore, TAX19[39] allows the reconstitution of branching pathway to 13α-acetoxy taxanes including taxusin (**6**), baccatin IV, baccatin VI (**11-Bz**), and 9-dihydro-13-acetylbaccatin III (9DHAB, **13**) that can be confirmed with chemical standards (Fig. 4e, Extended Data Figs. 4b and 6b). However, the potential roles of these 13α-acetoxy taxanes as alternative intermediates toward baccatin III remain to be explored. The prevalence of these 13α-acetoxy taxanes in *Taxus* plants suggests it is difficult to rule out that the natural biosynthetic pathway might proceed through 13α-acetoxy intermediates. However, we were unable to identify a corresponding C-13α-*O*-deacetylase (question mark from **13** to **16**), which prevented us from accessing baccatin III (**16**) via this putative, alternative route. Similarly, 10-deacetylbaccatin III, an abundant taxane in most *Taxus* species, might be an alternative biosynthetic intermediate with DBAT catalysing a late-stage acetylation (question mark from **13** to 10-deacetylbaccatin III)[78]. Full enzyme names are provided in Supplementary Table 1.

# Reporting Summary

## Statistics

For all statistical analyses, confirm that the following items are present in the figure legend, table legend, main text, or Methods section.

| n/a | Confirmed | |
|---|---|---|
| ☐ | ☒ | The exact sample size (*n*) for each experimental group/condition, given as a discrete number and unit of measurement |
| ☐ | ☒ | A statement on whether measurements were taken from distinct samples or whether the same sample was measured repeatedly |
| ☐ | ☒ | The statistical test(s) used AND whether they are one- or two-sided<br>*Only common tests should be described solely by name; describe more complex techniques in the Methods section.* |
| ☒ | ☐ | A description of all covariates tested |
| ☐ | ☒ | A description of any assumptions or corrections, such as tests of normality and adjustment for multiple comparisons |
| ☐ | ☒ | A full description of the statistical parameters including central tendency (e.g. means) or other basic estimates (e.g. regression coefficient) AND variation (e.g. standard deviation) or associated estimates of uncertainty (e.g. confidence intervals) |
| ☐ | ☒ | For null hypothesis testing, the test statistic (e.g. *F*, *t*, *r*) with confidence intervals, effect sizes, degrees of freedom and *P* value noted<br>*Give P values as exact values whenever suitable.* |
| ☒ | ☐ | For Bayesian analysis, information on the choice of priors and Markov chain Monte Carlo settings |
| ☒ | ☐ | For hierarchical and complex designs, identification of the appropriate level for tests and full reporting of outcomes |
| ☐ | ☒ | Estimates of effect sizes (e.g. Cohen's *d*, Pearson's *r*), indicating how they were calculated |

*Our web collection on statistics for biologists contains articles on many of the points above.*

## Software and code

Policy information about availability of computer code

**Data collection**

GCMS and LCMS data is collected with Agilent 7820MS/Enhanced MassHunter and Agilent MassHunter Workstation Data Acquisition version 10.1, respectively.
Sony SH800 Cell Sorter Software v2.2.4
FlowJo v10

**Data analysis**

Routine data compilation was performed in Microsoft Excel (version 16.8). GCMS data analysis was performed with Agilent MassHunter Qualitative Analysis B.07.00. LCMS data analysis was performed with Agilent MassHunter Qualitative Analysis 10.0. Bar graphs were plotted using GraphPad Prism 10. NMR data were processed and visualized on MestReNova vl4.3.l. Molecular weight calculation and chemical structural visualization were conducted with Chem Draw Professional v22.2.0. Python code used for transcriptomic and untargeted metabolomic analysis will be deposited on GitHub. Published transcriptome datasets were cleaned with trimmomatic and mapped to the Taxus chinensis genome with STARmap.Ambient RNA was removed with cellbender (v.0.3.0) 57. Using the doubletdetection (v.4.2) library58, doublets were removed, as well as cells with outlier numbers of reads or where most reads were the most expressed genes (pct_counts_in_top_20_genes < 25).

For manuscripts utilizing custom algorithms or software that are central to the research but not yet described in published literature, software must be made available to editors and reviewers. We strongly encourage code deposition in a community repository (e.g. GitHub). See the Nature Portfolio guidelines for submitting code & software for further information.

## Data

Policy information about availability of data

All manuscripts must include a data availability statement. This statement should provide the following information, where applicable:

- Accession codes, unique identifiers, or web links for publicly available datasets
- A description of any restrictions on data availability
- For clinical datasets or third party data, please ensure that the statement adheres to our policy

Raw and processed single-nuclei transcriptome data have been deposited at NCBI Gene Expression Omnibus (accession GSE292840). The bulk RNA-seq data from six previous studies for comparison were downloaded from NCBI (accession PRJNA493167, PRJNA251671, PRJNA733140, PRJNA427840, PRJNA497542, PRJNA499080, PRJNA86408). Python code used for transcriptomic and metabolomic analysis are deposited on GitHub (https://github.com/mcclune/nature2025). The raw NMR free induction decay (FID) data of individual compounds have been deposited in the Natural Products Magnetic Resonance Database (np-mrd.org) with the following ID: $4\alpha$,20-epoxy-taxadien-5$\alpha$-ol (2'd, NP0350670), $4\alpha$,20-epoxy-5$\alpha$-hydroxy-taxadien-13-one (NP0350671), $5\alpha$,13$\alpha$-diacetoxy-taxadiene (NP0350849), taxusin (6, NP0341906), 13$\beta$-taxusin (6', NP0341907), 1$\beta$-hydroxytaxusin (6-O1, NP0341908), 15-hydroxy-11(15→1)abeo-taxusin (6-O2, NP0341909); the processed NMR data are shown in Fig. S23-65 and Table S3-10.

## Research involving human participants, their data, or biological material

Policy information about studies with human participants or human data. See also policy information about sex, gender (identity/presentation), and sexual orientation and race, ethnicity and racism.

| | |
|---|---|
| Reporting on sex and gender | N/A |
| Reporting on race, ethnicity, or other socially relevant groupings | N/A |
| Population characteristics | N/A |
| Recruitment | N/A |
| Ethics oversight | N/A |

Note that full information on the approval of the study protocol must also be provided in the manuscript.

# Field-specific reporting

Please select the one below that is the best fit for your research. If you are not sure, read the appropriate sections before making your selection.

☒ Life sciences  ☐ Behavioural & social sciences  ☐ Ecological, evolutionary & environmental sciences

For a reference copy of the document with all sections, see nature.com/documents/nr-reporting-summary-flat.pdf

# Life sciences study design

All studies must disclose on these points even when the disclosure is negative.

| | |
|---|---|
| Sample size | All experiments in this manuscript were conducted with a sample size of at least three to ensure minimal statistical power for analysis. |
| Data exclusions | No data were excluded during analysis. |
| Replication | All experiments were replicated at least once, and in most circumstances, in greater than three independent experiments. |
| Randomization | Randomization is not relevant to the experiments of this manuscript. The various experimental conditions were specifically defined to probe for the function of distinct enzymes, and there was no random assigning of samples to experiment groups. |
| Blinding | Blinding was not relevant to data collection as the metabolic data were acquired in an unbiased manner on automatic LCMS/GCMS instruments, where all detectable metabolites were measured without prior assumptions. |

# Reporting for specific materials, systems and methods

We require information from authors about some types of materials, experimental systems and methods used in many studies. Here, indicate whether each material, system or method listed is relevant to your study. If you are not sure if a list item applies to your research, read the appropriate section before selecting a response.

## Materials & experimental systems

| n/a | Involved in the study |
|---|---|
| ☐ | ☒ Antibodies |
| ☒ | ☐ Eukaryotic cell lines |
| ☒ | ☐ Palaeontology and archaeology |
| ☒ | ☐ Animals and other organisms |
| ☒ | ☐ Clinical data |
| ☒ | ☐ Dual use research of concern |
| ☐ | ☒ Plants |

## Methods

| n/a | Involved in the study |
|---|---|
| ☒ | ☐ ChIP-seq |
| ☐ | ☒ Flow cytometry |
| ☒ | ☐ MRI-based neuroimaging |

# Antibodies

| | |
|---|---|
| Antibodies used | Mouse monoclonal anti-VS antibody (SVS-Pkl) (Invitrogen #R960-2S) for detection of VS-tagged proteins by immunoblot in the Nicotiana benthamiana gene expression system used.<br>HA tag horseradish peroxidase-conjugated antibody (Biotechne #HAM0601) for detection of HA-tagged proteins by immunoblot in Nicotiana benthamiana. Dilution for each antibody is indicated in the method. |
| Validation | The mouse monoclonal anti-VS antibody (Invitrogen #R960-2S) is validated by the manufacturer to be specific to VS-tagged proteins by immunoblot, immunofluorescence microscopy, and functional analysis of the antibody against a fusion protein containing a VS epitope. Certificates of analysis and antibody validation are available on the manufacturers website (https://www.thermofisher.com/antibody/product/VS-Tag-Antibodyclone-SVS-Pkl-Monoelonal/R960-25)<br>The HA tag horseradish peroxidase-conjugated antibody (Biotechne #HAM0601) is validated by the manufacturer to be specific to HA-tagged proteins via immunoblot. Certificates of analysis and antibody validation are available on the manufacturers website (https://www.rndsystems.com/products/ha-tag-horseradish-peroxidase-conjugated-antibody-1049f _ham 0601) |

# Dual use research of concern

Policy information about dual use research of concern

## Hazards

Could the accidental, deliberate or reckless misuse of agents or technologies generated in the work, or the application of information presented in the manuscript, pose a threat to:

| No | Yes | |
|---|---|---|
| ☒ | ☐ | Public health |
| ☒ | ☐ | National security |
| ☒ | ☐ | Crops and/or livestock |
| ☒ | ☐ | Ecosystems |
| ☒ | ☐ | Any other significant area |

## Experiments of concern

Does the work involve any of these experiments of concern:

| No | Yes | |
|---|---|---|
| ☒ | ☐ | Demonstrate how to render a vaccine ineffective |
| ☒ | ☐ | Confer resistance to therapeutically useful antibiotics or antiviral agents |
| ☒ | ☐ | Enhance the virulence of a pathogen or render a nonpathogen virulent |
| ☒ | ☐ | Increase transmissibility of a pathogen |
| ☒ | ☐ | Alter the host range of a pathogen |
| ☒ | ☐ | Enable evasion of diagnostic/detection modalities |
| ☒ | ☐ | Enable the weaponization of a biological agent or toxin |
| ☒ | ☐ | Any other potentially harmful combination of experiments and agents |

# Plants

| | |
|---|---|
| Seed stocks | Nicotiana benthamiana plants used in this study are a gift from the Mudgett lab (Stanford) and propagated in house. Taxus plants are obtained from FastGrowingTrees. |
| Novel plant genotypes | No novel plant genotype was produced. Only Agrobacterium-mediated transient expression was used in this manuscript. |
| Authentication | N/A |

# Flow Cytometry

## Plots

Confirm that:

☒ The axis labels state the marker and fluorochrome used (e.g. CD4-FITC).

☒ The axis scales are clearly visible. Include numbers along axes only for bottom left plot of group (a 'group' is an analysis of identical markers).

☒ All plots are contour plots with outliers or pseudocolor plots.

☒ A numerical value for number of cells or percentage (with statistics) is provided.

## Methodology

| | |
|---|---|
| Sample preparation | Nuclei were extracted as described in Methods section, resuspended in 1 ml NIB with 5 ng/ul 4,6-diamidino-2-phenylindole and 5 ng/ul propidium iodide and sorted on a Sony SH800 cell sorter with a 70 μm chip |
| Instrument | Sony SH800 |
| Software | Sony SH800 software used during sorting and FlowJo vl0 was used for post-sorting data analysis. |
| Cell population abundance | For single-nuclei sequencing experiments, 150,000-200,000 nuclei were sorted prior to sequencing. During initial experiments, nuclei were examined under the microscope determine that that they were still spherical and intact. Nuclei quality could also be determined by the narrow width of the peak in DAPI and Pl channels, indicating homogeneous nucleus size. Nuclei were extremely abundant (>30% of particles in sorted populations). |
| Gating strategy | Nuclei are gated on three subsequent gates: (i) size selection using forward scatter (FSC) vs side scatter (SSC), (ii) singlet selection using Pl fluorescence height and width, and (iii) co-staining with DAPI and Pl to identify clean nuclei. |

☒ Tick this box to confirm that a figure exemplifying the gating strategy is provided in the Supplementary Information.

