## [Peer Review File · Nature]

Discovery of FoTO1 and Taxol genes enables biosynthesis of baccatin III

Corresponding Author: Dr Elizabeth Sattely

Version 0:

Reviewer comments:

Referee #1

(Remarks to the Author)

A and B: The manuscript by McClune, Liu, et al. investigates the biosynthetic pathway of paclitaxel from yew trees. Paclitaxel is one of the most famous natural products from plants and a clinically relevant anticancer drug. Many groups have contributed over the years to provide a substantial, but still incomplete picture of its biosynthesis in plants. Building on this state of the art, the current manuscript now makes major contributions in three different aspects:

- 1) The authors introduce a multiplexed perturbation single cell transcriptome workflow (termed mpXsn) which successfully generated cell states actively expressing paclitaxel biosynthetic genes. While a few studies have recently used single cell transcriptome analyses to elucidate biosynthetic pathways in plants (e.g., Nature Chem Biol 2023, 19, 1031-1041 or Mol Plant 2024, 17, 1439-1457), previous approaches have strongly relied on individual specialised cell types that highly express certain pathway sections. The approach by Sattely and co-authors in this work now is a drastic improvement, as it can also be used in cases where biosynthetic pathways are only induced under certain conditions, even when these conditions are not completely known. I therefore believe that this highly original approach can become a key strategy to elucidate biosynthetic pathways that lack obvious co-expression patterns for the whole field.
- 2) As a demonstration of the potential of the multiplexed perturbation approach, the authors discover a nuclear transport factor 2-like protein (FoTO1) that markedly improves the product profile of taxane oxidation. This is a very exciting finding, as such a protein has never been linked to specialised metabolism before. Importantly, I would argue that by traditional co-expression experiments the discovery of this protein would have been extremely unlikely. The authors use microscale thermophoresis and co-immunoprecipitation to demonstrate that FoTO1 interacts with T5aH and TDS, even though all three enzymes show different subcellular localisation.
- 3) Lastly, from the single cell gene correlation dataset, the authors discover seven further genes of the pathway that allow them to produce the industrially important paclitaxel precursor baccatin III in the model plant *Nicotiana benthamiana* in amounts comparable to the natural content in *Taxus* needles, and almost complete the pathway to paclitaxel. The heterologous production of baccatin III has recently already been achieved by two other groups (refs 9 and 26 cited by the authors), but the authors describe well that (at least in their hands) the previously published gene sets for baccatin III yield substantially lower product (Figure 6c). As such, the gene discovery work described in this manuscript is an important improvement over the state of the art.

In summary, this manuscript redefines the state of the art how single cell sequencing can be used for pathway elucidation, describes a new type of protein involved in specialised metabolism, and strongly improves the heterologous production of baccatin III, precursor of the clinically used anticancer drug paclitaxel. In my opinion, this work will be a key contribution to the field. Nonetheless, there are several weaknesses that will need to be addressed in a revision (see section F).

C and D: The data presented in the manuscript was generally very convincing to me to support the interpretations of the authors. Presentation of data is done in an excellent way. Regarding the function of FoTO1, further data and experiments as suggested below (section F) would be appreciated to further substantiate that part of the manuscript. The presentation of NMR data could be slightly improved as suggested below.

I have no concerns about the use of statistics.

E: Overall, this manuscript appears highly robust to me and represents a tremendous work load. Nonetheless, I believe that improvements to the part on FoTO1 as well as many other minor improvements could further strengthen this work.

F: In my opinion, the discovery of FoTO1 is very exciting, but could be further improved by the authors. Most importantly, in my opinion, would be to provide more information on the modified product profile in the presence of FoTO1 (Figure 3e). The authors show that besides the desired product taxadien-5a-ol (2) two other products are formed, 2'd and 2'e. Unfortunately, almost no information is provided for these compounds. Do the authors have information on the structure of these compounds? If not, I would strongly encourage them to purify and characterise at least 2'd, ideally also the minor compound 2'e. I believe that this would provide important information regarding the change of activity induced by FoTO1.

Also, very little information is provided on the occurrence of FoTO1 homologues in plants. The authors tested another homologue from *Taxus* and one from *Arabidopsis* (Figure S4), but it remains unclear if these are the only ones or if these are part of larger gene families. It would be helpful if the authors could provide more details here, e.g., number of homologues per species, sequence identities between different homologues, phylogenetic analyses, etc.

The authors state: "Deletion of the C terminal helix, which breaks FoTO1's in planta metabolic phenotype, also eliminated binding affinity with both proteins, suggesting that this region may overlap with an important protein contact surface." Have the authors tried to use AlphaFold3 to make predictions about the interaction of FoTO1 with the other proteins?

I believe that the authors also slightly overstate the novelty of FoTO1 (e.g., "While all cytochromes P450 require a cytochrome P450 reductase (CPR) partner, this is the first example, to our knowledge, of a plant cytochrome P450 that acts in concert with an additional protein to guide product specificity."; "FoTO1 could be the first of several proteins with scaffolding roles"). While the authors acknowledge the recent discovery of GAME15 (ref 43), I strongly believe that they should also refer to work on MSBP1, originally found in the context of lignin biosynthesis (Nature Plants 2018, 4, 299-310). Recent application of MSBP1 in yeast has demonstrated that it can also shift the product profile towards higher oxidation products (Nature 2024, 629, 937-944).

The multiplexed perturbation approach by the authors is very interesting, but in my opinion the discussion of the perturbation conditions is a bit lacking. Apparently, some of their elicitation conditions successfully activate paclitaxel biosynthesis. Can the authors make any predictions or deductions from their mpXsn dataset which of the perturbations was critical for success? I could imagine that further information regarding elicitors of paclitaxel biosynthesis could be of high interest to many researchers.

Some of the nomenclature used by the authors can be a bit confusing. First, they encode the numbers of oxidations and acylations of metabolites (e.g., 3O2A), but this nomenclature is never introduced in the text. I would suggest that this should be properly introduced and explained. Also, some of the gene names appear unclear or inconsistent. For some P450s, the authors include the subfamily (e.g., T9aH-750C), but the meaning of the "750C" suffix only becomes clear later in the manuscript. For other enzymes, the meaning of the suffix number (e.g., T1bH-184) does not become clear at all. Is it the rank? But what is the meaning of something like AAE-867.5?

Figure 2a/b: Why is TAX19 shown in the PCC matrix but does not show up in the pathway?

Figure 2i: How is gene score defined? What does it mean?

Figure 3a: Stereochemistry of 2'a and 2'b is not shown completely. The presentation of the structures should be improved.

Figure 2 legend describes the meaning of green and orange colours, but unfortunately this colour code overlaps with the colours of the UMAP plot next to panel h. This might lead to confusion.

The use of species name appears inconsistent. Sometimes the authors write "*Taxus x media*", sometimes "*Taxus media*". This should be consistent or explained.

The presentation of NMR data could be improved in several regards:

- no double arrows should be used to indicate HMBC correlations; these are strictly unidirectional from a proton to a carbon (Table S4, Table S5). In the same light, all correlations that are shown should be carefully checked again.
- chemical shifts should be reported with consistent decimal places; please check Table S5 H-3, C-17; Table S3 C-1 ("38?"; what does the question mark mean?)
- authors should report temperature and frequency
- Table S3: Abbreviation N.A. is not defined

There are numerous very minor aspects (e.g., inconsistent use of FoTO1 vs. FOTO1; μ instead of μ ; compound numbers not bold; inconsistent reference to carbon atoms with or without hyphen; inconsistent or incorrect use of acyl vs. acetyl). These are highlighted in an annotated pdf file.

G: With the exception of previous work on MSBP1 (see section F), the authors provide appropriate credit to previous work.

H: In the abstract, the authors might want to clarify that the genus name *Taxus* refers to yew trees.

Signed
Jakob Franke

Referee #2

(Remarks to the Author)

This manuscript describes a clever and truly insightful study of paclitaxel (Taxol) biosynthesis. In particular, while the pathway to this vitally important anti-cancer agent has been the subject of investigation for decades, efforts to recombinantly transfer this into more amenable hosts than the native producing yews (from the *Taxus* genus) have been stymied not only by its incomplete nature (i.e., not all the necessary genes were known), but also the puzzling metabolic block at the first hydroxylation step, where the relevant enzyme produced relatively little of the desired product with others accumulating instead. This later issue has continued to plague even the recently reported studies from other groups published this year in *Science* (ref. 9) and *Molecular Plant* (ref. 26). Almost of all this is addressed here in what is a breakthrough study not only for this pathway, but that of such plant natural products more generally given the clever approach developed here. Specifically, as with others, these authors carried out RNA-Seq with bulk tissues isolated from yew plants along with elicitation known to induce Taxol production, but had limited (no?) success in identifying any new genes. However, here a novel clever 'multiplexed' approach was developed, using multiple perturbations of needles (including various timepoints) and then isolating single nuclei for transcriptome sequencing, which were then grouped by similarity to generate 'cell states' and co-expression modules. This led to three modules in which the known Taxol biosynthetic genes were highly ranked that then provided lists of other genes that were hypothesized to also play a role. The validity of this 'multiplex' approach were amply demonstrated by the various breakthroughs in Taxol biosynthesis reported here. Note that while the elucidated pathway is 'only' to the intermediate baccatin III, this is both the main taxane found in needles, and also the actually isolated compound that is subsequently converted to Taxol in a semi-synthetic route, so should still be considered a resounding success. In the context of the other recent studies, perhaps the most impactful and also surprising finding was the discovery of a scaffolding protein from the NTF2-like protein family (FoTO1) that modified the activity of the first hydroxylase to direct product outcome towards that relevant to taxane biosynthesis. All of the previously unknown enzymes for production of baccatin III also were identified, which revealed an intriguing role for 'excess' acylation in the pathway. At least in one case, the identified hydroxylase differs from that identified in the other recent studies. While that might raise questions, it seems most likely that identified here is correct, simply due to the efficiency of the reconstructed pathway reported here, which even in its currently unoptimized form equals that exhibited in yew needles themselves, relative to the rather anemic throughput reported in the other studies. Along these lines, although not a focus here, in the remaining steps from baccatin III to Taxol, the inefficiency of the previously identified 2'a-hydroxylase suggests that might be the relevant enzyme. Identification of the correct enzyme might be beyond the scope of this study, particularly given the uncertainty about Taxol actually being produced in needles (at least in any appreciable amounts), but seems worth mentioning. In any case, even beyond the important insights obtained into Taxol biosynthesis and strong potential for impact on commercial supply of this important anti-cancer agent, the developed multiplex approach to investigating plant biosynthetic pathways is expected to be widely applicable, adding further impact to this report.

The presentation is quite good, but could be bolstered in a few spots. For example, while the use of baccatin III in Taxol production is mentioned in the abstract and discussion, it should be noted in the introduction (first or last paragraph) and references provided (these should also be presented in the discussion). The issue about the previously identified 2'a-hydroxylase also should be raised at least in the results if not also discussion. With regard to the multiplexed approach, it seems to be implied that the needles were only elicited for a short time and then left in MS media until harvest (at the various time points), but this should be clarified and, if true, the exposure time specified in the methods. Also, it is not specified how the nuclei were then isolated from these needles (even if the same as with the bulk tissue samples that should be stated). Finally, a bit more explanation about how the tissues were 'chopped at ~200 rpm' should be provided (e.g., was this done with a machine or by hand?).

Referee #3

(Remarks to the Author)

This manuscript from the Sattely Lab represents an exceptional contribution to the field of plant natural product biosynthesis, and in particular the elusive pathway to Taxol. McClune et al. present a polished body of work that taken together provides a convincing, wholistic, and comprehensive story around baccatin III biosynthesis. In addition to identifying seven new genes (including an intriguing NTF2-like accessory), the group presents a newly developed "multiplexed perturbation" strategy. This approach will be adapted in future studies where bulk tissue RNA-seq fails to identify missing genes. The work both acknowledges overlapping scientific discoveries which have been recently made, while rectifying the disparate conclusions reached as a result of the complex nature of this biosynthetic pathway.

The authors came very close to de novo production of Taxol in *N. benthamiana* – which would have pushed this paper to the very top in terms of impact to a broader audience. However, the authors managed to produce baccatin III, an industrial precursor for Taxol, at titers higher than in native yew needles; this could have a real-world commercialization potential. Moreover, the authors identified independently regulated expression modules and provided evolutionary insight, two compelling premises in plant biosynthesis. If the following concerns can be addressed, I would recommend publication in this journal.

Major concerns:

Major concern 1 – The authors should clearly indicate which structures are proposed and which have been confirmed by NMR. Only the structures of 6, 6', 6-O1, and 6-O2 are confirmed the NMR, the rest are proposed via comparison to authentic standards or mass spectra; these techniques cannot provide conclusive evidence of molecular structure. The use of the

phrase “structural analysis” implies NMR characterization. Minimally, the authors should perform NMR characterization of titular baccatin III isolated from *N. benthamiana*, to bolster the claim that “this work represents a major step towards sustainable production of Taxol.”

Major concern 2 – The authors must carefully explain why they chose the following cutoffs to facilitate gene identification and analysis using their new mpXsn methodology, otherwise it may appear to a reader that these cutoffs were arbitrarily selected to make the discovery/technique appear more straightforward:

- Why were 200 total gene expression modules analyzed?
- How were “many” and “top” defined when determining the cutoffs to identify modules (ie. “we found three where the TOP genes included MANY known Taxol enzymes”) and why were these values selected? (For example there appears to be another “module 4” with five Taxol enzymes, which is precluded for analysis based on these cutoffs.)
- Why was the value 100 chosen to rank the modules by specialized metabolism in Fig. 2g?
- Why was the value 125 chosen for number of unfiltered genes to display in Fig. 2h?

Minor concerns:

Minor concern 1 – The authors should add a TOC to the supplementary information.

Minor concern 2 – The authors use FoTO1 and FOTO1 interchangeably.

Minor concern 3 – The authors state, “When examining the 14 genes previously established to be Taxol enzymes (Table S1, Fig 2a), all but TBT were prioritized substantially higher using the mpXsn data rather than bulk RNA-seq data (Fig. 1f, S2-S3).” However besides TBT, DBTNBT also appears to be prioritized lower.

Minor concern 4 – The authors refer to “co-expressed subsets of the pathway (Fig. 2b)” but it is not clear to me from looking at Fig. 2b what is being referred to.

Minor concern 5 – The authors should describe how many candidates they tested by “coexpressing them with this six-enzyme early pathway in *N. benthamiana*” in order to identify gene #13 FoTO1, and ideally present this data.

Minor concern 6 – The authors should explain how they concluded that 2'e is a side product of T5 α H, as well as the nature of the 2'd more carefully.

Minor concern 7 – The authors should describe how many candidates they tested when they “further screened the oxidases (2-ODDs and P450s) of module 2, revealing two 2-ODDs (2-ODD-184 and 2-ODD-686) that yielded single-oxidized products when expressed with various upstream pathways” in order to identify T1 β H-686 and T1 β H-184, and ideally present this data.

Minor concern 8 – The authors should include a negative control in Fig. 4c baccatin VI trace (ie. the EIC of 737.2787 without TAX19).

Minor concern 9 – The authors should describe how many $\alpha\beta$ -hydrolase candidates they in order to identify DeAc898 and DeAc1023, how these were selected, and ideally present this data.

Minor concern 10 – The authors should make Fig. 5g more clear – as drawn the reader may mistake the (17) chromatograms for (18). I would suggest labelling the peaks with compound numbers as the structures are just above.

Minor concern 11 – The authors mention that “all cytochromes P450 require a cytochrome P450 reductase (CPR) partner” but this is not the case.

Minor concern 12 – The authors write that FoTO1 activity represents “the first example, to our knowledge, of a plant cytochrome P450 that acts in concert with an additional protein to guide product specificity.” In my opinion the phrase “in concert with” does not have a meaningful definition and should not be used, as it could be argued that any enzyme downstream of a P450 “guides” product specificity “in concert” with the P450.

Minor concern 13 – The authors claim “FOTO1 leads to such profound improvements in the activity of multiple *Taxus* oxidases” but also mentioned “FoTO1 may be involved in altering product flux directly at the first oxidation of taxadiene (1), as the yield improvement did not increase as the sub-pathway was extended beyond T5 α H (Fig. 3c)” – which is it?

Version 1:

Reviewer comments:

Referee #1

(Remarks to the Author)

I thank the authors for their very careful and extensive revisions. All my requests have been fully addressed. I congratulate the authors to this outstanding work and can now fully endorse publication of this manuscript.

Referee #2

(Remarks to the Author)

This manuscript has been suitably revised to address my previous concerns. However, there are a few minor points. First, it might help to emphasize the importance of the produced levels of baccatin III by noting the commercially farmed source are the needles (in the intro and discussion), to which this unoptimized pathway already compares. Second, when introducing the nOmA compound nomenclature used here it might be more accurate to term what is currently called “oxidations” as “oxy groups”. Third, it might be helpful to note the sequence identity between 2-ODD184 and 2-ODD686 (i.e., upon their introduction in the results). Finally, please clarify if NMR was used to identify the “structurally characterized ... 4a,20-epoxy-5a-hydroxy-taxadien-13-one”.

Referee #3

(Remarks to the Author)

All concerns have been addressed.

Referee #4

(Remarks to the Author)

I was asked to assess the single nucleus sequencing analysis specifically. My view is that this has been done carefully, and agree that this multiplexed approach is both clever and informative - borne out by the subsequent analysis of candidates and validation of function in the pathway. The single nucleus section is also well explained to the reader.

We thank the reviewers for their thorough reading and constructive feedback on our manuscript. Below, we provide point-by-point responses to each comment. Major updates are as follows:

1. **NMR confirmation of the heterologously produced baccatin III.** We isolated and confirmed the production of baccatin III from our complete 18-gene pathway through NMR analysis. This represents the first and only NMR evidence for baccatin III heterologous production, aside from two prior studies claiming its production.
2. **Distribution of FoTO1 across the plant kingdom.** We included phylogenomic analysis that shows FoTO1 to be conserved across land plants, including uncharacterized homologs in major crop species. Duplication of this conserved gene in the gymnosperm lineage likely enabled one paralog to evolve a role in taxane biosynthesis.
3. **Characterization of T5 α H product.** We isolated and structurally characterized compound **2'd**, another product generated by T5 α H besides taxadien-5 α -ol (**2**) when FoTO1 is present. The epoxy group in **2'd** provides further insights to the mechanism of T5 α H.

A complete list of updates are as follows:

- **Main text:**
 - **p6:** clarify the identification of Taxol modules (**Fig. S1**)
 - **p8:** discuss **2'd** structure (**Fig. 3a, Table S3, Supplementary Note 1**)
 - **p8:** analyze FoTO1 phylogeny (**Fig. 3f, S5**)
 - **p9:** clarify nomenclature of taxanes
 - **p14:** additional evidence for T13 α H activity (**Extended Data Figure 9**)
- **Method:**
 - **p36:** construction of FoTO1 phylogeny
- **Extended Data Figure:**
 - **Extended Data Figure 7:** baccatin III NMR
 - **Extended Data Figure 9:** T13 α H product profile and NMR characterization
- **Supplementary Information:**
 - **Note: 1** discuss mechanism of T5 α H and T13 α H
 - **Fig. S1:** gene module analysis with different k (total modules)
 - **Fig. S2:** elicitation impact on early taxol metabolites
 - **Fig. S4:** T5 α H product profile
 - **Fig. S5:** phylogenetic analysis of FoTO1 in gymnosperm
 - **Fig. S7:** raw gel images
 - **Fig. S9,S10,S17:** replicates of EIC traces
 - **Fig. S3, S14, S16:** Representative result of candidate screening and identification for FoTO1, T1 β H, and deacetylase, respectively.
 - **Table S3, Fig. S47-52:** NMR of T5 α H product **2'd**
 - **Table S8-9, Fig. S53-58:** NMR of T13 α H product
- **Supplementary File:**
 - List of top 2,000 genes in the three Taxol modules

Referee #1 (Remarks to the Author):

A and B: The manuscript by McClune, Liu, et al. investigates the biosynthetic pathway of paclitaxel from yew trees. Paclitaxel is one of the most famous natural products from plants and a clinically relevant anticancer drug. Many groups have contributed over the years to provide a substantial, but still incomplete picture of its biosynthesis in plants. Building on this state of the art, the current manuscript now makes major contributions in three different aspects:

1) The authors introduce a multiplexed perturbation single cell transcriptome workflow (termed mpXsn) which successfully generated cell states actively expressing paclitaxel biosynthetic genes. While a few studies have recently used single cell transcriptome analyses to elucidate biosynthetic pathways in plants (e.g., *Nature Chem Biol* 2023, 19, 1031-1041 or *Mol Plant* 2024, 17, 1439-1457), previous approaches have strongly relied on individual specialised cell types that highly express certain pathway sections. The approach by Sattely and co-authors in this work now is a drastic improvement, as it can also be used in cases where biosynthetic pathways are only induced under certain conditions, even when these conditions are not completely known. I therefore believe that this highly original approach can become a key strategy to elucidate biosynthetic pathways that lack obvious co-expression patterns for the whole field.

2) As a demonstration of the potential of the multiplexed perturbation approach, the authors discover a nuclear transport factor 2-like protein (FoTO1) that markedly improves the product profile of taxane oxidation. This is a very exciting finding, as such a protein has never been linked to specialised metabolism before. Importantly, I would argue that by traditional co-expression experiments the discovery of this protein would have been extremely unlikely. The authors use microscale thermophoresis and co-immunoprecipitation to demonstrate that FoTO1 interacts with T5aH and TDS, even though all three enzymes show different subcellular localisation.

3) Lastly, from the single cell gene correlation dataset, the authors discover seven further genes of the pathway that allow them to produce the industrially important paclitaxel precursor baccatin III in the model plant *Nicotiana benthamiana* in amounts comparable to the natural content in *Taxus* needles, and almost complete the pathway to paclitaxel. The heterologous production of baccatin III has recently already been achieved by two other groups (refs 9 and 26 cited by the authors), but the authors describe well that (at least in their hands) the previously published gene sets for baccatin III yield substantially lower product (Figure 6c). As such, the gene discovery work described in this manuscript is an important improvement over the state of the art.

In summary, this manuscript redefines the state of the art how single cell sequencing can be used for pathway elucidation, describes a new type of protein involved in specialised metabolism, and strongly improves the heterologous production of baccatin III, precursor of the clinically used anticancer drug paclitaxel. In my opinion, this work will be a key contribution to the field. Nonetheless, there are several weaknesses that will need to be addressed in a revision (see section F).

C and D: The data presented in the manuscript was generally very convincing to me to support the interpretations of the authors. Presentation of data is done in an excellent way. Regarding the function of FoTO1, further data and experiments as suggested below (section F) would be appreciated to further substantiate that part of the manuscript. The presentation of NMR data could be slightly improved as suggested below. I have no concerns about the use of statistics.

E: Overall, this manuscript appears highly robust to me and represents a tremendous work load. Nonetheless, I believe that improvements to the part on FoTO1 as well as many other minor improvements could further strengthen this work.

F: In my opinion, the discovery of FoTO1 is very exciting, but could be further improved by the authors. Most importantly, in my opinion, would be to provide more information on the modified product profile in the presence of FoTO1 (Figure 3e). The authors show that besides the desired product taxadien-5a-ol (2) two other products are formed, 2'd and 2'e. Unfortunately, almost no information is provided for these compounds. Do the authors have information on the structure of these compounds? If not, I would strongly encourage them to purify and characterise at least 2'd, ideally also the minor compound 2'e. I believe that this would provide important information regarding the change of activity induced by FoTO1.

We thank the reviewer for their interest in our discovery of FoTO1 and for their suggestions to improve the discussion of its product profile. Below, we provide additional details regarding compounds 2'd and 2'e:

1. Compound 2'd:

As reported in our previous work (*Nat. Commun.* 2024, 15, 1419), compound 2'd (compound 9 in previous manuscript) is derived from taxadien-5 α -ol (2) based on a feeding study where 2 was infiltrated into leaves expressing T5aH. From *Nat. Commun.* 2024, **Figure 2d**: [REDACTED]

[FIGURE REDACTED]

We have now successfully purified 2'd and confirmed its structure as 4 α ,20-epoxy-taxadien-5 α -ol. This detailed characterization using MS and 2D-NMR analysis, is now provided in **Table S3**, **Fig. S4**, and **Fig. S47-52**. The accumulation of 2'd suggests that FoTO1 facilitates the epoxidation by T5aH and/or stabilizes the epoxide product, which is consistent with previously proposed T5aH epoxidation mechanism. Detailed mechanistic discussion of T5aH and comparison with T13 α H is provided in **Supplementary Note 1**.

Table S3. ^{13}C and ^1H δ assignments as well as 2D-NMR correlations of 4 α ,20-epoxy-taxadien-5 α -ol (2'd) recorded in CDCl_3 .

We have added the following discussion to the main text describing our structural analysis of 2'd:

"We have previously observed 2'd as the main product when taxadien-5 α -ol (2) is fed to *N. benthamiana* expressing T5aH¹⁵, suggesting that it is a secondary oxidation that occurs when T5aH incubates with its product 2. Here, we confirmed this, through purification from *N. benthamiana* leaves expressing TDS, T5aH and FoTO. 1D- and 2D- nuclear magnetic resonance (NMR) analysis supports a structural assignment of 2'd as 4 β ,20-epoxy-taxadien-5 α -ol (**Fig. 3a**, **Table S3**, **Supplementary Note 1**). It is notable that the stereochemistry of the 4 β ,20-epoxy in 2'd has not been previously reported in any natural taxanes, which exclusively features a 4 α ,20-epoxy stereochemistry. It is possible that the formation of 2'd is due to overoxidation of taxadien-5 α -ol (2) that accumulates to an unnaturally elevated level by T5aH in this partially reconstituted pathway."

2. Compound 2'e:

While we were unable to obtain sufficient quantities of pure 2'e suitable for NMR analysis due to its low abundance. However, the MS spectra of 2'e (**Fig. S4**) reveals a fragmentation pattern consistent with taxanes, particularly in the region m/z 50–150. The observed parent ion mass (m/z 274) suggests that 2'e could potentially be a saturated form (+2H) of taxadiene. Additional work will be required to confirm its identity and significance in FoTO1-T5aH catalysis.

The following text has been added to the caption for **Fig. S4**:

“While we were not able to obtain sufficient amounts of **2'e** for structural characterization, the MS spectra of **2'e** reveals a fragmentation pattern consistent with taxanes, particularly in the region m/z 50–150. The observed parent ion mass (m/z 274) suggests that **2'e** could potentially be a saturated form (+2H) of taxadiene. Additional work will be required to confirm its identity and significance in FoTO1-T5 α H catalysis.”

Also, very little information is provided on the occurrence of FoTO1 homologues in plants. The authors tested another homologue from *Taxus* and one from *Arabidopsis* (Figure S4), but it remains unclear if these are the only ones or if these are part of larger gene families. It would be helpful if the authors could provide more details here, e.g., number of homologues per species, sequence identities between different homologues, phylogenetic analyses, etc.

We appreciate the helpful comments by the reviewer on the occurrence of FoTO1 homologues and have conducted additional phylogenetic analysis of the FoTO1 family. We have constructed a phylogenetic tree of FoTO1 homologs and included it as part of **Fig. 3 (Fig. 3f)**. As we discuss in the main text, this phylogenetic analysis shows that FoTO1 belongs to an uncharacterized clade of proteins that is found throughout land plants. The presence of a single FoTO1 ortholog in many plant genomes is suggestive of a widely conserved function. However, the gymnosperm lineage experienced both an ancient duplication of this protein (**Fig. 3f**) and a recent duplication (**Fig. S5**), either of which may have enabled FoTO1 to neofunctionalize and become involved in taxane metabolism.

We have added the following discussion of these results to the main text:

“To examine the distribution of the FoTO1 gene family in plants, we employed HMMER to construct a profile hidden markov model based on FoTO139 and identified 1,957 homologs across the 1KP, RefSeq and Uniprot databases. We found that FoTO1 homologs are widespread across Viridiplantae and generally present as a single copy per genome, suggesting a conserved function (Fig. 3f). Gymnosperms, however, contain multiple paralogs that derive from both an ancient duplication (**Fig. 3f**) and recent duplications, including in the genus *Taxus* (**Fig. S5**). These duplications may have allowed one paralog to evolve alternative functions and contribute to taxane biosynthesis. Supporting this functional divergence, we found that neither the FoTO1 paralog from *Taxus media* nor the homologue from *Arabidopsis thaliana* could produce the same metabolomic change for early Taxol pathway reconstitution as FoTO1 (**Fig. 3g, Extended Data Figure 2**).”

The authors state: “Deletion of the C terminal helix, which breaks FoTO1's in planta metabolic phenotype, also eliminated binding affinity with both proteins, suggesting that this region may overlap with an important protein contact surface.” Have the authors tried to use AlphaFold3 to make predictions about the interaction of FoTO1 with the other proteins?

We acknowledge the reviewer's suggestion to use AlphaFold3 to predict potential interactions between FoTO1 and the other proteins. While we have indeed attempted to use AlphaFold3 to explore these interactions, it did not generate high-confidence or reproducible multimeric complexes between FoTO1 and either T5 α H or TDS2. AlphaFold3 may have a limited ability to model these interactions due to limited (i) structural training data and (ii) homologous sequences (for AlphaFold's coevolution analysis) to leverage from Viridiplantae and Gymnospermae, especially for new types of protein interactions.

I believe that the authors also slightly overstate the novelty of FoTO1 (e.g., "While all cytochromes P450 require a cytochrome P450 reductase (CPR) partner, this is the first example, to our knowledge, of a plant cytochrome P450 that acts in concert with an additional protein to guide product specificity."; "FoTO1 could be the first of several proteins with scaffolding roles"). While the authors acknowledge the recent discovery of GAME15 (ref 43), I strongly believe that they should also refer to work on MSBP1, originally found in the context of lignin biosynthesis (Nature Plants 2018, 4, 299-310). Recent application of MSBP1 in yeast has demonstrated that it can also shift the product profile towards higher oxidation products (Nature 2024, 629, 937-944).

We thank the reviewer for highlighting the relevance of MSBP and its role as scaffold protein for P450s in lignin biosynthesis. We have revised the paragraph discussing scaffolding protein in plant metabolism to explicitly include MSBP. Additionally, we have clarified the wording in the sentence that the reviewer identified to more accurately restrict the discussion of additional facilitator proteins within the context of taxane biosynthesis. The revised text is shown below.

We added the following statement regarding MSBP to the main text :

"In *Arabidopsis*, membrane steroid-binding proteins (MSBPs) have been identified as the scaffolds for three lignin biosynthetic P450s on the ER membrane.⁵¹"

We also revised this statement (in the context of taxane metabolism):

"FoTO1 could be the first of several proteins with scaffolding roles"

to read:

"there might be additional facilitator proteins beyond FoTO1"

The multiplexed perturbation approach by the authors is very interesting, but in my opinion the discussion of the perturbation conditions is a bit lacking. Apparently, some of their elicitation conditions successfully activate paclitaxel biosynthesis. Can the authors make any predictions or deductions from their mpXsn dataset which of the perturbations was critical for success? I could imagine that further information regarding elicitors of paclitaxel biosynthesis could be of high interest to many researchers.

We appreciate the reviewer's insightful question regarding the perturbation conditions that successfully activated paclitaxel biosynthesis, which would be valuable for future research. To address this, we conducted further metabolomic analysis on the *Taxus* needles after multiplexed perturbation, which is now shown in **Fig. S2** (below). The results show that certain treatments, including chitosan and high concentrations of methyl jasmonate, lead to increased accumulation of taxusin but fail to increase the levels of paclitaxel and baccatin III beyond the mock control. This suggests that these elicitors may be particularly effective in activating the early steps of the paclitaxel biosynthetic pathway. The lack of measurable increases in paclitaxel and baccatin may be due to pre-existing accumulation of these metabolites in the tissue prior to elicitation or a potential delay in their biosynthetic response beyond the three-day timeframe. Our findings highlight chitosan and methyl jasmonate as key factors in modulating taxane biosynthesis and provide a foundation for further exploration of their regulatory roles.

We have noted these results in the main text:

"Elicitation was crucial for activating module 1, which consists of the early portion of the Taxol pathway, as it was not strongly expressed in the naive young and mature *Taxus* tissues we profiled (**Fig. 2h**). Indeed, metabolomic analysis suggests that elicitors like chitosan and methyl jasmonate lead to increased accumulation of an early pathway intermediate in *Taxus* needles (**Fig. S2**), in line with previous reports that these are elicitors of gymnosperm stress responses³³ and taxane biosynthesis³⁴."

Fig. S2. Accumulation of known taxanes after elicitation. Mature *Taxus media* needles were placed in MS media containing each indicated elicitor (see **Table S2** for details) for 3 days. Subsequently, tissues were lyophilized, homogenized and extracted (15 mg DW / mL) into 75% ACN/water. Paclitaxel and baccatin III were not observed to accumulate to higher levels after 3 days of any elicitor, possibly due to initial accumulated stores of these metabolites. In contrast, chitosan and high levels of methyl jasmonate (MeJA) result in increases in taxusin accumulation, suggesting these elicitors can activate expression of early pathway enzymes.

Some of the nomenclature used by the authors can be a bit confusing. First, they encode the numbers of oxidations and acylations of metabolites (e.g., 3O2A), but this nomenclature is never introduced in the text. I would suggest that this should be properly introduced and explained. Also, some of the gene names appear unclear or inconsistent. For some P450s, the authors include the subfamily (e.g., T9aH-750C), but the meaning of the "750C" suffix only becomes clear later in the manuscript. For other enzymes, the meaning of the suffix number (e.g., T1bH-184) does not become clear at all. Is it the rank? But what is the meaning of something like AAE-867.5?

We agree with the reviewer on the need to clarify the nomenclature used in our manuscript. Below, we outline the steps taken to ensure the nomenclature is properly introduced and consistently applied:

1. Taxane Nomenclature (nOmA):

We have introduced the definition of the taxane nomenclature (nOmA) at its first appearance on page 9. For clarity, this system describes the collection of taxane isomers bearing n oxidations (e.g., hydroxylation and epoxidation) and m acylations (e.g., acetylation and benzoylation) on the taxadiene scaffold. This explanation provides readers with the necessary context for interpreting compound labels such as 3O2A.

The following texts have been added to the main text:

“3O₂A; hereafter, nomenclature nOmA describes the collection of taxane isomers bearing n oxidations, e.g., hydroxylation and epoxidation, and m acylations, e.g., acetylation and benzylation, on the taxadiene scaffold”

2. P450 Nomenclature (T9 α H-750C and T9 α H-725A):

The naming of P450 enzymes incorporates their respective subfamilies, as detailed in the manuscript. For example:

- T9 α H-750C refers to a P450 in the CYP750C family. To clarify this for readers, we included the text: “... we found a P450 in the CYP750C family (T9 α H-750C).”
- T9 α H-725A refers to a P450 in the CYP725A family (text: “... a distantly related CYP725A P450, with <20% identity at the protein level to T9 α H-750C, as the taxane T9 α H (referred to here as T9 α H-725A for distinction).”

3. Other Enzymes (e.g., 2-ODD184, AAE-867.5):

Enzymes such as 2-ODD184 and AAE-867.5 are named according to our internal cataloging system. We have minimized the use of internal designations throughout the manuscript and opted for the more common names (e.g., T1 β H-184 and PCL) wherever possible. Furthermore, to improve clarity, we have added a column to **Table S1** labeled “*Other name used in this study*,” which provides cross-references between these internal designations and the common names that we hope improve the overall readability.

Figure 2a/b: Why is TAX19 shown in the PCC matrix but does not show up in the pathway?

TAX19 was included in the PCC matrix in **Fig. 2b** because it is an C-13-O-acetyltransferase with characterized activity toward lightly oxidized (< 4 hydroxyl) taxanes as shown in *Archives of Biochemistry and Biophysics* 2004, 430, 237–246 and is involved in the biosynthesis of taxusin, as shown later in our manuscript (**Fig. 3b**). However, it remains unclear whether taxusin is a biologically-relevant intermediate (in *Taxus*) leading to Taxol, which does not possess C-13-O-acetyl, or if it is a separate metabolic branch. As noted in our discussion, it is possible that the Taxol pathway proceeds through C-13-O-acetylated intermediates that would require TAX19. In contrast, our reconstituted pathway represents an alternative route involving C-13 α -hydroxyl intermediates, which does not require TAX19, and this is the pathway we chose to present in **Fig. 2a**.

We updated the Figure 2b PCC title to read “PCC between known taxane ~~taxol~~ genes” for clarification

Figure 2i: How is gene score defined? What does it mean?

We have adapted our text to clarify module score and provide a high level, intuitive description of the cNMF (consensus non-negative matrix factorization) pipeline we utilized to produce this score. Though we referenced the analysis package used, we have also made more explicit textual references to the paper that developed cNMF, to help readers find the details of the underlying data processing. We hope the new text (below) helps to clarify this score:

“To systematically organize genes into co-expressed modules, we factored the large gene-by-cell matrix from the mpXsn dataset using a consensus non-negative matrix factorization (cNMF) approach previously developed for single-cell datasets³¹. Factorization approximates the observed dataset as the product of two smaller, meaningful matrices: (i) a gene-module matrix (a weight value for each gene in each module), and (ii) a cell-module matrix (expression values of each module in each cell) (**Fig. 2c**). The weight values of the gene-module matrix can be used as scores that identify the genes that dominate each module; top-scoring genes from the same module have coordinated expression patterns and are likely part of the same molecular processes. This approach adapts to the rich, but noisy data inherent in single cell analysis and reveals patterns of coordinated gene expression that might not be apparent from linear correlation analysis. For example, it allows for genes to be in multiple, overlapping modules, which likely better represents how genes in a highly branched metabolism may be expressed. We ran cNMF analysis with different numbers of total modules (50, 75, 100, 125, 150, 175, 200, 300, or 400); in all cNMF runs with greater than 125 modules, known taxol enzymes consistently dominated not one, but three separate gene modules (subsequently called Taxol modules 1, 2, and 3) (**Fig. 2d-e, 2h-i, S1**). Based on the consistency within this parameter range, we proceeded with an intermediate value (200 total modules) for subsequent analysis to avoid both over-clustering and under-clustering. The observation that different subsets of Taxol enzymes rank highly in separate modules, roughly segregating by proposed order of activity (**Fig. 2e**), suggests that the full Taxol biosynthesis consists of separately regulated transcriptional programs.”

Figure 3a: Stereochemistry of 2'a and 2'b is not shown completely. The presentation of the structures should be improved.

The structures of 2'a and 2'b in Fig. 3 have been updated to accurately depict the stereochemistry at C-3, C-4, and C-11, based on their original characterization as reported in *Journal of Biological Chemistry* 2008, 283, 10, 6067-6075 and *Chemical Science* 2016, 7, 3102-3107.

Figure 2 legend describes the meaning of green and orange colours, but unfortunately this colour code overlaps with the colours of the UMAP plot next to panel h. This might lead to confusion.

We appreciate the reviewer's concern and have changed the color scheme of Fig. 2i to avoid overlap.

The use of species name appears inconsistent. Sometimes the authors write "Taxus x media", sometimes "Taxus media". This should be consistent or explained.

We have standardized all instances to "*Taxus media*" to ensure consistency.

The presentation of NMR data could be improved in several regards:

- no double arrows should be used to indicate HMBC correlations; these are strictly unidirectional from a proton to a carbon (Table S4, Table S5). In the same light, all correlations that are shown should be carefully checked again.

We have checked and revised all representations of HMBC correlation using uni-directional arrows from protons to carbons.

- chemical shifts should be reported with consistent decimal places; please check Table S5 H-3, C-17; Table S3 C-1 ("38?"; what does the question mark mean?)

The reports of chemical shifts have been standardized to ensure consistent decimal places: carbon chemical shifts are presented with one decimal place, while proton chemical shifts are shown with two decimal places. C-1 of 13 β -taxusin (**6'**) in Table S3 was updated from "38?" to "~38 (overlapping)" to better represent its approximate value due to the influence of overlapping signals in the HMBC spectrum.

- authors should report temperature and frequency

General NMR analysis is outlined in "NMR analysis of purified compound" in the Method session, including "spectra were acquired on a Varian Inova 600 MHz or a Bruker Neo 500 MHz spectrometer at room temperature". In addition, specific NMR conditions, such as "CDCl₃, 600 MHz, 298 K", have been incorporated into the titles of each spectrum within the supplementary information for clarity.

- Table S3: Abbreviation N.A. is not defined

We have incorporated the definition, "N.A. = not assigned (due to weak signal intensity or overlapping signals)", into the relevant tables.

There are numerous very minor aspects (e.g., inconsistent use of FoTO1 vs. FOTO1; u instead of μ ; compound numbers not bold; inconsistent reference to carbon atoms with or without hyphen; inconsistent or incorrect use of acyl vs. acetyl). These are highlighted in an annotated pdf file.

We appreciate the reviewer's careful read of the manuscript and the detailed annotation of inconsistencies. We have carefully addressed all the issues raised in the annotated PDF file. Additionally, we have further inspected and revised the manuscript to ensure overall consistency and accuracy in language throughout. Thank you!

G: With the exception of previous work on MSBP1 (see section F), the authors provide appropriate credit to previous work.

H: In the abstract, the authors might want to clarify that the genus name *Taxus* refers to yew trees.

We have revised the abstract to clarify that *Taxus* refers to yew trees by stating "... anti-cancer therapeutic paclitaxel (Taxol®) from the yew (*Taxus*) trees."

Signed
Jakob Franke

Referee #2 (Remarks to the Author):

This manuscript describes a clever and truly insightful study of paclitaxel (Taxol) biosynthesis. In particular, while the pathway to this vitally important anti-cancer agent has been the subject of investigation for decades, efforts to recombinantly transfer this into more amenable hosts than the native producing yews (from the *Taxus* genus) have been stymied not only by its incomplete nature (i.e., not all the necessary genes were known), but also the puzzling metabolic block at the first hydroxylation step, where the relevant enzyme produced relatively little of the desired product with others accumulating instead. This later issue has continued to plague even the recently reported studies from other groups published this year in *Science* (ref. 9) and *Molecular Plant* (ref. 26). Almost of all this is addressed here in what is a breakthrough study not only for this pathway, but that of such plant natural products more generally given the clever approach developed here. Specifically, as with others, these authors carried out RNA-Seq with bulk tissues isolated from yew plants along with elicitation known to induce Taxol production, but had limited (no?) success in identifying any new genes. However, here a novel clever ‘multiplexed’ approach was developed, using multiple perturbations of needles (including various timepoints) and then isolating single nuclei for transcriptome sequencing, which were then grouped by similarity to generate ‘cell states’ and co-expression modules. This led to three modules in which the known Taxol biosynthetic genes were highly ranked that then provided lists of other genes that were hypothesized to also play a role. The validity of this ‘multiplex’ approach were amply demonstrated by the various breakthroughs in Taxol biosynthesis reported here. Note that while the elucidated pathway is ‘only’ to the intermediate baccatin III, this is both the main taxane found in needles, and also the actually isolated compound that is subsequently converted to Taxol in a semi-synthetic route, so should still be considered a resounding success. In the context of the other recent studies, perhaps the most impactful and also surprising finding was the discovery of a scaffolding protein from the NTF2-like protein family (FoTO1) that modified the activity of the first hydroxylase to direct product outcome towards that relevant to taxane biosynthesis. All of the previously unknown enzymes for production of baccatin III also were identified, which revealed an intriguing role for ‘excess’ acylation in the pathway. At least in one case, the identified hydroxylase differs from that identified in the other recent studies. While that might raise questions, it seems most likely that identified here is correct, simply due to the efficiency of the reconstructed pathway reported here, which even in its currently unoptimized form equals that exhibited in yew needles themselves, relative to the rather anemic throughput reported in the other studies. Along these lines, although not a focus here, in the remaining steps from baccatin III to Taxol, the inefficiency of the previously identified 2’a-hydroxylase suggests that might be the relevant enzyme. Identification of the correct enzyme might be beyond the scope of this study, particularly given the uncertainty about Taxol actually being produced in needles (at least in any appreciable amounts), but seems worth mentioning. In any case, even beyond the important insights obtain into Taxol biosynthesis and strong potential for impact on commercial supply of this important anti-cancer agent, the developed multiplex approach to investigating plant biosynthetic pathways is expected to be widely applicable, adding further impact to this report.

The presentation is quite good, but could be bolstered in a few spots. For example, while the use of baccatin III in Taxol production is mentioned in the abstract and discussion, it should be noted in the introduction (first or last paragraph) and references provided (these should also be presented in the discussion).

We have revised text in the first paragraph of the introduction to note the use of baccatin III in Taxol production with reference to the original BMS semi-synthesis patent. “While many elegant synthetic routes have been developed, none are economically viable; drug supply still relies on the extraction of late-stage intermediates, such as baccatin III, from yew tissue sourced from plant cell cultures or farming.”

The issue about the previously identified 2'a-hydroxylase also should be raised at least in the results if not also discussion.

We have discussed the T2'a-hydroxylase in the final paragraph of our Results section "Side-chain biosynthetic enzymes enable total biosynthesis of 3'-N-debenzoyl-2'-deoxypaclitaxel in *Nicotiana* leaves". In this section, we note that we attempted to incorporate the published T2'a-hydroxylase gene into the final steps of our pathway, but were unable to see activity.

With regard to the multiplexed approach, it seems to be implied that the needles were only elicited for a short time and then left in MS media until harvest (at the various time points), but this should be clarified and, if true, the exposure time specified in the methods. Also, it is not specified how the nuclei were then isolated from these needles (even if the same as with the bulk tissue samples that should be stated). Finally, a bit more explanation about how the tissues were 'chopped at ~200 rpm' should be provided (e.g., was this done with a machine or by hand?).

We appreciate the suggestions to clarify our Methods section. To address these concerns, we have clarified in this section that tissues were subjected to elicitation in MS media supplemented with each elicitor; tissues were incubated in this altered MS media for 1-4 days. We have also clarified how tissues were pooled after elicitation, prior to nuclei isolation, and that tissues were chopped by hand with a razorblade.

Referee #3 (Remarks to the Author):

This manuscript from the Sattely Lab represents an exceptional contribution to the field of plant natural product biosynthesis, and in particular the elusive pathway to Taxol. McClune et al. present a polished body of work that taken together provides a convincing, wholistic, and comprehensive story around baccatin III biosynthesis. In addition to identifying seven new genes (including an intriguing NTF2-like accessory), the group presents a newly developed "multiplexed perturbation" strategy. This approach will be adapted in future studies where bulk tissue RNA-seq fails to identify missing genes. The work both acknowledges overlapping scientific discoveries which have been recently made, while rectifying the disparate conclusions reached as a result of the complex nature of this biosynthetic pathway.

The authors came very close to de novo production of Taxol in *N. benthamiana* – which would have pushed this paper to the very top in terms of impact to a broader audience. However, the authors managed to produce baccatin III, an industrial precursor for Taxol, at titers higher than in native yew needles; this could have a real-world commercialization potential. Moreover, the authors identified independently regulated expression modules and provided evolutionary insight, two compelling premises in plant biosynthesis. If the following concerns can be addressed, I would recommend publication in this journal.

Major concerns:

Major concern 1 – The authors should clearly indicate which structures are proposed and which have been confirmed by NMR. Only the structures of 6, 6', 6-O1, and 6-O2 are confirmed the NMR, the rest are proposed via comparison to authentic standards or mass spectra; these techniques cannot provide conclusive evidence of molecular structure. The use of the phrase "structural analysis" implies NMR characterization. Minimally, the authors should perform NMR characterization of titular baccatin III isolated from *N. benthamiana*, to bolster the claim that "this work represents a major step towards sustainable production of Taxol."

We appreciate the reviewer's feedback regarding the differentiation between proposed and confirmed structures, and we have taken steps to clarify this in the manuscript. Specifically:

1. **Figure Updates:** To avoid confusion between structural analysis and comparisons to standards, we have revised Fig. 3, 4, and 5. Structures confirmed by NMR are now clearly marked with a filled diamond symbol with text "Diamond indicates the structure is supported by NMR." in the figure legend. This visual distinction ensures that readers can easily differentiate between confirmed and proposed structures.

2. **Fig. 6f Identification Methods Summary:** We have updated the summary of the Taxol biosynthetic pathway in **Fig. 6f** to include italicized text beneath each structure to specify evidence of structural assignment with the following text in figure legend: "Structures are assigned using the indicated methods: HRMS (predicted chemical formula based on high-resolution mass spectrometry), NMR (1D- and 2D-NMR structural analysis followed by purification), and/or HRMS/MS (comparison of MS/MS spectra to authentic standards)."

3. **Baccatin III Purification:** To further support our claim that "this work represents a major step towards sustainable production of Taxol," we have partially purified baccatin III produced via heterologous expression of the complete pathway in *N. benthamiana*. While impurities are present, all signature peaks are clearly distinguishable in the spectra. ¹H NMR of baccatin III is provided in **Extended Data Figure 7b** with the following legend:

¹H-NMR spectra of partially purified baccatin III (**16**) from *N. benthamiana* and baccatin III (**16**) standard (CDCl₃, 500 MHz, 298 K). The spectra of our partially purified baccatin III (**16**) align with the standard, exhibiting all characteristic peaks (labeled with carbon number) as well as the H-20 coupling constant ($J = 8.3$ Hz) of the oxetane. The full baccatin III (**16**) pathway excluding T5αH was used to infiltrate 53 *N*.

benthamiana plants [30.70 g dry weight (DW)] to yield baccatin III (**16**), whose yield (~270 µg) is derived from the total yield (1.33 mg) with an estimated 20% purity. “

Major concern 2 – The authors must carefully explain why they chose the following cutoffs to facilitate gene identification and analysis using their new mpXsn methodology, otherwise it may appear to a reader that these cutoffs were arbitrarily selected to make the discovery/technique appear more straightforward:

- Why were 200 total gene expression modules analyzed?
- How were “many” and “top” defined when determining the cutoffs to identify modules (ie. “we found three where the TOP genes included MANY known Taxol enzymes”) and why were these values selected? (For example there appears to be another “module 4” with five Taxol enzymes, which is precluded for analysis based on these cutoffs.)
- Why was the value 100 chosen to rank the modules by specialized metabolism in Fig. 2g?
- Why was the value 125 chosen for number of unfiltered genes to display in Fig. 2h?

We are thankful for the reviewers' clarifying question and we would like to provide a detailed explanation of the numerical choices used in different parts of our analysis. Specifically, we address: (1) the selection of 200 total gene expression modules, (2) the criteria for identifying Taxol modules, and (3) the rationale behind the values chosen in the sub-figures. To support this, we have revised the “Identification of three Taxol biosynthetic gene modules” paragraph, added **Fig. S1** (displayed below), and provided a **Supplementary File** as additional resources for interested readers.

Defining the number (200) of gene expression modules

We have added the following text to clarify our choice of 200 modules:

“We ran cNMF analysis with different numbers of total modules (50, 75, 100, 125, 150, 175, 200, 300, or 400); in all cNMF runs with greater than 125 modules, known taxol enzymes consistently dominated not one, but three separate gene modules (subsequently called Taxol modules 1, 2, and 3) (**Fig. 2d-e, 2h-i, S1**). Based on the consistency within this parameter range, we proceeded with an intermediate value (200 total modules) for subsequent analysis to avoid both over-clustering and under-clustering. The observation that different subsets of Taxol enzymes rank highly in separate modules, roughly segregating by proposed order of activity (**Fig. 2e**), suggests that the full Taxol biosynthesis consists of separately regulated transcriptional programs..”

Defining criteria for identifying Taxol modules

After defining the total number of modules, we chose to focus on three modules as Taxol module because they:

- I. arose consistently when cNMF was run with various module number k (**Fig. S1**)
- II. were consistently dominated by known Taxol when cNMF was run with various module number k
- III. were enriched in specialized metabolism enzymes (**Fig. 2g**)
- IV. were generally highly ranked components from the matrix factorization. Analogous to the ranked of components in PCA, lower ranking modules explain less of the transcriptome data. With k=200 total modules, the three modules we refer to as Taxol modules 1-3 were ranked within the top 50 modules, while the “module 4” noted by reviewers ranked 110 and did not appear in most other cNMF runs, suggesting that it may have been a minor residual of another module.

Rationale of using specific value (top 100) in sub-figures

After defining the three Taxol modules, we focuses on the top 100 genes in each module due to three observations:

- I. While module “sizes” were different, in that the scores declined at different rates, examining the top 100 genes for most modules captures most of the high scoring genes of our Taxol modules. Scores have declined by 80-90% by gene ranked 100.
- II. With a pathway of ~25 enzymes and 2-4 near-identical variants of each known enzyme, we anticipated that a complete co-expression cluster for the Taxol pathway would contain approximately 75 catalytic genes. Taking into account the number of co-expressed accessory proteins like transcription factors or transporters suggests that a Taxol module would likely contain no less than 100 genes.
- III. The 16 previously known Taxol enzymes (and their many near-identical variants) densely occupied the top 50 genes of module 1 and 100 genes of module 3.

Based on the above observations, we chose the cutoffs used in the following sub-figures for best visualization in each case:

- **Fig. 2e and 2f.** Rank top 200 was used to allow ease of distinguishing the top 100 genes (light to dark blue) versus the 100-200 genes (white to light blue) on a colored gradient heatmap.
- **Fig. 2g.** Rank top 100 was used for analysis of specialized metabolism enzymes content as discussed above defining the gene rank cutoff for module analysis.
- **Fig. 2h/i.** Rank top 125 was used to plot gene scores to help visualize the different effective sizes of different modules: while all modules technically contain all genes, the scores for the top genes of different modules decay at different rates. In other words, different modules have different numbers of highly ranked genes. For Taxol modules 1 and 3, an “elbow” in this gene score plot is visible around gene 50. However, the scores decay more slowly for Taxol module 2, suggesting it is a larger module than Taxol modules 1 and 3. Plotting slightly further beyond top 100 helps visualize the difference in score decay more easily. We chose 125 as it conveniently includes three more known Taxol genes in module 2, our largest module.

While our analysis primarily focuses on the top 100 genes in the Taxol modules, we have now included a **Supplementary File** containing the top 2,000 genes and their scores for each Taxol module, allowing interested readers to examine the lower-ranked genes of these modules in more depth.

Fig. S1. Gene modules with different parameters for matrix factorization. Consensus non-negative matrix factorization (cNMF)²⁵ was run with different numbers of modules (k). The ranking of 17 genes previously shown to be associated with taxane biosynthesis (x-axis) were visualized for all modules produced by each cNMF run. When $k \geq 150$, these known taxane enzymes consistently cluster into three dominant modules. In each plot, modules are ranked by total usage in our dataset, as defined by the cNMF analysis (e.g. module #1 from each run has the

highest usage, or expression, across cells in the dataset).

Minor concerns:

Minor concern 1 – The authors should add a TOC to the supplementary information.

We have included a TOC at the beginning of the supplementary information document.

Minor concern 2 – The authors use FoTO1 and FOTO1 interchangeably.

We have addressed the inconsistency by standardizing the use of “FoTO1” throughout the manuscript.

Minor concern 3 – The authors state, “When examining the 14 genes previously established to be Taxol enzymes (Table S1, Fig 2a), all but TBT were prioritized substantially higher using the mpXsn data rather than bulk RNA-seq data (Fig. 1f, S2-S3).” However besides TBT, DBTNBT also appears to be prioritized lower.

The reviewer is correct in pointing out that both TBT and DBTNBT are prioritized lower using bulk RNA-seq data and we have revised the sentence to “all but two genes were prioritized higher using the mpXsn data compared to bulk RNA-seq data (Fig. 1f, Extended Data Figure 1b-d).”

Minor concern 4 – The authors refer to “co-expressed subsets of the pathway (Fig. 2b)” but it is not clear to me from looking at Fig. 2b what is being referred to.

“Co-expressed subsets of the pathway” was referring to the formation of multiple co-expressed clusters among all Taxol genes, versus a single highly co-expressed cluster encompassing all Taxol genes. To clarify the ambiguity, we have revised the sentence as follows:

“While the expression of the first enzyme in the pathway, TDS, does correlate to most Taxol genes (Fig. 1f, Extended Data Figure 1c-e), some known Taxol genes show stronger co-expression relationships with one another, forming distinct clusters of co-expression within the pathway (Fig. 2b).”

Minor concern 5 – The authors should describe how many candidates they tested by “coexpressing them with this six-enzyme early pathway in *N. benthamiana*” in order to identify gene #13 FoTO1, and ideally present this data.

We iteratively tested a large number of genes from module 1, most of which ranked lower than FoTO1 in this module, because our initial efforts focused on genes with precedence for taxane metabolism roles like cytochromes p450s, 2ODDs and acetyltransferases. This was driven by initial hypotheses that perhaps another enzyme was needed to replace T5aH or function in concert with it.

We have expanded and updated the text describing our discovery of FoTO to clarify how many genes in total were tested from Module 1:

“We hypothesized that Taxol module 1 might contain previously undiscovered proteins that could facilitate this initial oxidation and prevent the formation of side-products. In the course of our work testing approximately 77 gene candidates highly ranked in Taxol module 1 (Fig. 3b), we found that the addition of a single protein, a NTF2-like protein ranked #13 in module 1, increased the yield of a putative taxane product (Fig. S3). We hypothesized that this protein may be involved in altering product flux of the early oxidative steps.”

Furthermore, we have now added a supplementary figure (S3) showing the results of the our pooled candidate screen in which FoTO1 was discovered:

Minor concern 6 – The authors should explain how they concluded that 2'e is a side product of T5αH, as well as the nature of the 2'd more carefully.

We appreciate the reviewer's comments regarding the identification and characterization of compounds 2'd and 2'e. Below, we address these points in detail:

1. Compound 2'd:

As reported in our previous work (*Nat. Commun.* 2024, 15, 1419), compound 2'd (compound 9 in previous manuscript) is derived from taxadien-5α-ol (2) based on a feeding study where 2 was infiltrated into leaves expressing T5αH. From *Nat. Commun.* 2024, **Figure 2d**: [REDACTED]

[FIGURE REDACTED]

We have now successfully purified 2'd and confirmed its structure as 4α,20-epoxy-taxadien-5α-ol. This detailed characterization using MS and 2D-NMR analysis, is now provided in **Table S3**, **Fig. S4**, and **Fig. S47-52**. The accumulation of 2'd suggests that FoTO1 facilitates the epoxidation by T5αH and/or stabilizes the epoxide product, which is consistent with previously proposed T5αH epoxidation mechanism. Detailed mechanistic discussion of T5αH and comparison with T13αH is provided in **Supplementary Note 1**.

Table S3. ^{13}C and ^1H δ assignments as well as 2D-NMR correlations of $4\alpha,20$ -epoxy-taxadien- 5α -ol (**2'd**) recorded in CDCl_3 .

We have added the following discussion to the main text describing our structural analysis of **2'd**:

“We have previously observed **2'd** as the main product when taxadien- 5α -ol (**2**) is fed to *N. benthamiana* expressing T5aH¹⁵, suggesting that it is a secondary oxidation that occurs when T5aH incubates with its product **2**. Here, we confirmed this, through purification from *N. benthamiana* leaves expressing TDS, T5aH and FoTO. 1D- and 2D- nuclear magnetic resonance (NMR) analysis supports a structural assignment of **2'd** as $4\beta,20$ -epoxy-taxadien- 5α -ol (Fig. 3a, Table S3, Supplementary Note 1). It is notable that the stereochemistry of the $4\beta,20$ -epoxy in **2'd** has not been previously reported in any natural taxanes, which exclusively features a $4\alpha,20$ -epoxy stereochemistry. It is possible that the formation of **2'd** is due to overoxidation of taxadien- 5α -ol (**2**) that accumulates to an unnaturally elevated level by T5aH in this partially reconstituted pathway.”

2. Compound 2'e:

While we were unable to obtain sufficient quantities of pure **2'e** suitable for NMR analysis due to its low abundance. However, the MS spectra of **2'e** (Fig. S4) reveals a fragmentation pattern consistent with taxanes, particularly in the region m/z 50–150. The observed parent ion mass (m/z 274) suggests that **2'e** could potentially be a saturated form (+2H) of taxadiene. Additional work will be required to confirm its identity and significance in FoTO1-T5aH catalysis.

The following text has been added to the caption for Fig. S3:

“While we were not able to obtain sufficient amounts of **2'e** for structural characterization, the MS spectra of **2'e** reveals a fragmentation pattern consistent with taxanes, particularly in the region m/z 50–150. The observed parent ion mass (m/z 274) suggests that **2'e** could potentially be a saturated form (+2H) of taxadiene. Additional work will be required to confirm its identity and significance in FoTO1-T5aH catalysis.”

Minor concern 7 – The authors should describe how many candidates they tested when they “further screened the oxidases (2-ODDs and P450s) of module 2, revealing two 2-ODDs (2-ODD-184 and 2-ODD-686) that yielded single-oxidized products when expressed with various upstream pathways” in order to identify T1 β H-686 and T1 β H-184, and ideally present this data.

We have noted the total number of oxidase gene candidates that were screened from module 2 in an effort to identify the missing T1 β H gene. Furthermore, we have included data on pooled acetylase screening and the discovery of T1BH-686 in a new supplementary figure (Fig. S14):

Minor concern 8 – The authors should include a negative control in Fig. 4c baccatin VI trace (ie. the EIC of 737.2787 without TAX19).

We appreciate the helpful comments by the reviewer and we have included the negative controls where TAX19 is absent for both 11-Ac and 11-Bz (baccatin VI) in Fig. 4c.

Minor concern 9 – The authors should describe how many αβ-hydrolase candidates they in order to identify DeAc898 and DeAc1023, how these were selected, and ideally present this data.

We have updated the main text to note the total number of αβ-hydrolase candidates that were screened at any point of this pathway discovery project. We thank the reviewer for pointing out that we did not mention that we selected these genes based on their presence in our Taxol modules and have corrected the main text to clarify this. Furthermore, we have included data on pooled acetylase screening and the discovery of T9dA in a new supplementary figure (Fig. S16):

Minor concern 10 – The authors should make Fig. 5g more clear – as drawn the reader may mistake the (17) chromatograms for (18). I would suggest labelling the peaks with compound numbers as the structures are just above.

We appreciate the feedback and have revised Fig. 5g accordingly – adding compound number 17 directly above the peak – to avoid confusion.

Minor concern 11 – The authors mention that “all cytochromes P450 require a cytochrome P450 reductase (CPR) partner” but this is not the case.

The reviewer is correct in pointing out that not all P450s require CPR as some self-sufficient P450s do not require CPR. We have revised the text to “While *most* P450s require a...”

Minor concern 12 – The authors write that FoTO1 activity represents “the first example, to our knowledge, of a plant cytochrome P450 that acts in concert with an additional protein to guide product specificity.” In my opinion the phrase “in concert with” does not have a meaningful definition and should not be used, as it could be argued that any enzyme downstream of a P450 “guides” product specificity “in concert” with the P450.

We agree with the assessment of the reviewer and have removed “in concert” in the sentence. The sentence was later revised for clarity into:

“Our discovery of FoTO1 and its interaction with T5αH challenges a long-standing assumption that T5αH is the sole actor in the first oxidative steps of the Taxol pathway. While most P450s require interaction with a cytochrome P450 reductase (CPR) partner, only a few have been reported to associate with non-enzymatic scaffold proteins.”

Minor concern 13 – The authors claim “FoTO1 leads to such profound improvements in the activity of multiple *Taxus* oxidases” but also mentioned “FoTO1 may be involved in altering product flux directly at the first oxidation of taxadiene (1), as the yield improvement did not increase as the sub-pathway was extended beyond T5αH (Fig. 3c)” – which is it?

We thank the reviewer for pointing out the confusion on “multiple *Taxus* oxidases” and “first oxidation of taxadiene (1)...T5αH (Fig. 3c)”. We have revised both sentences to read:

“FoTO1 leads to such profound improvements in the activity of *Taxus* oxidases *T5αH* and *T13αH*” and “We hypothesized that FoTO1 may be involved in altering product flux of the *early oxidative steps*.”

This more accurately represent the fact that both T5αH and T13αH are capable of the oxidation of taxadiene (1) (Fig 6d, Extended Data Figure 9). Additionally, we have provided a detailed discussion on the activity of T5αH and T13αH in **Supplementary Note 1**.

Referee #1 (Remarks to the Author):

I thank the authors for their very careful and extensive revisions. All my requests have been fully addressed. I congratulate the authors to this outstanding work and can now fully endorse publication of this manuscript.

We sincerely thank the reviewers for their invaluable feedback, which has greatly improved our manuscript. Thank you for your time and effort in evaluating our work!

Referee #2 (Remarks to the Author):

This manuscript has been suitably revised to address my previous concerns. However, there are a few minor points. First, it might help to emphasize the importance of the produced levels of baccatin III by noting the commercially farmed source are the needles (in the intro and discussion), to which this unoptimized pathway already compares. Second, when introducing the nOmA compound nomenclature used here it might be more accurate to term what is currently called “oxidations” as “oxy groups”. Third, it might be helpful to note the sequence identity between 2-ODD184 and 2-ODD686 (i.e., upon their introduction in the results). Finally, please clarify if NMR was used to identify the “structurally characterized ... 4 α ,20-epoxy-5 α -hydroxy-taxadien-13-one”.

We appreciate the reviewer’s additional comments and have revised the manuscript to address the four concerns. Below are the revised texts:

- Noting the commercial source of baccatin III is the needles in the intro and discussion.
 - Intro: *“These modules resolved seven new genes allowing the de novo 17-gene biosynthesis and isolation of baccatin III, the industrial precursor for Taxol, in Nicotiana benthamiana leaves at levels comparable to Taxus needles.”*
 - Discussion: *“Without optimization, our reconstituted 17-gene pathway yields 10-30 μ g/g baccatin III (16) in N. benthamiana leaves, equivalent to its natural abundance in Taxus media needles (Extended Data Figure 7d). As baccatin III (16) extracted from Taxus is the primary precursor for industrial semi-synthesis, our work represents a major step towards sustainable production of Taxol and other taxane-based therapeutics.”*
- “Oxy groups” instead of “oxidations” in nOmA nomenclature.
 - *“...hereafter, nomenclature nOmA describes the collection of taxane isomers bearing n oxy groups, e.g., hydroxylation and epoxidation, and m acylations, e.g., acetylation and benzylation, on the taxadiene scaffold.”*
- Note the sequence identity between 2-ODD184 and 2-ODD686.
 - *“This revealed two 2-ODDs (2-ODD184 and 2-ODD686, protein sequence identity 72%)...”*
- Clarify if NMR was used to identify “4 α ,20-epoxy-5 α -hydroxy-taxadien-13-one”.
 - *“We structurally characterized an over-oxidized derivative, 4 α ,20-epoxy-5 α -hydroxy-taxadien-13-one (Supplementary Note 3, Table S8-9), and a TAX19*

derivative, 5 α ,13 α -diacetoxy-taxadiene (Extended Data Figure 8d, Table S10) by NMR...”

Referee #3 (Remarks to the Author):

All concerns have been addressed.

We sincerely thank the reviewers for their invaluable feedback, which has greatly improved our manuscript. Thank you for your time and effort in evaluating our work!

Referee #4 (Remarks to the Author):

I was asked to assess the single nucleus sequencing analysis specifically. My view is that this has been done carefully, and agree that this multiplexed approach is both clever and informative - borne out by the subsequent analysis of candidates and validation of function in the pathway. The single nucleus section is also well explained to the reader.

We thank the reviewer for assessing the single nucleus sequencing section of the manuscript.